# ON THE IMPACT OF THE UTILITY IN SEMIVALUE-BASED DATA VALUATION

**Mélissa Tamine** *
Criteo AI lab, FairPlay joint team, France
CREST, ENSAE, Institut Polytechnique de Paris
`m.tamine@criteo.com`

**Benjamin Heymann**
Criteo AI lab, FairPlay joint team, France
`b.heymann@criteo.com`

**Maxime Vono**
Criteo AI lab, FairPlay joint team, France
`m.vono@criteo.com`

**Patrick Loiseau**
Inria, FairPlay joint team, France
`patrick.loiseau@inria.fr`

## ABSTRACT

Semivalue–based data valuation uses cooperative-game theory intuitions to assign each data point a *value* reflecting its contribution to a downstream task. Still, those values depend on the practitioner's choice of utility, raising the question: *How robust is semivalue-based data valuation to changes in the utility?* This issue is critical when the utility is set as a trade-off between several criteria and when practitioners must select among multiple equally valid utilities. We address this by introducing the notion of a dataset's *spatial signature*: given a semivalue, we embed each data point into a lower-dimensional space in which any utility becomes a linear functional, making the data valuation framework amenable to a simpler geometric picture. Building on this, we propose a practical methodology centered on an explicit robustness metric that informs practitioners whether and by how much their data valuation results will shift as the utility changes. We validate this approach across diverse datasets and semivalues, demonstrating strong agreement with rank-correlation analyses and offering analytical insight into how choosing a semivalue can amplify or diminish robustness.

## 1 INTRODUCTION

Supervised machine learning (ML) relies on data, but real-world datasets often suffer from noise and biases as they are collected from multiple sources and are subject to measurement and annotation errors (Northcutt et al., 2021). Such variability can impact learning outcomes, highlighting the need for systematic methods to evaluate data quality. In response, *data valuation* has emerged as a growing research field that aims to quantify individual data points' contribution to a downstream ML task, helping identify informative samples and mitigate the impact of low-quality data. A popular approach to tackling the data valuation problem is to adopt a cooperative game-theoretic perspective, where each data point is modeled as a player in a coalitional game, and the usefulness of any data subset is measured by a *utility function*. This approach leverages game theory solution concepts called *semivalues* (Dubey et al., 1981), which input data and utility to assign an importance score to each data point, thereby inducing a ranking of points in the order of their contribution to the ML task (Ghorbani & Zou, 2019; Kwon & Zou, 2022; Wang & Jia, 2023; Jia et al., 2019; Wang et al., 2024a).

**Motivation.** When computing semivalues, the utility is typically selected by the practitioner to reflect the downstream task. In some contexts, this choice is obvious. For example, when fine-tuning a large language model (LLM), one might balance two competing objectives: helpfulness (how well the model follows user instructions) and harmlessness (its tendency to refuse or safely complete unsafe requests) (Bai et al., 2022a;b). If the practitioner then asks, "Which training examples most contributed to my desired helpfulness–harmlessness balance?", the only sensible utility for semivalue-based data valuation is this composite trade-off itself. By contrast, in more open-ended tasks, the

---

*Corresponding author.

utility can be genuinely ambiguous. Imagine training a dog vs. cat image classifier and asking, "Which data points contributed most to overall performance?" Then, accuracy, precision, recall, F1, AUROC, balanced accuracy, and many others are all defensible choices. However, none is uniquely dictated by the task.

These two example settings respectively motivate two general scenarios: (1) the *utility trade-off* scenario, where the utility is a convex combination of fixed criteria, with a tunable weight $\nu$, and (2) the *multiple-valid-utility* scenario, where the utility must be chosen among several equally defensible metrics, none of which being uniquely dictated by the task. In both cases, we argue that data valuation practitioners are well advised to ask themselves:

*How robust are my data valuation results to the utility choice?*

In what follows, we explain why.

***Utility trade-off* scenario: Anticipating costly re-training.** In scenarios where the utility itself is a trade-off, e.g., when fine-tuning an LLM by combining helpfulness and harmlessness into a single objective parameterized by $\nu$, practitioners often rely on data values to identify the $k$ most valuable training examples and train next models on this smaller subset to reduce computational cost as is common practice when using data valuation (Ghorbani & Zou, 2019; Jiang et al., 2023). However, if the top-$k$ set shifts dramatically with changes in $\nu$, as priorities between harmlessness and helpfulness evolve, practitioners risk repeated, costly re-training. Quantifying robustness to utility choice makes this risk explicit, alerting practitioners up front to whether data valuation is a safe, one-time investment or whether they must plan for ongoing computational overhead as their utility trade-off evolves.

***Multiple-valid-utility* scenario: Detecting when data valuation fails as a heuristic.** In many real-world tasks, like the earlier dog vs. cat classifier example, practitioners must select a utility from several valid options, none uniquely dictated by the problem. Now, one would "morally" expect their induced orderings of points to be consistent. After all, each utility is a valid measure for the same task. It may be hard to think that a data point deemed highly important under accuracy would suddenly vanish from the top tier under F1-score, or vice versa. In practice, however, we observe precisely such discrepancies, depending on both the training dataset and the semivalue. We compute data values under both accuracy and F1-score on several public datasets, using three popular semivalues: Shapley (Ghorbani & Zou, 2019), $(4, 1)$-Beta Shapley (Kwon & Zou, 2022), and Banzhaf (Wang & Jia, 2023) (see Appendix A.1 for experimental details). Table 1 reports the Kendall rank correlation between the two score sets for each combination of dataset and semivalue. Low correlations reveal cases where rankings change substantially depending on whether accuracy or F1-score is used as the utility [1]. And because no utility is inherently better, a practitioner has no principled way to choose between the data values ranking produced under accuracy versus F1-score (or any other valid utility). In such settings, arbitrary utility choices can drive the ordering of data points in entirely different directions: the context (dataset + semivalue) is therefore not "data-valuationable", and semivalue-based data valuation *fails* as a reliable heuristic. By contrast, if rankings remain consistent across all valid utilities for the task, data values truly capture the underlying importance ordering of points. Therefore, knowing how robust the scores ranking is to the utility choice enables a practitioner to determine whether semivalue-based data valuation can be trusted as a meaningful heuristic in that context, or whether it is too sensitive to utility to provide reliable guidance.

In this paper, we propose a methodology that enables data valuation practitioners to assess how robust their semivalue-based data valuation results are to the utility choice, in both scenarios. We summarize our main contributions as follows.

1. **Unified geometric modeling of the two scenarios.** We observe that the same geometric representation can capture both scenarios. In this representation, we can, given a semivalue, embed each training data point into a lower-dimensional space (we call the set of embedded points the dataset's *spatial signature*) where any utility becomes a linear functional, making the data valuation framework amenable to a simpler geometric interpretation.

2. **A robustness metric derived from the geometric representation.** Building on the notion of *spatial signature*, we introduce a metric that practitioners can compute to quantify how

---

[1] We extend these experiments to additional classification utilities and rank correlation metrics for completeness (see Appendices A.2 and A.3), and observe variability as well.

Table 1: Mean Kendall rank correlation (± standard error) between data values computed with accuracy versus F1-score. For each semivalue and dataset, we approximate data values 5 times via Monte Carlo sampling. Standard errors reflect the variability across these 5 trials.

| Dataset | Semivalue | | |
|---|---|---|---|
| | Shapley | $(4, 1)$-Beta Shapley | Banzhaf |
| Breast | 0.95 (0.003) | 0.95 (0.003) | 0.97 (0.008) |
| Titanic | −0.19 (0.007) | −0.17 (0.01) | 0.94 (0.003) |
| Credit | −0.47 (0.01) | −0.44 (0.02) | 0.87 (0.01) |
| Heart | 0.64 (0.006) | 0.68 (0.004) | 0.96 (0.003) |
| Wind | 0.81 (0.008) | 0.82 (0.008) | 0.99 (0.002) |
| Cpu | 0.59 (0.02) | 0.62 (0.02) | 0.86 (0.007) |
| 2dplanes | 0.38 (0.01) | 0.44 (0.01) | 0.75 (0.03) |
| Pol | 0.67 (0.02) | 0.77 (0.01) | 0.40 (0.04) |

robust the data values' orderings are to the utility choice, providing a practical methodology for assessing the robustness of data valuation results.

3. **Empirical evaluation of robustness and insights.** We compute the robustness metric across multiple public datasets and semivalues and find results consistent with our rank-correlation experiments: contexts with low rank correlation exhibit a low robustness score, and vice versa. Moreover, we observe that Banzhaf consistently achieves higher robustness scores than other semivalues, a phenomenon for which we provide analytical insights.

**Related works.**  Our focus on the robustness of semivalue-based data valuation to the utility choice differs from most prior work, which has concentrated on *defining* and *efficiently computing* data valuation scores. The Shapley value (Shapley, 1953; Ghorbani & Zou, 2019), in particular, has been widely studied as a data valuation method because it uniquely satisfies four key axioms: linearity, dummy player, symmetry, and efficiency. Alternative approaches have emerged by relaxing some of these axioms. Relaxing efficiency gives rise to the semivalue family (Dubey et al., 1981), which encompasses Leave-One-Out (Koh & Liang, 2017), Beta Shapley (Kwon & Zou, 2022), and Data Banzhaf (Wang & Jia, 2023), while relaxing linearity leads to the Least Core (Yan & Procaccia, 2021). Extensions such as Distributional Shapley (Ghorbani et al., 2020; Kwon et al., 2021) further adapt the framework to handle underlying data distributions instead of a fixed dataset. On the algorithmic front, exact semivalue computation is often intractable, as each semivalue requires training models over all possible data subsets, whose number grows exponentially with the dataset size. Consequently, a rich literature on approximation methods has emerged to make data valuation practical at scale (Mann & Shapley, 1960; Maleki, 2015; Jia et al., 2019; Ghorbani & Zou, 2019; Wang et al., 2024a; Garrido Lucero et al., 2024). By contrast, when and why data valuation scores remain consistent across different utilities has received far less attention. Prior work has explored related robustness questions in special cases: Wang & Jia (2023) examines how semivalue–based valuations fluctuate when the utility function is corrupted by inherent randomness in the learning algorithm, and Wang et al. (2024b) studies how different choices of utility affect the reliability of Data Shapley for data-subset selection. Our proposed methodology broadens this scope by quantifying when data valuation remains robust to shifts in the utility function. Specifically, we independently observe the same sensitivity to utility specification as in Diehl & Wilson (2025), which argues that semivalue-based data valuation can be arbitrary and gameable under utility underspecification. While this work focuses on exposing this vulnerability and its implications, we develop a practical geometric framework that quantifies the sensitivity itself and measures its impact on ranking stability.

**Notations.**  We set $\mathbb{N}^* = \mathbb{N} \setminus \{0\}$. For $n \in \mathbb{N}^*$, we denote $[n] := \{1, .., n\}$. For a dataset $\mathcal{D}$, we denote as $2^{\mathcal{D}}$ its powerset, i.e., the set of all possible subsets of $\mathcal{D}$, including the empty set $\emptyset$ and $\mathcal{D}$ itself. For $d \in \mathbb{N}^*$, we denote $\mathcal{X} \subseteq \mathbb{R}^d$ and $\mathcal{Y} \subseteq \mathbb{R}$ as an input space and an output space, respectively.

## 2 BACKGROUND

The data valuation problem involves a dataset of interest $\mathcal{D} = \{z_i = (x_i, y_i)\}_{i \in [n]}$, where for any $i \in [n]$ each $x_i \in \mathcal{X}$ is a feature vector and $y_i \in \mathcal{Y}$ is the corresponding label. Data valuation aims to

assign a scalar score to each data point in $\mathcal{D}$, quantifying its contribution to a downstream ML task. These scores will be referred to as *data values*.

**Utility functions.** Most data valuation methods rely on *utility functions* to compute data values. A utility is a set function $u : 2^{\mathcal{D}} \to \mathbb{R}$ that maps any subset $S$ of the training set $\mathcal{D}$ to a score indicating its usefulness for performing the considered task. Formally, this can be expressed as $u(S) = \texttt{perf}(\mathcal{A}(S))$, where $\mathcal{A}$ is a learning algorithm that takes a subset $S$ as input and returns a trained model, and $\texttt{perf}$ is a metric used to evaluate the model's ability to perform the task on a hold-out test set. For convenience, we interchangeably refer to the utility $u$ and the performance metric $\texttt{perf}$ as $u$ inherently depends on $\texttt{perf}$.

**Semivalues.** The most popular data valuation methods assign a value score to each data point in $\mathcal{D}$ using solution concepts from cooperative game theory, known as semivalues (Dubey et al., 1981). The collection of methods that fall under this category is referred to as *semivalue-based data valuation*. They rely on the notion of *marginal contribution*. Formally, for any $i, j \in [n]$, let $\mathcal{D}_j^{\setminus z_i}$ denote the set of all subsets of $\mathcal{D}$ of size $j - 1$ that exclude $z_i$. Then, the marginal contribution of $z_i$ with respect to samples of size $j - 1$ is defined as

$$\Delta_j(z_i; u) := \frac{1}{\binom{n-1}{j-1}} \sum_{S \subseteq \mathcal{D}_j^{\setminus z_i}} u\left(S \cup \{z_i\}\right) - u(S) .$$

The marginal contribution $\Delta_j(z_i; u)$ considers all possible subsets $S \in \mathcal{D}_j^{\setminus z_i}$ with the same cardinality $j - 1$ and measures the average changes of $u$ when datum of interest $z_i$ is removed from $S \cup \{z_i\}$.

Each semivalue-based method is characterized by a weight vector $\omega := (\omega_1, \ldots, \omega_n)$ and assigns a score $\phi(z_i; \omega, u)$ to each data point $z_i \in \mathcal{D}$ by computing a weighted average of its marginal contributions $\{\Delta_j(z_i; u)\}_{j \in [n]}$. Specifically,

$$\phi(z_i; \omega, u) := \sum_{j=1}^{n} \omega_j \Delta_j(z_i; u). \tag{1}$$

Below, we define the weights for three commonly used semivalue-based methods. Their differences in weighting schemes have geometric implications discussed in Section 4.1.

**Definition 2.1.** *Data Shapley* (Ghorbani & Zou, 2019) is derived from the *Shapley value* (Shapley, 1953), a solution concept from cooperative game theory that fairly allocates the total gains generated by a coalition of players based on their contributions. In the context of data valuation, Data Shapley takes a simple average of all the contributions so that its corresponding weight vector $\omega_{\text{shap}} = (\omega_{\text{beta},j})_{j \in [n]}$ is such that for all $j \in [n]$, $\omega_{\text{shap},j} = \frac{1}{n}$.

**Definition 2.2.** $(\alpha, \beta)$-*Beta Shapley* (Kwon & Zou, 2022) extends Data Shapley by introducing tunable parameters $(\alpha, \beta) \in \mathbb{R}^2$, which control the emphasis placed on marginal contributions from smaller or larger subsets. The corresponding weight vector $\omega_{\text{beta}} = (\omega_{\text{beta},j})_{j \in [n]}$ is such that for all $j \in [n]$, $\omega_{\text{beta},j} = \binom{n-1}{j-1} \cdot \frac{\texttt{Beta}(j+\beta-1, n-j+\alpha)}{\texttt{Beta}(\alpha,\beta)}$, where $\texttt{Beta}(\alpha, \beta) = \Gamma(\alpha)\Gamma(\beta)/\Gamma(\alpha + \beta)$ and $\Gamma$ is the Gamma function.

**Definition 2.3.** *Data Banzhaf* (Wang & Jia, 2023) is derived from the *Banzhaf value* (Banzhaf, 1965), a cooperative game theory concept originally introduced to measure a player's influence in weighted voting systems. Data Banzhaf weight's vector $\omega_{\text{banzhaf}} = (\omega_{\text{banzhaf},j})_{j \in [n]}$ is such that for all $j \in [n]$, $\omega_{\text{banzhaf},j} = \binom{n-1}{j-1} \cdot \frac{1}{2^{n-1}}$.

These semivalue-based methods satisfy fundamental axioms (Dubey et al., 1981) that ensure desirable properties in data valuation. In particular, any semivalue $\phi(.; \omega, .)$ satisfy the *linearity* axiom which states that for any $\alpha_1, \alpha_2 \in \mathbb{R}$, and any utility $u$, $v$, $\phi(z_i; \omega, \alpha_1 u + \alpha_2 v) = \alpha_1 \phi(z_i; \omega, u) + \alpha_2 \phi(z_i; \omega, v)$.

# 3 A METHODOLOGY TO ASSESS DATA VALUATION ROBUSTNESS TO THE UTILITY CHOICE

We now turn to the two scenarios introduced in Section 1, namely the *utility trade-off* scenario and the *multiple-valid-utility* scenario, and show how both admit a common geometric formalization. In what follows, we let $\mathcal{D} = \{z_i\}_{i \in [n]}$ be the dataset that the practitioner seeks to score and rank by order of importance, and we let $\omega$ be the chosen semivalue weight vector, so that each datum score is given by $\phi(z_i; \omega, u)$ as defined in (1). We start by giving a formal definition of each scenario.

***Utility trade-off* scenario.** In this scenario, the practitioner defines utility as a convex combination of multiple fixed criteria. In the simplest case where one considers only two fixed criteria $u^A$ and $u^B$ (e.g. helpfulness vs. harmlessness when fine-tuning an LLM), the utility is

$$u_\nu = \nu u^A + (1 - \nu)u^B, \qquad \nu \in [0, 1],$$

where the scalar $\nu$ is explicitly chosen by the practitioner (based on operational priorities) to set the desired trade-off between $u^A$ and $u^B$. Note that this choice naturally extends to $K$ fixed criteria $u^1, \ldots, u^K$ by taking $u_\nu = \sum_{k=1}^{K} \nu_k u^k$.

By semivalue linearity, each data point's score under $u_\nu$ is

$$\phi(z_i; \omega, u_\nu) = \nu\phi(z_i; \omega, u^A) + (1 - \nu)\phi(z_i; \omega, u^B) .$$

***Multiple-valid-utility* scenario.** In this scenario, there is no single *correct* utility: practitioners must choose among several equally defensible performance metrics. In the common case of binary classification, one might measure model quality with accuracy, F1-score, or negative log-loss: each is valid but may yield different data valuation results; see Table 1. Almost all of these utilities admit a *linear-fractional* form (Koyejo et al., 2014) in two test-set statistics: the empirical true-positive rate $\lambda(S) = \frac{1}{m} \sum_{j=1}^{m} \mathbf{1}[g_S(x_j) = 1, y_j = 1]$ and the empirical positive-prediction rate $\gamma(S) = \frac{1}{m} \sum_{j=1}^{m} \mathbf{1}[g_S(x_j) = 1]$, where $g_S = \mathcal{A}(S)$ is the classifier trained on $S$. Specifically, they can be written as

$$u(S) = \frac{c_0 + c_1\lambda(S) + c_2\gamma(S)}{d_0 + d_1\lambda(S) + d_2\gamma(S)}, \tag{2}$$

with coefficients $(c_\bullet, d_\bullet)$ determined by the chosen utility (see Table 22). Any linear–fractional utility of the form (2) with $d_0 \neq 0$ admits the first–order expansion at $(\lambda, \gamma) = (0, 0)$:

$$u(S) = \frac{c_0}{d_0} + \frac{c_1 d_0 - c_0 d_1}{d_0^2}\lambda(S) + \frac{c_2 d_0 - c_0 d_2}{d_0^2}\gamma(S) + o\big(\|(\lambda(S), \gamma(S))\|\big).$$

Thus, to first order, $u$ is affine in $(\lambda, \gamma)$ and we validate this surrogate empirically (see Appendix B.1). Thus, by linearity of the semivalue and the fact that constants vanish, for each $z_i$, it is reasonable to consider that

$$\phi(z_i; \omega, u) = \frac{c_1 d_0 - c_0 d_1}{d_0^2}\phi(z_i; \omega, \lambda) + \frac{c_2 d_0 - c_0 d_2}{d_0^2}, \phi(z_i; \omega, \gamma).$$

*Remark.* The *multiple-valid utility* scenario also extends to multiclass classification metrics with $u = \sum_{k=1}^{K} \alpha_k u_k$ for $K > 2$ (see Appendix C.5 for details).

## 3.1 A UNIFIED GEOMETRIC MODELING OF THE TWO SCENARIOS

Both scenarios can be unified by observing that the practitioner's utility lies in a two-dimensional family spanned by the two fixed base utilities $u_1$ and $u_2$. Concretely, we consider

$$u_\alpha(S) = \alpha_1 u_1(S) + \alpha_2 u_2(S), \quad \alpha = (\alpha_1, \alpha_2) \in \mathbb{R}^2,$$

so that varying the utility means moving $\alpha$ in this two-dimensional parameter space. In the *utility trade-off* scenario restricted to two fixed criteria, $(u_1, u_2) = (u^A, u^B)$, and $(\alpha_1, \alpha_2) = (\nu, 1 - \nu)$ ranges over $[0, 1]^2$. In the *multiple-valid-utility* scenario for binary classification, $(u_1, u_2) = (\lambda, \gamma)$, and $(\alpha_1, \alpha_2) \in \mathbb{R}^2$. In either case, the objective is the same: to quantify robustness, i.e., how stable the ranking of semivalue scores $\{\phi(z_i; \omega, u_\alpha)\}$ is as we change $\alpha$.

With this unified view in hand, we have the following proposition, which can be extended to the general case $u_\alpha = \sum_{k=1}^{K} \alpha_k u_k$. A detailed extension is provided in Appendix B.2.

**Proposition 3.1.** *Let $\mathcal{D}$ be any dataset of size $n$ and let $\omega \in \mathbb{R}^n$ be a semivalue weight vector. Then there exists a map $\psi_{\omega,\mathcal{D}} : \mathcal{D} \longrightarrow \mathbb{R}^2$ such that for every utility $u_\alpha = \alpha_1 u_1 + \alpha_2 u_2$, $\phi\big(z; \omega, u_\alpha\big) = \big\langle \psi_{\omega,\mathcal{D}}(z), \alpha \big\rangle$, for any $z \in \mathcal{D}$. We call $\mathcal{S}_{\omega,\mathcal{D}} = \{\psi_{\omega,\mathcal{D}}(z) \mid z \in \mathcal{D}\}$ the* spatial signature *of $\mathcal{D}$ under semivalue $\omega$.*

Consequently, ranking the data points in $\mathcal{D}$ by $u_\alpha$ is equivalent to sorting their projections onto the vector $\alpha$:

$$\phi(z_i; \omega, u_\alpha) > \phi(z_j; \omega, u_\alpha) \quad \Longleftrightarrow \quad \langle \psi_{\omega,\mathcal{D}}(z_i), \alpha \rangle > \langle \psi_{\omega,\mathcal{D}}(z_j), \alpha \rangle.$$

Moreover, since scaling $\alpha$ by any positive constant does not change the sign of $\langle \psi_{\omega,\mathcal{D}}(z_i), \alpha \rangle - \langle \psi_{\omega,\mathcal{D}}(z_j), \alpha \rangle$, any two utilities $u_\alpha$ and $u_{\alpha'}$ whose coefficient vectors point in the same direction induce identical rankings. Thus, each utility in the parametric family can be uniquely identified by its normalized vector $\bar{\alpha} = \frac{\alpha}{\|\alpha\|} \in \mathcal{S}^1$, with $\bar{\alpha}$ ranging over the unit circle $\mathcal{S}^1$. Consequently, ranking stability to the utility choice reduces to analyzing how the projections order of $\{\langle \psi_{\omega,\mathcal{D}}(z), \alpha \rangle \mid z \in \mathcal{D}\} \subset \mathbb{R}^2$ changes as we rotate the unit-vector $\bar{\alpha}$ around $\mathcal{S}^1$. Figure 1 illustrate the geometric mapping at hand.

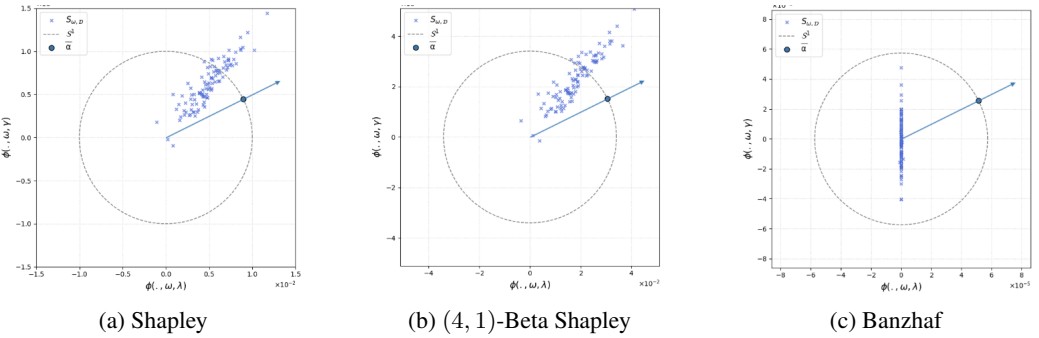

| (a) Shapley | (b) $(4, 1)$-Beta Shapley | (c) Banzhaf |

Figure 1: Spatial signature of the WIND dataset for three semivalues (a) Shapley, (b) $(4, 1)$-Beta Shapley, and (c) Banzhaf. Each cross marks the embedding $\psi_{\omega,\mathcal{D}}(z)$ of a data point (with $u_1 = \lambda$, $u_2 = \gamma$), the dashed circle is the unit circle $\mathcal{S}^1$, and the filled dot indicates one utility direction $\bar{\alpha}$.

Figure 1 shows that, under Banzhaf, the points lie almost exactly on a single line through the origin, much more so than under Shapley or $(4, 1)$-Beta Shapley. This near-collinearity persists across all datasets used in the experiments (see Appendix D.1). In Proposition 3.3 and Section 4.1, we give insight into how this geometric property directly leads to Banzhaf's higher robustness.

## 3.2 A ROBUSTNESS METRIC DERIVED FROM THE GEOMETRIC REPRESENTATION

Building on the geometric mapping of semivalue-based data valuation proposed in Section 3.1, a natural way to quantify how robust a semivalue scores ranking is to changes in the utility is to ask *how far on the unit circle one must rotate from a given utility direction before the induced ordering undergoes a significant change?*

Formally, let $\bar{\alpha}_0$ be the starting utility direction, whose semivalue scores induce a reference ranking of the data points. We say that two points $z_i$ and $z_j$ experience a *pairwise swap* when their order under a new direction $\bar{\alpha}$ is opposite to their order under $\bar{\alpha}_0$. We then aim to define robustness as the smallest geodesic distance on $\mathcal{S}^1$ that one must travel from $\bar{\alpha}_0$ before $p$ pairwise swaps have occurred.

To make this concrete, we express the required geodesic distance in closed form by characterizing the critical angles on $\mathcal{S}^1$ at which pairwise swaps occur. For each unordered pair $(i, j)$, let $v_{ij} = \psi_{\omega,\mathcal{D}}(z_i) - \psi_{\omega,\mathcal{D}}(z_j)$ and observe that the condition $\langle \alpha, v_{ij} \rangle = 0$ defines two antipodal "cut" points on the unit circle: $H_{ij} = \{\alpha \in \mathcal{S}^1 : \langle \alpha, v_{ij} \rangle = 0\}$. Across all $\binom{n}{2} = N$ pairs, these give $2N$ cuts, whose polar angles we list in ascending order as

$$0 \leq \theta_1 \leq \theta_2 \leq \cdots \leq \theta_{2N} < 2\pi,$$

and then wrap around by setting $\theta_{2N+1} = \theta_1 + 2\pi$. The open arcs between successive cuts are $A_k = (\theta_k, \theta_{k+1})$ of length $\lambda_k = \theta_{k+1} - \theta_k, \quad k = 1, \ldots, 2N$ so that $\sum_{k=1}^{2N} \lambda_k = 2\pi$. These open

arcs partition $\mathcal{S}^1$ into ranking regions, meaning that the induced semivalue ordering is identical for every utility direction $\bar{\alpha} \in A_k$. Figure 2 illustrates two example spatial signatures and their induced ranking regions. We view these arcs cyclically by taking indices modulo $2N$. Now let our reference direction $\bar{\alpha}_0$ have polar angle $\varphi_0 \in (\theta_k, \theta_{k+1})$. To induce $p$ swaps, one must cross $p$ distinct arcs: counterclockwise this is $S_k^+(p) = \sum_{i=1}^{p} \lambda_{(k+i) \bmod 2N}$ while clockwise it is $S_k^-(p) = \sum_{i=1}^{p} \lambda_{(k-i) \bmod 2N}$. Writing $t = \varphi_0 - \theta_k \in (0, \lambda_k)$, the minimal geodesic distance from $\bar{\alpha}_0$ to achieve $p$ swaps is[2]

$$\rho_p(\bar{\alpha}_0) = \min\{S_k^+(p) - t, \quad S_k^-(p) + t\}.$$

We now define our robustness metric based on $\rho_p$.

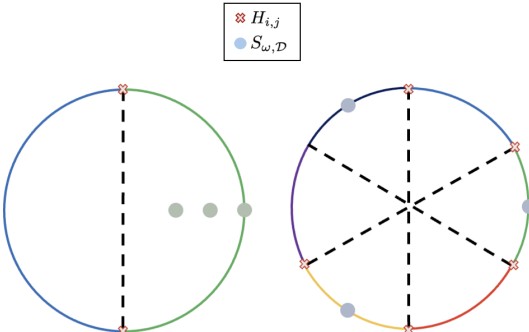

Figure 2: Ranking regions induced by utilities on the unit circle $\mathcal{S}^1$ for two example spatial signatures. Each colored arc on the unit circle corresponds to one of the open arcs $A_k$. Within any single arc, the projection order (and hence the data-point ranking) remains unchanged.

**Definition 3.2** (Robustness metric $R_p$). Let $\mathcal{S}_{\omega,\mathcal{D}} = \{\psi_{\omega,\mathcal{D}}(z_i)\}_{i \in [n]}$ be the spatial signature for dataset $\mathcal{D}$ under semivalue weights $\omega$. For $\bar{\alpha} \in \mathcal{S}^1$, let $\rho_p(\bar{\alpha})$ denote the minimal geodesic distance on $\mathcal{S}^1$ one must travel from $\bar{\alpha}$ to incur $p < \binom{n}{2}$ pairwise swaps in the induced ranking. Define the average $p$–swaps distance $\mathbb{E}_{\bar{\alpha} \sim \mathrm{Unif}(\mathcal{S}^1)}[\rho_p(\bar{\alpha})] = \frac{1}{2\pi} \int_0^{2\pi} \rho_p(t)dt$. Then the *robustness metric* $R_p \in [0, 1]$ is

$$R_p(S_{\omega,\mathcal{D}}) = \frac{\mathbb{E}_{\bar{\alpha} \sim \mathrm{Unif}(\mathcal{S}^1)}[\rho_p(\bar{\alpha})]}{\max_{S_{\omega,\mathcal{D}}} \mathbb{E}_{\bar{\alpha} \sim \mathrm{Unif}(\mathcal{S}^1)}[\rho_p(\bar{\alpha})]} = \frac{\mathbb{E}_{\bar{\alpha} \sim \mathrm{Unif}(\mathcal{S}^1)}[\rho_p(\bar{\alpha})]}{\pi/4},$$

where the denominator $\pi/4$ is the maximum possible value of $\mathbb{E}_{\bar{\alpha}}[\rho_p(\bar{\alpha})]$ which occurs precisely when all embedded points $\psi_{\omega,\mathcal{D}}(z_i)$ are collinear [3].

Concretely, given a spatial signature, the $p$-robustness metric $R_p$ of this signature is the normalized average minimal angular distance one must rotate on the unit circle to force exactly $p$ pairs of points to swap in order in the induced ranking.

**Interpretation.** $R_p$ close to 1 means that one can rotate $\bar{\alpha}$ significantly without flipping more than $p$ pairs, so the ranking is stable. $R_p$ close to 0 means that even a tiny rotation will likely flip $p$ pairs. Moreover, if there are no tied ranks, $R_p$ captures how far in expectation one must move from a utility direction before the Kendall rank correlation degrades by $2p/\binom{n}{2}$ (see Appendix B.4 for details).

**Computation.** We derive a closed-form expression for $\mathbb{E}_{\bar{\alpha} \sim \mathrm{Unif}(\mathcal{S}^1)}[\rho_p(\bar{\alpha})]$ that computes exactly in $\mathcal{O}(n^2 \log n)$ time (see Appendix B.5). In contrast, semivalue approximation methods based on Monte Carlo sampling require $\mathcal{O}(n^2 \log n)$ *model trainings* to estimate the data values (Jia et al., 2019). Therefore, in practice, once the semivalue scores are in hand, the additional cost of computing $R_p$ is negligible compared to the heavy model-training overhead, making this robustness metric an affordable add-on to any data valuation pipeline.

---

[2]All cut-angles, arc-lengths, and resulting geodesic distance $\rho_p$ are entirely determined by the spatial signature $\mathcal{S}_{\omega,\mathcal{D}}$. For brevity, we omit the explicit dependence on it from our notations.

[3]Proof of this claim is given in Appendix B.6.

**Extension to $K > 2$.** The robustness metric $R_p$ extends naturally to $K > 2$ base utilities, where utility directions $\bar{\alpha}$ lie on the unit sphere $\mathcal{S}^{K-1}$. While no closed-form exists for $\mathbb{E}[\rho_p]$ in this case, it can be efficiently approximated via Monte Carlo sampling. Appendix B.5 provides convergence guarantees.

## 3.3 SPATIAL ALIGNMENT AND THE ROBUSTNESS OF SEMIVALUES

The robustness metric $R_p$ (Definition 3.2) measures the stability of the data-value ranking as the utility varies. It increases with the *collinearity* of the spatial signature $S_{\omega,\mathcal{D}} = \{\psi_{\omega,\mathcal{D}}(z) : z \in \mathcal{D}\} \subset \mathbb{R}^2$, which is captured by the Pearson correlation between the two coordinate score vectors for base utilities $u_1$ and $u_2$. In Proposition 3.3, we express this correlation directly in terms of marginal contributions, and we characterize how it depends on semivalue weights under mild assumptions.

Let $\phi(u_a) = (\phi(z_1; \omega, u_a), \ldots, \phi(z_n; \omega, u_a)) \in \mathbb{R}^n$ for $a \in \{1, 2\}$. For $v, w \in \mathbb{R}^n$, write $\bar{v} = \frac{1}{n}\sum_i v_i$, $\mathrm{Var}(v) = \frac{1}{n}\sum_i (v_i - \bar{v})^2$, and $\mathrm{Cov}(v, w) = \frac{1}{n}\sum_i (v_i - \bar{v})(w_i - \bar{w})$. We study

$$\mathrm{Corr}\big(\phi(u_1), \phi(u_2)\big) = \frac{\mathrm{Cov}(\phi(u_1), \phi(u_2))}{\sqrt{\mathrm{Var}(\phi(u_1))\mathrm{Var}(\phi(u_2))}}.$$

**Proposition 3.3** (Utility alignment and semivalue weights). *Let $u_1, u_2$ be two base utilities and $\phi(u_1), \phi(u_2) \in \mathbb{R}^n$ their semivalue score vectors. If for all $j \neq k$ the marginal-contribution vectors $\Delta_j(u_1) := (\Delta_j(z_1, u_1), \ldots, \Delta_j(z_n, u_1))$ and $\Delta_k(u_2) := (\Delta_k(z_1, u_2), \ldots, \Delta_k(z_n, u_2))$ are uncorrelated across points, then*

$$\mathrm{Corr}(\phi(u_1), \phi(u_2)) = \frac{\sum_{j=1}^n \omega_j^2 \mathrm{Cov}\big(\Delta_j(u_1), \Delta_j(u_2)\big)}{\sqrt{\sum_{j=1}^n \omega_j^2 \mathrm{Var}\big(\Delta_j(u_1)\big)}\sqrt{\sum_{j=1}^n \omega_j^2 \mathrm{Var}\big(\Delta_j(u_2)\big)}}.$$

*Defining the size-$j$ alignment factor*

$$r_j := \mathrm{Cov}\big(\Delta_j(u_1), \Delta_j(u_2)\big) = \mathrm{Corr}\big(\Delta_j(u_1), \Delta_j(u_2)\big)\sqrt{\mathrm{Var}(\Delta_j(u_1))\mathrm{Var}(\Delta_j(u_2))},$$

*then the correlation increases as the semivalue weights $\{\omega_j\}$ concentrate on sizes $j$ where $r_j$ is large.*

The proof is given in Appendix B.7.

## 4 EMPIRICAL EVALUATION OF ROBUSTNESS AND DISCUSSION

### 4.1 MULTIPLE-VALID UTILITY SCENARIO

In this section, we empirically validate the $p$-robustness metric $R_p$ in the *multiple-valid-utility* scenario. We evaluate $R_p$ for three semivalues, Shapley, $(4, 1)$-Beta Shapley, and Banzhaf, on several public binary classification datasets. The results in Figure 3 (detailed in Table 7) closely track Section 1's correlation experiments reported in Table 1: datasets and semivalues that exhibit low rank correlations between different utilities also show low $R_p$, and vice versa.

We also observe that across practically every dataset and choice of $p$, using the Banzhaf weights achieves the highest $R_p$. This makes sense geometrically: Figure 1 and the analogous plots for the other datasets in Appendix D.1 show that the Banzhaf weighting scheme tends to *collinearize* the spatial signature, i.e., push the points closer to a common line through the origin. And since the maximum possible average swap-distance occurs when all embedded points are collinear, this near-collinearity explains why Banzhaf yields the greatest robustness to utility shifts. This observation aligns with prior empirical findings (Wang & Jia, 2023; Li & Yu, 2023), which reported that Banzhaf scores tend to vary less than other semivalues under changing conditions.

These geometric insights are made rigorous by Proposition 3.3, applied to the correlation between the semivalue vectors for $\lambda$ and $\gamma$, i.e., $\mathrm{Corr}(\phi(\lambda), \phi(\gamma))$. It says that under a mild assumption on cross–size correlations of marginal contributions (empirically verified on BREAST and TITANIC notably; see Appendix A.5), this correlation decomposes into a weighted average of size–specific alignment factors $r_j$, with weights $\omega_j^2$. Figure 4 plots the normalized $r_j$ versus coalition size $j$ and overlays the Shapley, $(4, 1)$-Beta, and Banzhaf weight profiles. On BREAST, $r_j$ is uniformly high

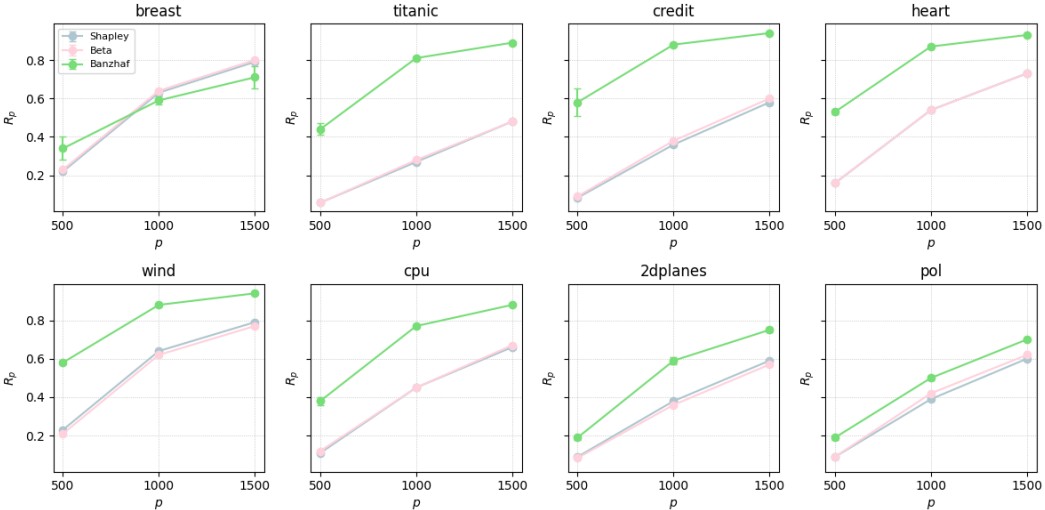

Figure 3: Mean $p$-robustness $R_p$ (error bars = standard errors over 5 Monte Carlo approximations) plotted against $p \in \{500, 1000, 1500\}$ for each dataset and semivalue. Each plot corresponds to one dataset, with Shapley (blue), $(4, 1)$-Beta Shapley (pink), and Banzhaf (green) curves. Higher $R_p$ indicates greater ranking stability under utility shifts.

across $j$, so all three semivalues yield similar collinearity, which is consistent with the overlapping robustness curves in Figure 3. On TITANIC, $r_j$ peaks at intermediate $j$ and decays at the extremes; because Banzhaf concentrates weight in this middle region, it attains a larger weighted average (hence higher overall correlation), explaining why its robustness curve sits well above Shapley and $(4, 1)$-Beta in Figure 3.

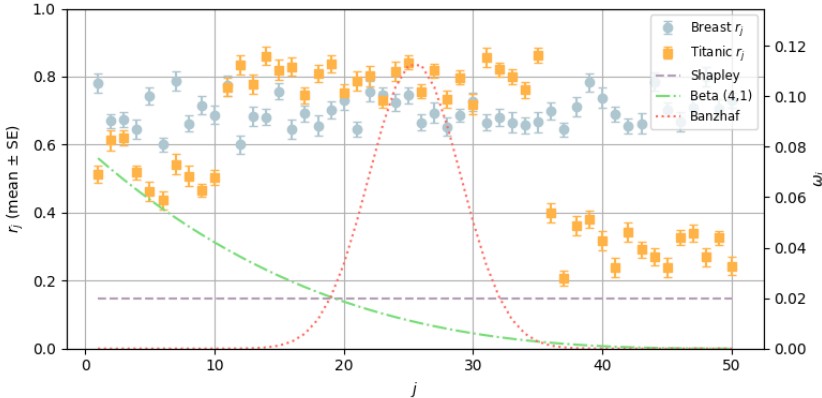

Figure 4: Mean (normalized) $r_j$ (error bars = standard errors over 5 semivalue approximations) for BREAST (blue) and TITANIC (red) vs. coalition size $j$, with semivalue weights $\omega$ overlaid.

Additional robustness experiments, including comparison to top-$k$ stability metrics and extensions to $K > 2$ base utilities, are reported in Appendices A.7 and A.9 and are discussed in Appendix A.10.

## 4.2 UTILITY TRADE-OFF SCENARIO

We also evaluate $R_p$ in the *utility trade-off* scenario, where utility is defined as a convex combination of competing criteria. Specifically, we consider utilities of the form $u_\nu = \nu u_1 + (1 - \nu)u_2$ with $\nu \in [0, 1]$, and analyze how semivalue-based rankings (using Shapley, $(4, 1)$-Beta Shapley, and Banzhaf) evolve as $\nu$ varies. We run this on *regression* datasets (DIABETES, CALIFORNIA HOUSING, AMES) for utility pairs MSE/MAE, MSE/R$^2$, and MAE/R$^2$, and on *multiclass classification* datasets

(DIGITS, WINE, IRIS) for utility pairs Accuracy/macro-F1, Accuracy/macro-Recall, and macro-F1/macro-Recall. Across all settings, Banzhaf achieves the highest $R_p$, indicating more stable rankings. These results are consistent with the ones obtained in *multiple-valid utility* scenario (see Section 4.1). The data sources are given in Appendix A.1 while full results with experimental settings are reported in Tables 9, 10, 11, 12, 13, and 14. Additional experiments for the case $K > 2$ base utilities are detailed in Appendix A.6.2 and are discussed in Appendix A.10.

## 5  CONCLUSION

This work studies the robustness of semivalue-based data valuation methods under utility shifts in two scenarios where it matters, by introducing a unified geometric view via the *spatial signature* and a parametric robustness measure $R_p$. This yields a practical way to quantify the stability of data-value rankings as the utility varies.

**Limitation.** While the framework is general, our analysis of the *multiple-valid-utility* scenario focuses on binary classification metrics in the linear–fractional family and on a subset of multiclass metrics. Non-linear-fractional binary metrics (e.g., negative log–loss) and regression utilities fall outside our scope in this scenario.

**Future works.** By revealing cases in which semivalue-based data valuation fails to produce reliable scores, we aim to encourage future research to assess whether these methods genuinely solve the problem they claim to address.

## REPRODUCIBILITY STATEMENT

The full codebase is publicly available at https://github.com/taminemelissa/utility-impact.git. It reproduces all tables and figures in the paper (with scripts to generate them). Full experimental protocols, including datasets, pre-processing, hyperparameters, and compute settings, are documented in Appendix A and are cross-referenced at the relevant points in the main text. All missing proofs and supporting theoretical results are given in Appendix B, where assumptions are stated, and derivations are provided.

## ACKNOWLEDGMENTS

This work was partially supported by the French National Research Agency (ANR) through grants ANR-20-CE23-0007 and ANR-23-CE23-0002 and through the PEPR IA FOUNDRY project (ANR-23-PEIA-0003).

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

## A ADDITIONAL SETTINGS & EXPERIMENTS

For the reader's convenience, we first outline the main points covered in this section.

- Appendix A.1: Experiment settings for empirical results in the main text.
- Appendix A.2: Additional results on rank correlation for more binary classification metrics.
- Appendix A.3: Additional results on rank correlation using the Spearman rank correlation.
- Appendix A.4: Table for $R_p$ results in Figure 3.
- Appendix A.5: Empirical verification of the assumption of Proposition 3.3.
- Appendix A.6: Results for the *utility-trade-off* scenario summarized in Section 4.2 and extension to $K > 2$ base utilities.
- Appendix A.7: Results for the *multiple-valid utility* scenario extended to $K > 2$ base utilities.
- Appendix A.8: What if we $\mathcal{A}$ varies instead of `perf`?
- Appendix A.9: Empirical link between the robustness metric $R_p$ and top-$k$ stability metrics (overlap@$k$ and Jaccard@$k$).
- Appendix A.10: Overall discussion about empirical robustness results.

### A.1 EXPERIMENT SETTINGS FOR EMPIRICAL RESULTS IN THE MAIN TEXT

In this section, we describe our experimental protocol for estimating semivalue scores, which serve to obtain all the tables and figures included in this paper.

**Datasets.** Table 2 summarizes the datasets used in our experiments, all of which are standard benchmarks in the data valuation literature (Ghorbani & Zou, 2019; Kwon & Zou, 2022; Jia et al., 2019; Wang & Jia, 2023; Jiang et al., 2023). Due to the computational cost of repeated model retraining in our experiments, we select a subset of 100 instances for training and 50 instances for testing from each classification dataset and 300 instances for training and 100 instances for testing from regression datasets.

Table 2: A summary of datasets used in experiments.

| Dataset | Source |
|---|---|
| BREAST | https://www.openml.org/d/13 |
| TITANIC | https://www.openml.org/d/40945 |
| CREDIT | Pozzolo et al. (2015) |
| HEART | https://www.openml.org/d/43398 |
| WIND | https://www.openml.org/d/847 |
| CPU | https://www.openml.org/d/761 |
| 2DPLANES | https://www.openml.org/d/727 |
| POL | https://www.openml.org/d/722 |
| DIABETES | Efron et al. (2004) |
| CALIFORNIA HOUSING | Kelley Pace & Barry (1997) |
| AMES | De Cock (2011) |
| IRIS | Fisher (1936) |
| WINE | https://archive.ics.uci.edu/ml/datasets/Wine |
| DIGITS | Dua & Graff (2019) |

Because our primary objective is to measure how changing the utility alone affects semivalue rankings, we must eliminate any other sources of variation, such as different train/test splits, model initialization, or Monte Carlo sampling noise, that could confound our results. To this end, we enforce two strict controls for semivalue scores computation across utilities:

1. A *fixed learning context* $(\mathcal{A}, \mathcal{D}_{\text{test}})$,
2. *Aligned sampling* for semivalue approximations.

FIXED LEARNING CONTEXT.    As outlined in Section 2, a utility function $u$ is defined as:

$$u(S) = \texttt{perf}(\mathcal{A}(S), \mathcal{D}_{\text{test}}),$$

where $\mathcal{A}$ is a learning algorithm that outputs a model trained on a dataset $S$, and $\texttt{perf}$ evaluates the model on a test set $\mathcal{D}_{\text{test}}$. The learning algorithm $\mathcal{A}$ specifies the model class, objective function, optimization procedure, and hyperparameters (e.g., learning rate, weight initialization).

By fixing $(\mathcal{A}, \mathcal{D}_{\text{test}})$, we ensure that swapping between two utilities, say, accuracy versus F1-score, amounts solely to changing the performance metric $\texttt{perf}$. Consequently, any shift in the semivalue scores' ranking (measured by rank correlation metrics) can only be attributed to the utility choice.

**Controlling for sampling noise in semivalue estimates.**    The above discussion assumes access to exact semivalue scores, but in practice, we approximate them via Monte Carlo permutation sampling, which injects random noise into each run. Without accounting for this sampling variability, differences in semivalue scores' rankings could reflect estimator noise rather than genuine sensitivity to the utility.

To enforce this, we introduce *aligned sampling* alongside the fixed learning context $(\mathcal{A}, \mathcal{D}_{\text{test}})$. Aligned sampling consists of pre-generating a single pool of random permutations (or sampling seeds) and reusing those same permutations when estimating semivalues for each utility. By sharing both the model-training environment and the permutation draws, we ensure that any differences in resulting rankings are driven solely by the change in utility.

**Fixed set of permutations.**    Let $\mathcal{P} = \{\pi_1, \pi_2, \ldots, \pi_m\}$ denote a fixed set of $m$ random permutations of the data points in $\mathcal{D}$. We apply this exact set of permutations across multiple utilities $\{u_1, u_2, \ldots, u_K\}$ such that $u_k(\cdot) = \texttt{perf}_k[(\mathcal{A})(\cdot), \mathcal{D}_{\text{test}}]$ with fixed $(\mathcal{A}, \mathcal{D}_{\text{test}})$ for all $k \in [K]$.

For a given performance metric $\texttt{perf}_k$ and the set of permutations $\mathcal{P}$, we estimate the marginal contributions $\{\hat{\Delta}_j(z_i; u_k)\}_{j=1}^n$ for each data point $z_i \in \mathcal{D}$ with respect to the utility $u_k$ such as

$$\hat{\Delta}_j(z_i; u_k) := \frac{1}{m} \sum_{s=1}^m \left( u_k\left(S_j^{\pi_s} \cup \{z_i\}\right) - u_k\left(S_j^{\pi_s}\right) \right),$$

where $m$ is the number of permutations used, $\pi_s$ denotes the $s$-th permutation and $S_j^{\pi_s}$ represents the subset of data points of size $j-1$ that precedes $z_i$ in the order defined by permutation $\pi_s$.

**Determining the number of permutations $m$.**    The number of permutations $m$ used in the marginal contribution estimator is determined based on a maximum limit and a convergence criterion applied across all utilities $u_1, \ldots, u_K$. Formally,

$$m = \max\left(m_{\min}, \min\left(m_{\max}, m_{\text{conv}}\right)\right),$$

where $m_{\min}$ is a predefined minimum number of permutations to avoid starting convergence checks prematurely, $m_{\max}$ is a predefined maximum number of permutations set to control computational feasibility, $m_{\text{conv}}$ is the smallest number of permutations required for the Gelman-Rubin (GR) (Vats & Knudson, 2021) statistic to converge across all utility functions $u_1, \ldots, u_K$. Using the Gelman-Rubin statistic as a convergence criterion follows established practices in the literature (Jiang et al., 2023; Kwon & Zou, 2022).

For each data point $z_i$, the GR statistic $R_i$ is computed for every 100 permutations across all utilities. The sampling process halts when the maximum GR statistic across all data points and all utilities falls below a threshold, indicating convergence. We adopt the conventional threshold of $1.05$ for GR convergence, consistent with prior studies in data valuation (Jiang et al., 2023).

In this framework, the GR statistic, $R_i^k$, is used to assess the convergence of marginal contribution estimates for each data point $z_i$ across $C$ independent chains of $s$ sampled permutations under each utility $u_k$. The GR statistic evaluates the agreement between chains by comparing the variability within each chain to the variability across the chains, with convergence indicated when $R_i^k$ approaches 1. Specifically, to compute the GR statistic for each data point $z_i$ under utility $u_k$, we determine

  1. The within-chain variance $W_i^k$ which captures the variability of marginal contributions for $z_i$ within each chain. Specifically, if there are $c$ independent chains, $W_i^k$ is calculated as the

average of the sample variances within each chain

$$W_i^k = \frac{1}{C} \sum_{c=1}^{C} s_{i,c}^2,$$

where $s_{i,c}^2$ is the sample variance of marginal contributions for $z_i$ within chain $c$. This term reflects the dispersion of estimates within each chain,

2. And the between-chain variance $B_i^k$, which measures the variability between the mean marginal contributions across the chains. It indicates how much the chains differ from each other. The between-chain variance is defined as

$$B_i^k = \frac{s}{C-1} \sum_{c=1}^{C} \left( \bar{\Delta}_c(z_i; u_k) - \bar{\Delta}(z_i; u_k) \right)^2,$$

where $\bar{\Delta}_c(z_i; u_k)$ is the mean marginal contribution for $z_i$ in chain $c$, and $\bar{\Delta}(z_i; u_k)$ is the overall mean across all chains

$$\bar{\Delta}(z_i; u_k) = \frac{1}{C} \sum_{c=1}^{C} \bar{\Delta}_c(z_i; u_k).$$

The term $B_i^k$ quantifies the extent of disagreement among the chain means.

Combining both $W_i^k$ and $B_i^k$, the GR statistic $R_i^k$ for data point $z_i$ under utility $u_k$ is defined as:

$$R_i^k = \sqrt{\frac{(s-1)}{s} + \frac{B_i^k}{W_i^k \cdot s}}.$$

**Intra-permutation truncation.** Building on existing literature (Ghorbani & Zou, 2019; Jiang et al., 2023), we further improve computational efficiency by implementing an intra-permutation truncation criterion that restricts coalition growth once contributions stabilize. Given a random permutation $\pi_s \in \mathcal{P}$, the marginal contribution for each data point $z_{\pi_{s,j}}$ (the $j$-th point in the permutation $\pi_s$) is calculated incrementally as the coalition size $j$ increases from 1 up to $n$. However, instead of expanding the coalition size through all $n$ elements, the algorithm stops increasing $j$ when the marginal contributions become stable based on a relative change threshold.

For each step $l \in [n]$ within a permutation, the relative change $V_l^k$ in the utility $u_k$ is calculated as:

$$V_l^k := \frac{\left| u_k \left( \{z_{\pi_{s,j}}\}_{j=1}^{l} \cup \{z_{\pi_{s,l+1}}\} \right) - u_k \left( \{z_{\pi_{s,j}}\}_{j=1}^{l} \right) \right|}{u_k \left( \{z_{\pi_{s,j}}\}_{j=1}^{l} \right)}.$$

where $\{z_{\pi_{s,j}}\}_{j=1}^{l}$ represents the coalition formed by the first $l$ data points in $\pi_s$. This measures the relative change in the utility $u_k$ when adding the next data point to the coalition. The truncation criterion stops increasing the coalition size at the smallest value $j$ satisfying the following condition:

$$j^* = \arg\min \left\{ j \in [n] : \left| \{l \le j : V_l \le 10^{-8}\} \right| \ge 10 \right\}.$$

This means that the coalition size $j^*$ is fixed at the smallest $j$ for which there are at least 10 prior values of $V_l$ (for $l \le j$) that are smaller than a threshold of $10^{-8}$. This condition ensures that the utility $u_k$ has stabilized, indicating convergence within the permutation. This intra-permutation truncation reduces computational cost by avoiding unnecessary calculations once marginal contributions stabilize.

**Aggregating marginal contributions for semivalues estimation.** Once the marginal contributions have been estimated consistently across all permutations and utilities, they are aggregated to compute various semivalues, such as the Shapley, Banzhaf, and $(4, 1)$-Beta Shapley values. Each semivalue method applies a specific weighting scheme (see Definition 2.1, 2.2, 2.3) to the marginal contributions to reflect the intended measure of data point importance.

For a data point $z_i$ under utility $u_k$, its approximated data value $\hat{\phi}(z_i; \omega, u_k)$ is computed by applying a weighting scheme $\omega$ to the marginal contributions across coalition sizes

$$\hat{\phi}(z_i; \omega, u_k) = \sum_{j=1}^{n} \omega_j \, \hat{\Delta}_j(z_i; u_k),$$

where $\hat{\Delta}_j(z_i; u_k)$ is the estimated marginal contribution for coalition size $j - 1$, and $\omega_j$ is the weight assigned to coalition size $j - 1$.

**Learning algorithm $\mathcal{A}$.** For binary classification experiments, $\mathcal{A}$ is a logistic-regression classifier (binary cross-entropy loss) trained via L-BFGS with $\ell_2$ regularization ($\lambda = 1.0$). For multiclass classification experiments, $\mathcal{A}$ is a feed-forward MLP (ReLU hidden layers, softmax output) trained with cross-entropy via L-BFGS and $\ell_2$ regularization ($\lambda = 1.0$). For regression experiments, $\mathcal{A}$ is a linear ridge model (squared-error loss, $\ell_2$ regularization $\lambda = 1.0$) trained with L-BFGS. We initialize all weights from $\mathcal{N}(0, 1)$ with a fixed random seed, disable early stopping, and fix the maximum number of training epochs to 100. The optimizer's step size is 1.0.

**Decision-threshold calibration for binary classification.** Because we compare multiple binary classification utilities (accuracy, F1-score, etc.), using a fixed probability cutoff (e.g., 0.5) can unfairly favor some metrics over others, especially under class imbalance. To ensure that differences in semivalue scores' rankings arise from the utility definition (and not an arbitrary threshold), we calibrate the decision boundary to the empirical class prevalence. Concretely, if $p$ is the fraction of positive labels in the training set, we set the cutoff at the $(1 - p)$-quantile of the model's predicted probabilities. This way, each trained model makes exactly $p\%$ positive predictions, aligning base-rate assumptions across utilities and isolating the effect of the performance metric itself.

**Computational resources and runtime.** All experiments ran on a single machine (Apple M1 (8-core CPU) with 16 GB RAM) without parallelization. A full semivalue estimation, consisting of 5 independent Monte Carlo approximations, for one dataset of 100 data points takes approximately 15 minutes.

## A.2 Additional results on rank correlation for more binary classification metrics

In Table 1, we compare semivalue score rankings under accuracy versus F1-score. Here, we broaden this analysis to include other widely used binary classification utilities (recall, negative log-loss, and arithmetic mean). Tables 3 and 4 show that ranking variability persists across datasets and semivalue choices when using these additional metrics.

Table 3: Mean Kendall rank correlations (standard error in parentheses rounded to one significant figure for clarity) between accuracy (`acc`) and negative log-loss (`nll`), and between F1-score (`f1`) and negative log loss, for three semivalues (Shapley, Beta (4,1), Banzhaf). Values are averaged over 5 estimations.

| Dataset | Shapley | | (4,1)-Beta Shapley | | Banzhaf | |
|---|---|---|---|---|---|---|
| | `acc-nll` | `f1-nll` | `acc-nll` | `f1-nll` | `acc-nll` | `f1-nll` |
| BREAST | -0.59 (0.02) | -0.60 (0.02) | -0.65 (0.01) | -0.66 (0.01) | 0.18 (0.01) | 0.18 (0.01) |
| TITANIC | -0.53 (0.01) | 0.54 (0.01) | -0.60 (0.01) | -0.61 (0.01) | 0.14 (0.02) | -0.07 (0.01) |
| CREDIT | -0.59 (0.02) | -0.43 (0.01) | -0.66 (0.01) | -0.49 (0.01) | 0.38 (0.01) | 0.28 (0.03) |
| HEART | -0.04 (0.02) | 0.01 (0.02) | -0.20 (0.02) | -0.17 (0.03) | -0.07 (0.01) | -0.05 (0.01) |
| WIND | 0.67 (0.02) | 0.69 (0.01) | 0.74 (0.02) | 0.73 (0.01) | 0.26 (0.01) | 0.44 (0.01) |
| CPU | 0.55 (0.01) | 0.68 (0.01) | 0.59 (0.01) | 0.69 (0.01) | -0.53 (0.01) | 0.52 (0.01) |
| 2DPLANES | 0.22 (0.02) | 0.98 (0.01) | 0.41 (0.01) | 0.98 (0.01) | -0.03 (0.01) | 0.18 (0.01) |
| POL | 0.58 (0.01) | 0.79 (0.01) | 0.74 (0.01) | 0.81 (0.01) | -0.01 (0.02) | 0.13 (0.02) |

## A.3 Additional results on rank correlation using the Spearman rank correlation

For completeness, we re-evaluate all of our pairwise semivalue ranking comparisons using Spearman rank correlation instead of Kendall rank correlation. As shown in Tables 5 and 6, datasets and semivalues that exhibit low Kendall correlations between different utilities also yield low Spearman correlations, and vice versa.

Table 4: Mean Kendall rank correlations (standard error in parentheses rounded to one significant figure for clarity) between recall (`rec`) and accuracy (`acc`) for three semivalues (Shapley, Beta (4,1), Banzhaf). Values are averaged over 5 estimations.

| Dataset | Shapley | | | (4,1)-Beta Shapley | | | Banzhaf | | |
|---|---|---|---|---|---|---|---|---|---|
| | acc-am | acc-rec | am-rec | acc-am | acc-rec | am-rec | acc-am | acc-rec | am-rec |
| BREAST | 0.93 (0.01) | 0.98 (0.01) | 0.92 (0.01) | 0.94 (0.01) | 0.98 (0.01) | 0.92 (0.01) | 0.82 (0.03) | 0.99 (0.01) | 0.81 (0.03) |
| TITANIC | -0.25 (0.04) | 0.77 (0.02) | -0.05 (0.05) | -0.27 (0.03) | 0.62 (0.04) | 0.08 (0.05) | 0.46 (0.02) | 0.81 (0.01) | 0.65 (0.01) |
| CREDIT | -0.31 (0.01) | 0.07 (0.01) | 0.60 (0.02) | -0.31 (0.02) | 0.12 (0.04) | 0.62 (0.01) | 0.35 (0.01) | 0.58 (0.01) | 0.76 (0.01) |
| HEART | 0.19 (0.02) | 0.98 (0.01) | 0.18 (0.02) | 0.22 (0.01) | 0.98 (0.01) | 0.19 (0.01) | 0.61 (0.01) | 0.98 (0.01) | 0.59 (0.02) |
| WIND | 0.08 (0.03) | 0.98 (0.01) | 0.07 (0.03) | 0.10 (0.02) | 0.98 (0.01) | 0.08 (0.04) | 0.77 (0.01) | 0.98 (0.01) | 0.75 (0.01) |
| CPU | 0.19 (0.04) | 0.75 (0.02) | 0.18 (0.01) | 0.22 (0.03) | 0.78 (0.02) | 0.22 (0.02) | 0.79 (0.01) | 0.93 (0.01) | 0.86 (0.01) |
| 2DPLANES | 0.31 (0.02) | 0.99 (0.01) | 0.31 (0.02) | 0.33 (0.02) | 0.99 (0.01) | 0.33 (0.02) | 0.037 (0.01) | 0.99 (0.01) | 0.37 (0.01) |
| POL | 0.56 (0.01) | 0.73 (0.01) | 0.29 (0.01) | 0.56 (0.01) | 0.79 (0.01) | 0.34 (0.01) | 0.67 (0.01) | 0.69 (0.01) | 0.36 (0.01) |

Table 5: Mean Spearman rank correlations (standard error in parentheses rounded to one significant figure for clarity) between accuracy (`acc`) and negative log-loss (`nll`), and between F1-score (`f1`) and negative log loss, for three semivalues (Shapley, Beta (4,1), Banzhaf). Values are averaged over 5 estimations.

| Dataset | Shapley | | | (4,1)-Beta Shapley | | | Banzhaf | | |
|---|---|---|---|---|---|---|---|---|---|
| | acc-f1 | acc-nll | f1-nll | acc-f1 | acc-nll | f1-nll | acc-f1 | acc-nll | f1-nll |
| BREAST | 0.99 (0.01) | -0.76 (0.02) | -0.78 (0.02) | 0.99 (0.01) | -0.82 (0.01) | -0.83 (0.01) | 0.98 (0.01) | 0.22 (0.01) | 0.23 (0.01) |
| TITANIC | -0.20 (0.01) | -0.71 (0.01) | 0.74 (0.01) | -0.18 (0.01) | -0.79 (0.01) | -0.80 (0.01) | 0.95 (0.01) | 0.18 (0.02) | -0.20 (0.01) |
| CREDIT | -0.50 (0.02) | -0.76 (0.02) | -0.61 (0.02) | -0.52 (0.01) | -0.83 (0.01) | -0.68 (0.02) | 0.90 (0.01) | 0.53 (0.01) | 0.40 (0.03) |
| HEART | 0.71 (0.01) | -0.04 (0.02) | 0.03 (0.03) | 0.67 (0.01) | -0.28 (0.04) | -0.23 (0.04) | 0.96 (0.01) | -0.10 (0.02) | -0.08 (0.02) |
| WIND | 0.85 (0.01) | 0.84 (0.01) | 0.86 (0.01) | 0.85 (0.01) | 0.90 (0.01) | 0.89 (0.01) | 0.97 (0.01) | 0.34 (0.01) | 0.62 (0.01) |
| CPU | 0.47 (0.02) | 0.73 (0.01) | 0.85 (0.01) | 0.45 (0.01) | 0.77 (0.01) | 0.86 (0.01) | 0.87 (0.01) | -0.71 (0.01) | 0.70 (0.01) |
| 2DPLANES | 0.24 (0.01) | 0.33 (0.02) | 0.99 (0.01) | 0.28 (0.02) | 0.58 (0.01) | 0.99 (0.01) | 0.75 (0.01) | -0.04 (0.02) | 0.24 (0.05) |
| POL | 0.70 (0.01) | 0.77 (0.01) | 0.92 (0.01) | 0.69 (0.01) | 0.90 (0.01) | 0.93 (0.01) | 0.53 (0.01) | -0.01 (0.03) | 0.21 (0.02) |

Table 6: Mean Spearman rank correlations (standard error in parentheses rounded to one significant figure for clarity) between recall (`rec`) and accuracy (`acc`) for three semivalues (Shapley, Beta (4,1), Banzhaf). Values are averaged over 5 estimations.

| Dataset | Shapley | | | (4,1)-Beta Shapley | | | Banzhaf | | |
|---|---|---|---|---|---|---|---|---|---|
| | acc-am | acc-rec | am-rec | acc-am | acc-rec | am-rec | acc-am | acc-rec | am-rec |
| BREAST | 0.99 (0.01) | 0.99 (0.01) | 0.99 (0.01) | 0.99 (0.01) | 0.99 (0.01) | 0.99 (0.01) | 0.90 (0.02) | 0.99 (0.01) | 0.89 (0.03) |
| TITANIC | -0.37 (0.05) | 0.91 (0.02) | -0.08 (0.08) | -0.37 (0.04) | 0.89 (0.02) | 0.10 (0.08) | 0.62 (0.03) | 0.93 (0.01) | 0.84 (0.01) |
| CREDIT | -0.45 (0.01) | 0.09 (0.02) | 0.79 (0.01) | -0.40 (0.03) | 0.11 (0.02) | 0.83 (0.01) | 0.5 (0.01) | 0.75 (0.01) | 0.92 (0.01) |
| HEART | 0.29 (0.02) | 0.99 (0.01) | 0.27 (0.02) | 0.28 (0.02) | 0.89 (0.02) | 0.27 (0.01) | 0.80 (0.02) | 0.99 (0.01) | 0.78 (0.02) |
| WIND | 0.12 (0.04) | 0.99 (0.01) | 0.11 (0.04) | 0.12 (0.03) | 0.97 (0.02) | 0.10 (0.01) | 0.92 (0.01) | 0.99 (0.01) | 0.92 (0.01) |
| CPU | 0.27 (0.01) | 0.90 (0.01) | 0.27 (0.01) | 0.27 (0.02) | 0.92 (0.03) | 0.31 (0.03) | 0.93 (0.01) | 0.99 (0.01) | 0.97 (0.01) |
| 2DPLANES | 0.44 (0.03) | 0.99 (0.01) | 0.44 (0.03) | 0.47 (0.03) | 0.99 (0.01) | 0.47 (0.03) | 0.52 (0.01) | 0.99 (0.01) | 0.52 (0.01) |
| POL | 0.75 (0.01) | 0.90 (0.01) | 0.42 (0.01) | 0.74 (0.01) | 0.93 (0.01) | 0.48 (0.02) | 0.85 (0.01) | 0.87 (0.01) | 0.52 (0.01) |

## A.4 TABLE FOR $R_p$ RESULTS IN FIGURE 3

In support of Figure 3 displayed in Section 4, Table 7 below reports the mean and standard error of the $p$-robustness metric $R_p$ for $p \in \{500, 1000, 1500\}$ on each dataset and semivalue.

## A.5 EMPIRICAL VERIFICATION OF THE ASSUMPTION OF PROPOSITION 3.3

In this section, we verify empirically that the assumption of Proposition 3.3 holds for the two datasets we take as examples in Figure 4, namely BREAST and TITANIC. For $(u_1, u_2) = (\lambda, \gamma)$ we compute the cross–size covariance matrix

$$\widehat{\Sigma}^{u_1 u_2}_{jk} := \mathrm{Cov}\big(\Delta_j(u_1), \Delta_k(u_2)\big), \qquad j, k \in \{1, \dots, n\},$$

using the same Monte Carlo runs as for the semivalues. We then check that off–diagonal terms are negligible compared to the diagonal by computing two metrics:

$$\hat{\varepsilon} := \max_j \frac{\sum_{k \neq j} \big|\widehat{\Sigma}^{u_1 u_2}_{jk}\big|}{\widehat{\Sigma}^{u_1 u_2}_{jj}} \quad \text{and} \quad \hat{\delta} := \frac{\big|\mathrm{Corr}(\phi(u_1), \phi(u_2)) - \mathrm{Corr}_{\mathrm{diag}}(\phi(u_1), \phi(u_2))\big|}{\big|\mathrm{Corr}(\phi(u_1), \phi(u_2))\big|},$$

Table 7: Mean $p$-robustness $R_p$ (standard error in parentheses) for $p \in \{500, 1000, 1500\}$ estimated over 5 Monte Carlo trials (each trial corresponding to approximating the semivalue scores). Boldface marks the semivalue with the highest $R_p$ for each dataset and $p$. Higher $R_p$ indicates greater stability of the induced ranking under utility shifts.

| Dataset | $R_{500}$ | | | $R_{1000}$ | | | $R_{1500}$ | | |
|---|---|---|---|---|---|---|---|---|---|
| | Shapley | (4,1)-Beta Shapley | Banzhaf | Shapley | (4,1)-Beta Shapley | Banzhaf | Shapley | (4,1)-Beta Shapley | Banzhaf |
| BREAST | 0.22 (0.004) | 0.23 (0.004) | **0.34 (0.06)** | 0.63 (0.004) | **0.64 (0.003)** | 0.59 (0.02) | 0.79 (0.003) | **0.80 (0.002)** | 0.71 (0.06) |
| TITANIC | 0.058 (0.001) | 0.058 (0.001) | **0.44 (0.03)** | 0.27 (0.004) | 0.28 (0.004) | **0.81 (0.01)** | 0.48 (0.004) | 0.48 (0.004) | **0.89 (0.007)** |
| CREDIT | 0.084 (0.005) | 0.091 (0.005) | **0.82 (0.07)** | 0.36 (0.01) | 0.38 (0.01) | **0.97 (0.01)** | 0.58 (0.01) | 0.60 (0.01) | **0.99 (0.002)** |
| HEART | 0.16 (0.003) | 0.16 (0.003) | **0.53 (0.01)** | 0.54 (0.008) | 0.54 (0.009) | **0.87 (0.007)** | 0.73 (0.006) | 0.73 (0.006) | **0.93 (0.003)** |
| WIND | 0.23 (0.009) | 0.21 (0.01) | **0.58 (0.01)** | 0.64 (0.01) | 0.62 (0.009) | **0.88 (0.005)** | 0.79 (0.005) | 0.77 (0.007) | **0.94 (0.004)** |
| CPU | 0.11 (0.003) | 0.12 (0.003) | **0.38 (0.02)** | 0.45 (0.009) | 0.45 (0.009) | **0.77 (0.009)** | 0.66 (0.009) | 0.67 (0.009) | **0.88 (0.004)** |
| 2DPLANES | 0.090 (0.001) | 0.084 (0.002) | **0.19 (0.012)** | 0.38 (0.004) | 0.36 (0.006) | **0.59 (0.02)** | 0.59 (0.004) | 0.57 (0.006) | **0.75 (0.01)** |
| POL | 0.090 (0.003) | 0.09 (0.003) | **0.19 (0.01)** | 0.39 (0.008) | 0.42 (0.007) | **0.50 (0.01)** | 0.60 (0.006) | 0.62 (0.006) | **0.70 (0.01)** |

where $\mathrm{Corr}_{\mathrm{diag}}$ keeps only the diagonal entries $\widehat{\Sigma}_{jj}^{u_1 u_2}$. On BREAST and TITANIC, we find $\hat{\varepsilon} < 0.12$ meaning that, row-wise, the total magnitude of off–diagonal covariances $\sum_{k \neq j} |\widehat{\Sigma}_{jk}^{u_1 u_2}|$ is at most 12% of the corresponding diagonal term $\widehat{\Sigma}_{jj}^{u_1 u_2}$, i.e., off–diagonal cross–size effects are negligible. Moreover, we find that $\hat{\delta} \leq 7\%$ showing that using only the diagonal of $\widehat{\Sigma}^{u_1 u_2}$ reproduces the full correlation within a few percent, which is exactly what one would expect if $\mathrm{Cov}(\Delta_j(u_1), \Delta_k(u_2)) \approx 0$ for $j \neq k$. Exact means $\pm$ 95% CIs are reported in Table 8.

Table 8: Verification of the cross–size independence assumption (Proposition 3.3): $\hat{\varepsilon} := \max_j \sum_{k \neq j} |\widehat{\Sigma}_{jk}^{u_1 u_2}| / \widehat{\Sigma}_{jj}^{u_1 u_2}$ (smaller is better) and $\hat{\delta} := |\mathrm{Corr}(\phi(u_1), \phi(u_2)) - \mathrm{Corr}_{\mathrm{diag}}(\phi(u_1), \phi(u_2))| / |\mathrm{Corr}(\phi(u_1), \phi(u_2))|$ (smaller is better). Mean $\pm$ 95% CI over $R=5$ seeds.

| Dataset | $\hat{\varepsilon}$ (mean $\pm$ 95% CI) | $\hat{\delta}$ (mean $\pm$ 95% CI) |
|---|---|---|
| BREAST | $0.08 \pm 0.03$ | $0.03 \pm 0.01$ |
| TITANIC | $0.10 \pm 0.02$ | $0.05 \pm 0.02$ |

### A.6 RESULTS FOR THE *utility-trade-off* SCENARIO SUMMARIZED IN SECTION 4.2 AND EXTENSION TO $K > 2$ BASE UTILITIES

In this section, we evaluate robustness in the *utility trade-off* setting for regression, binary classification, and multiclass classification.

#### A.6.1 CASE WHERE $K = 2$ BASE UTILITIES

In this setting, the utility is a convex combination of two task-relevant metrics,

$$u_\nu = \nu u_1 + (1 - \nu)u_2, \qquad \nu \in [0, 1].$$

We consider the following utility pairs:

- *Regression.* MSE/MAE (Table 9), MSE/$R^2$ (Table 10), and MAE/$R^2$ (Table 11).
- *Multiclass classification.* Accuracy/macro-F1 (Table 12), Accuracy/macro-Recall (Table 13), and macro-F1/macro-Recall (Table 14).

For each pair, we compute semivalue-based rankings (Shapley, $(4, 1)$-Beta Shapley, Banzhaf) and evaluate robustness along the convex path using $R_{500}$.

#### A.6.2 CASE WHERE $K > 2$ BASE UTILITIES

To study trade-offs beyond pairs of utilities, we consider $K = 3$ base utilities simultaneously, which allows us to visualize corresponding spatial signatures in 3D (see Figures 12-19 in Appendix D.2). In this setting, the utility is therefore a convex combination of three task-relevant metrics,

$$u_\nu = \nu_1 u_1 + \nu_2 u_2 + \nu_3 u_3, \quad (\nu_1, \nu_2, \nu_3) \in \Delta^2,$$

Table 9: **Regression.** Robustness scores $R_{500}$ along the MSE-MAE convex path. Semivalues are approximated over 5 runs using a linear regression model trained with L-BFGS. Datasets: DIABETES ($n = 442, d = 10$), CALIFORNIA HOUSING ($n = 20,640, d = 8$), AMES HOUSING ($n = 2,930, d = 10$); each subsampled to 300 training points. $R_{500}$ is reported as mean $\pm$ standard error across the 5 semivalue approximations.

| Dataset | Semivalue | $\mathbf{R_{500}}$ (mean $\pm$ SE) |
|---|---|---|
| DIABETES | Shapley | $0.99 \pm 0.01$ |
| | (4,1)-Beta Shapley | $0.99 \pm 0.01$ |
| | Banzhaf | $0.99 \pm 0.01$ |
| CALIFORNIA | Shapley | $0.72 \pm 0.01$ |
| | (4,1)-Beta Shapley | $0.71 \pm 0.01$ |
| | Banzhaf | $0.75 \pm 0.01$ |
| AMES | Shapley | $0.99 \pm 0.01$ |
| | (4,1)-Beta Shapley | $0.99 \pm 0.01$ |
| | Banzhaf | $0.99 \pm 0.01$ |

Table 10: **Regression.** Robustness scores $R_{500}$ along the MSE-$R^2$ convex path, reported as mean $\pm$ standard error across 5 semivalue approximations.

| Dataset | Semivalue | $\mathbf{R_{500}}$ (mean $\pm$ SE) |
|---|---|---|
| DIABETES | Shapley | $0.89 \pm 0.01$ |
| | (4,1)-Beta Shapley | $0.89 \pm 0.02$ |
| | Banzhaf | $0.91 \pm 0.01$ |
| CALIFORNIA | Shapley | $0.70 \pm 0.01$ |
| | (4,1)-Beta Shapley | $0.67 \pm 0.01$ |
| | Banzhaf | $0.81 \pm 0.01$ |
| AMES | Shapley | $0.99 \pm 0.01$ |
| | (4,1)-Beta Shapley | $0.99 \pm 0.01$ |
| | Banzhaf | $0.99 \pm 0.01$ |

Table 11: **Regression.** Robustness scores $R_{500}$ along the MAE-$R^2$ convex path. Mean $\pm$ standard error over the 5 approximations.

| Dataset | Semivalue | $\mathbf{R_{500}}$ (mean $\pm$ se) |
|---|---|---|
| DIABETES | Shapley | $0.90 \pm 0.02$ |
| | (4,1)-Beta Shapley | $0.89 \pm 0.02$ |
| | Banzhaf | $0.94 \pm 0.01$ |
| CALIFORNIA | Shapley | $0.66 \pm 0.01$ |
| | (4,1)-Beta Shapley | $0.65 \pm 0.01$ |
| | Banzhaf | $0.72 \pm 0.02$ |
| AMES | Shapley | $0.98 \pm 0.01$ |
| | (4,1)-Beta Shapley | $0.98 \pm 0.01$ |
| | Banzhaf | $0.98 \pm 0.01$ |

where $\Delta^2$ denotes the standard 2-simplex. Specifically, we consider the following utility triplets:

- *Binary classification.* Accuracy, F1, and Recall (Table 15)

- *Regression.* MSE, MAE, and $R^2$ (Table 16)

- *Multiclass classification.* macro-F1, macro-Recall, and Accuracy (Table 17)

For each task, we compute the 3D spatial signatures $S_{\omega,\mathcal{D}} \in \mathbb{R}^3$ and then approximate $R_{500}$ using the sampling scheme of Remark B.6 with $m = 1000$ Monte Carlo sampling. Examples of 3D spatial signatures are given in Appendix D.2.

Table 12: **Multiclass.** Robustness scores $R_{500}$ along the Accuracy-macro-F1 convex path. Semi-values are approximated over 5 runs using an MLP (SGD, fixed seeds). Datasets: DIGITS ($n = 1,797, d = 64, 10$ classes), WINE ($n = 178, d = 13, 3$ classes), IRIS ($n = 150, d = 4, 3$ classes); each subsampled to 100 training points. Mean $\pm$ standard error over the 5 approximations.

| Dataset | Semivalue | $\mathbf{R_{500}}$ (mean $\pm$ se) |
|---------|-----------|------------------------|
| DIGITS | Shapley | $0.78 \pm 0.03$ |
| | (4,1)-Beta Shapley | $0.75 \pm 0.04$ |
| | Banzhaf | $0.82 \pm 0.04$ |
| WINE | Shapley | $0.64 \pm 0.05$ |
| | (4,1)-Beta Shapley | $0.61 \pm 0.05$ |
| | Banzhaf | $0.68 \pm 0.04$ |
| IRIS | Shapley | $0.56 \pm 0.06$ |
| | (4,1)-Beta Shapley | $0.53 \pm 0.05$ |
| | Banzhaf | $0.60 \pm 0.06$ |

Table 13: **Multiclass.** Mean robustness $R_{500}$ along the Accuracy-macro Recall convex path (mean $\pm$ SE over 5 semivalue approximations).

| Dataset | Semivalue | $\mathbf{R_{500}}$ (mean $\pm$ SE) |
|---------|-----------|------------------------|
| DIGITS | Shapley | $0.70 \pm 0.01$ |
| | (4,1)-Beta Shapley | $0.68 \pm 0.01$ |
| | Banzhaf | $0.76 \pm 0.03$ |
| WINE | Shapley | $0.60 \pm 0.01$ |
| | (4,1)-Beta Shapley | $0.57 \pm 0.01$ |
| | Banzhaf | $0.63 \pm 0.04$ |
| IRIS | Shapley | $0.52 \pm 0.02$ |
| | (4,1)-Beta Shapley | $0.50 \pm 0.03$ |
| | Banzhaf | $0.56 \pm 0.03$ |

### A.7 RESULTS FOR THE MULTIPLE-VALID UTILITY SCENARIO EXTENDED TO $K > 2$ BASE UTILITIES

We extend the multiple valid scenario to $K > 2$ base utilities in the multiclass classification setting by using the analytical decomposition derived in Appendix C.5. In fact, each multiclass utility can be written as $u_\alpha = \sum_{c=1}^{C} a_c u_c$, where $u_c$ is a class-wise utility for class $c$. Particularly, in our experiment, we consider the per-class precision for $u_c$ that is defined in Appendix C.5. Hence $K = C$, the number of classes. We evaluate this setting on the three multiclass datasets used for the *utility trade-off scenario* experiments (namely DIGITS, WINE, and IRIS) and approximate $R_{500}$ by sampling directions $\alpha \in \mathcal{S}^{C-1}$ and applying the approximation scheme of Remark B.6 with $m = 1000$ Monte Carlo sampling. The results are given in Table 18.

### A.8 WHAT IF WE $\mathcal{A}$ VARIES INSTEAD OF PERF?

Since $u = \texttt{perf} \circ \mathcal{A}$, one can alter the utility either by changing the algorithm $\mathcal{A}$ or by changing the performance metric $\texttt{perf}$. Our main study held $\mathcal{A}$ fixed and varied $\texttt{perf}$. To illustrate the effect of $\mathcal{A}$, we run an additional experiment with a fixed metric (Accuracy) and two learning algorithms: (i) logistic regression trained with L-BFGS and (ii) a multilayer perceptron (MLP) trained with SGD (introducing randomness via initialization and optimization). Table 19 reports the mean Spearman rank correlation (with standard error) between semivalue-based rankings obtained across multiple runs with the two algorithms, for each semivalue and dataset.

These results show that rankings can vary with the learning algorithm, though not as strongly as when changing the performance metric (cf. Table 1 in the main paper). We also observe smaller standard errors for Banzhaf than for Shapley or (4,1)-Beta, suggesting Banzhaf rankings are less sensitive to

Table 14: **Multiclass.** Mean robustness $R_{500}$ along the macro F1-macro Recall convex path (mean $\pm$ SE over 5 semivalue approximations).

| Dataset | Semivalue | $R_{500}$ (mean $\pm$ SE) |
|---|---|---|
| DIGITS | Shapley | $0.71 \pm 0.01$ |
| | (4,1)-Beta Shapley | $0.71 \pm 0.01$ |
| | Banzhaf | $0.75 \pm 0.03$ |
| WINE | Shapley | $0.67 \pm 0.02$ |
| | (4,1)-Beta Shapley | $0.68 \pm 0.01$ |
| | Banzhaf | $0.77 \pm 0.04$ |
| IRIS | Shapley | $0.80 \pm 0.02$ |
| | (4,1)-Beta Shapley | $0.78 \pm 0.02$ |
| | Banzhaf | $0.83 \pm 0.05$ |

Table 15: **Binary.** Mean robustness $R_{500}$ along the Accuracy-F1-Recall convex path (mean $\pm$ SE over 5 semivalue approximations) for binary classification datasets used in Figure 3.

| Dataset | Semivalue | $R_{500}$ (mean $\pm$ SE) |
|---|---|---|
| BREAST | Shapley | $0.30 \pm 0.01$ |
| | (4,1)-Beta Shapley | $0.33 \pm 0.01$ |
| | Banzhaf | $0.34 \pm 0.06$ |
| TITANIC | Shapley | $0.11 \pm 0.01$ |
| | (4,1)-Beta Shapley | $0.11 \pm 0.01$ |
| | Banzhaf | $0.13 \pm 0.03$ |
| CREDIT | Shapley | $0.15 \pm 0.01$ |
| | (4,1)-Beta Shapley | $0.16 \pm 0.02$ |
| | Banzhaf | $0.11 \pm 0.01$ |
| HEART | Shapley | $0.16 \pm 0.01$ |
| | (4,1)-Beta Shapley | $0.16 \pm 0.01$ |
| | Banzhaf | $0.83 \pm 0.04$ |
| WIND | Shapley | $0.33 \pm 0.01$ |
| | (4,1)-Beta Shapley | $0.31 \pm 0.02$ |
| | Banzhaf | $0.58 \pm 0.01$ |
| CPU | Shapley | $0.21 \pm 0.01$ |
| | (4,1)-Beta Shapley | $0.22 \pm 0.01$ |
| | Banzhaf | $0.29 \pm 0.03$ |
| 2DPLANES | Shapley | $0.16 \pm 0.01$ |
| | (4,1)-Beta Shapley | $0.18 \pm 0.01$ |
| | Banzhaf | $0.22 \pm 0.03$ |
| POL | Shapley | $0.21 \pm 0.01$ |
| | (4,1)-Beta Shapley | $0.24 \pm 0.02$ |
| | Banzhaf | $0.39 \pm 0.04$ |

the randomness in the MLP, which aligns with prior analytical and empirical findings (Wang & Jia, 2023; Li & Yu, 2023).

### A.9 EMPIRICAL LINK BETWEEN THE ROBUSTNESS METRIC $R_p$ AND TOP-$k$ STABILITY METRICS (OVERLAP@$k$ AND JACCARD@$k$)

Appendix B.8 establishes an analytical link between our robustness metric $R_p$ and top-$k$ stability metrics (overlap@$k$ and Jaccard@$k$). We complement this analysis with an empirical study that directly relates $R_p$ to these metrics, in the same spirit as our comparison between $R_p$ and rank correlation measures (Kendall and Spearman) in Section 4.1.

We consider the family of binary classification utilities spanned by base utilities $(\lambda, \gamma)$ used

Table 16: **Regression.** Mean robustness scores $R_{500}$ ($\pm$ standard errors) along the MSE-MAE-$R^2$ convex path. Semivalues are approximated over 5 runs using a linear regression model trained with L-BFGS. Datasets: DIABETES ($n = 442, d = 10$), CALIFORNIA HOUSING ($n = 20,640, d = 8$), AMES HOUSING ($n = 2,930, d = 10$); each subsampled to 300 training points. $R_{500}$ is reported as mean $\pm$ standard error across the 5 semivalue approximations.

| Dataset | Semivalue | $R_{500}$ (mean $\pm$ SE) |
|---|---|---|
| DIABETES | Shapley | $0.83 \pm 0.01$ |
| | (4,1)-Beta Shapley | $0.82 \pm 0.02$ |
| | Banzhaf | $0.85 \pm 0.01$ |
| CALIFORNIA | Shapley | $0.69 \pm 0.01$ |
| | (4,1)-Beta Shapley | $0.68 \pm 0.01$ |
| | Banzhaf | $0.86 \pm 0.03$ |
| AMES | Shapley | $0.89 \pm 0.02$ |
| | (4,1)-Beta Shapley | $0.89 \pm 0.02$ |
| | Banzhaf | $0.91 \pm 0.03$ |

Table 17: **Multiclass.** Mean robustness scores $R_{500}$ ($\pm$ standard errors) along the Accuracy-macro-F1-macro-Recall convex path. Semivalues are approximated over 5 runs using an MLP (SGD, fixed seeds). Datasets: DIGITS ($n = 1,797, d = 64, 10$ classes), WINE ($n = 178, d = 13, 3$ classes), IRIS ($n = 150, d = 4, 3$ classes); each subsampled to 100 training points. Mean $\pm$ standard error over the 5 approximations.

| Dataset | Semivalue | $R_{500}$ (mean $\pm$ SE) |
|---|---|---|
| DIGITS | Shapley | $0.63 \pm 0.03$ |
| | (4,1)-Beta Shapley | $0.61 \pm 0.03$ |
| | Banzhaf | $0.78 \pm 0.04$ |
| WINE | Shapley | $0.44 \pm 0.01$ |
| | (4,1)-Beta Shapley | $0.42 \pm 0.02$ |
| | Banzhaf | $0.69 \pm 0.03$ |
| IRIS | Shapley | $0.63 \pm 0.01$ |
| | (4,1)-Beta Shapley | $0.60 \pm 0.01$ |
| | Banzhaf | $0.66 \pm 0.02$ |

to compute $R_p$ in Section 4.1. For one pair of distinct metrics in this family, precisely Accuracy vs. F1 score, we compute the rankings induced by the corresponding semivalue and, for different values of $k$, we evaluate the associated top-$k$ overlap@$k$ and Jaccard@$k$ between the two rankings. This yields, for each dataset and semivalue, a collection of top-$k$ stability scores that summarize how sensitive top-$k$ selections are to switching between these utilities. We then compare these top-$k$ stability scores given in Table 20 with the robustness scores $R_p$ plotted in Figure 3 and summarized in Table 7.

Across most settings, we observe the same qualitative pattern as with Kendall and Spearman correlation metrics: semivalues that achieve larger $R_p$ for a given utility family (typically Banzhaf) also exhibit higher overlap@$k$ and Jaccard@$k$ when comparing the induced rankings under different utilities in that family. In other words, methods that are more robust according to $R_p$ are also those whose top-$k$ selections change the least when moving between these equally valid utility choices.

## A.10 OVERALL DISCUSSION ABOUT EMPIRICAL ROBUSTNESS RESULTS

The empirical study presented across the paper provides a comprehensive validation of the robustness metric $R_p$ introduced in Section 3.2. The empirical results consistently suggest that $R_p$ captures stability phenomena observed when varying the utility across the two scenarios studied: the *multiple-valid-utility* scenario (Sections 4.1, A.9, A.7) and the *utility-trade-off* scenario (Sections 4.2, A.6). We summarize the main conclusions below.

Table 18: Mean robustness scores $R_{500}$ ($\pm$ standard errors) for the *multiple-valid-utility* scenario with $K > 2$ base utilities in the multiclass classification setting, using the class-wise precision decomposition described in Appendix C.5. Semivalues are approximated over 5 runs using an MLP (SGD, fixed seeds). Datasets: DIGITS ($n = 1{,}797, d = 64$, 10 classes), WINE ($n = 178, d = 13$, 3 classes), IRIS ($n = 150, d = 4$, 3 classes); each subsampled to 100 training points. Mean $\pm$ standard error over the 5 approximations.

| Dataset | Semivalue | $\mathbf{R_{500}}$ (mean $\pm$ SE) |
|---|---|---|
| DIGITS | Shapley | $0.17 \pm 0.03$ |
| | (4,1)-Beta Shapley | $0.19 \pm 0.02$ |
| | Banzhaf | $0.57 \pm 0.04$ |
| WINE | Shapley | $0.21 \pm 0.01$ |
| | (4,1)-Beta Shapley | $0.20 \pm 0.01$ |
| | Banzhaf | $0.68 \pm 0.02$ |
| IRIS | Shapley | $0.65 \pm 0.02$ |
| | (4,1)-Beta Shapley | $0.64 \pm 0.01$ |
| | Banzhaf | $0.70 \pm 0.03$ |

Table 19: Spearman rank correlation (mean $\pm$ standard error) between semivalue rankings computed with a logistic regression model and an MLP, using accuracy as the metric. Results are averaged over 5 runs (varying both the MLP initialization/optimization and the semivalue approximation).

| Dataset | Semivalue | Spearman (mean $\pm$ se) |
|---|---|---|
| BREAST | Shapley | $0.62 \pm 0.21$ |
| | (4,1)-Beta Shapley | $0.67 \pm 0.18$ |
| | Banzhaf | $0.67 \pm 0.05$ |
| TITANIC | Shapley | $0.71 \pm 0.13$ |
| | (4,1)-Beta Shapley | $0.71 \pm 0.07$ |
| | Banzhaf | $0.80 \pm 0.03$ |
| HEART | Shapley | $0.65 \pm 0.21$ |
| | (4,1)-Beta Shapley | $0.62 \pm 0.22$ |
| | Banzhaf | $0.93 \pm 0.07$ |
| WIND | Shapley | $0.82 \pm 0.11$ |
| | (4,1)-Beta Shapley | $0.87 \pm 0.10$ |
| | Banzhaf | $0.85 \pm 0.03$ |

**(1) Agreement of $R_p$ with rank-based stability metrics.** The robustness metric $R_p$ demonstrates consistency with traditional rank correlation measures (Kendall and Spearman) across all experiments. In the *multiple-valid-utility* scenario for binary classification (Section 4.1, Tables 1 and 7), datasets exhibiting low rank correlations between accuracy and F1-score consistently show correspondingly low $R_p$ values. This alignment extends to top-$k$ stability metrics (Table 20), where higher $R_p$ values correlate with greater overlap@$k$ and Jaccard@$k$ scores when switching between equally valid utilities. This suggests that $R_p$ captures meaningful ranking stability that aligns with practitioner intuition while offering a geometric interpretation.

**(2) Banzhaf's consistent robustness advantage.** In all binary classification experiments (Tables 7 and 15), multiclass experiments (Tables 12-14 and 17) and regression experiments (Tables 9-11 and 16), the same trend emerges: the Banzhaf value attains very often the highest robustness. The 2D spatial signatures in Appendix D.1 provide a geometric insight for this phenomenon: Banzhaf weights concentrate on intermediate coalition sizes, where the empirical alignment factors $r_j$ are largest (Figure 4), yielding embeddings that are nearly collinear in $\mathbb{R}^2$ and thus achieve maximal $R_p$. In contrast, Shapley and $(4, 1)$-Beta Shapley, whose weights are either uniform or emphasize small coalition sizes, produce less aligned 2D signatures and therefore lower $R_p$.

**(3) Extension to $K > 2$ base utilities.** The robustness metric naturally extends to higher-dimensional utility families. Our *utility-trade-off* experiments for binary classification with three base

Table 20: Top-$k$ stability metrics (Accuracy vs. F1). Mean $\pm$ standard errors for $k \in \{10, 20, 50\}$. Higher values indicate greater robustness of top-$k$ selections under utility shifts.

| Dataset | Semivalue | Overlap@$k$ | | | Jaccard@$k$ | | |
|---|---|---|---|---|---|---|---|
| | | $k = 10$ | $k = 20$ | $k = 50$ | $k = 10$ | $k = 20$ | $k = 50$ |
| BREAST | Shapley | $0.75 \pm 0.04$ | $0.62 \pm 0.03$ | $0.45 \pm 0.03$ | $0.61 \pm 0.04$ | $0.47 \pm 0.03$ | $0.32 \pm 0.03$ |
| | Beta | $0.76 \pm 0.04$ | $0.63 \pm 0.03$ | $0.46 \pm 0.03$ | $0.62 \pm 0.04$ | $0.48 \pm 0.03$ | $0.33 \pm 0.03$ |
| | Banzhaf | $0.82 \pm 0.04$ | $0.70 \pm 0.04$ | $0.52 \pm 0.03$ | $0.71 \pm 0.04$ | $0.57 \pm 0.04$ | $0.40 \pm 0.03$ |
| TITANIC | Shapley | $0.25 \pm 0.03$ | $0.18 \pm 0.02$ | $0.12 \pm 0.02$ | $0.15 \pm 0.03$ | $0.10 \pm 0.02$ | $0.07 \pm 0.02$ |
| | Beta | $0.28 \pm 0.03$ | $0.20 \pm 0.02$ | $0.14 \pm 0.02$ | $0.17 \pm 0.03$ | $0.12 \pm 0.02$ | $0.08 \pm 0.02$ |
| | Banzhaf | $0.88 \pm 0.03$ | $0.76 \pm 0.04$ | $0.59 \pm 0.03$ | $0.79 \pm 0.03$ | $0.64 \pm 0.04$ | $0.48 \pm 0.03$ |
| CREDIT | Shapley | $0.18 \pm 0.03$ | $0.13 \pm 0.03$ | $0.09 \pm 0.02$ | $0.10 \pm 0.03$ | $0.07 \pm 0.02$ | $0.05 \pm 0.02$ |
| | Beta | $0.22 \pm 0.03$ | $0.16 \pm 0.03$ | $0.11 \pm 0.02$ | $0.13 \pm 0.03$ | $0.09 \pm 0.02$ | $0.06 \pm 0.02$ |
| | Banzhaf | $0.94 \pm 0.02$ | $0.89 \pm 0.03$ | $0.76 \pm 0.03$ | $0.89 \pm 0.02$ | $0.80 \pm 0.03$ | $0.64 \pm 0.03$ |
| HEART | Shapley | $0.65 \pm 0.03$ | $0.56 \pm 0.03$ | $0.41 \pm 0.02$ | $0.50 \pm 0.03$ | $0.41 \pm 0.02$ | $0.29 \pm 0.02$ |
| | Beta | $0.68 \pm 0.03$ | $0.59 \pm 0.03$ | $0.43 \pm 0.02$ | $0.53 \pm 0.03$ | $0.44 \pm 0.02$ | $0.31 \pm 0.02$ |
| | Banzhaf | $0.91 \pm 0.03$ | $0.83 \pm 0.03$ | $0.65 \pm 0.03$ | $0.84 \pm 0.03$ | $0.72 \pm 0.03$ | $0.55 \pm 0.03$ |
| WIND | Shapley | $0.78 \pm 0.03$ | $0.68 \pm 0.03$ | $0.49 \pm 0.03$ | $0.65 \pm 0.03$ | $0.54 \pm 0.03$ | $0.37 \pm 0.03$ |
| | Beta | $0.79 \pm 0.03$ | $0.69 \pm 0.03$ | $0.50 \pm 0.03$ | $0.66 \pm 0.03$ | $0.55 \pm 0.03$ | $0.38 \pm 0.03$ |
| | Banzhaf | $0.92 \pm 0.03$ | $0.85 \pm 0.03$ | $0.70 \pm 0.03$ | $0.86 \pm 0.03$ | $0.75 \pm 0.03$ | $0.60 \pm 0.03$ |
| CPU | Shapley | $0.58 \pm 0.03$ | $0.46 \pm 0.03$ | $0.33 \pm 0.03$ | $0.42 \pm 0.03$ | $0.32 \pm 0.03$ | $0.22 \pm 0.02$ |
| | Beta | $0.61 \pm 0.03$ | $0.49 \pm 0.03$ | $0.35 \pm 0.03$ | $0.45 \pm 0.03$ | $0.34 \pm 0.03$ | $0.23 \pm 0.02$ |
| | Banzhaf | $0.85 \pm 0.03$ | $0.78 \pm 0.04$ | $0.61 \pm 0.03$ | $0.76 \pm 0.03$ | $0.65 \pm 0.04$ | $0.49 \pm 0.03$ |
| 2DPLANES | Shapley | $0.52 \pm 0.03$ | $0.41 \pm 0.03$ | $0.29 \pm 0.03$ | $0.36 \pm 0.03$ | $0.27 \pm 0.03$ | $0.18 \pm 0.02$ |
| | Beta | $0.57 \pm 0.03$ | $0.46 \pm 0.03$ | $0.32 \pm 0.03$ | $0.41 \pm 0.03$ | $0.31 \pm 0.03$ | $0.20 \pm 0.02$ |
| | Banzhaf | $0.74 \pm 0.04$ | $0.64 \pm 0.04$ | $0.47 \pm 0.04$ | $0.60 \pm 0.04$ | $0.49 \pm 0.04$ | $0.35 \pm 0.03$ |
| POL | Shapley | $0.72 \pm 0.03$ | $0.61 \pm 0.03$ | $0.43 \pm 0.02$ | $0.57 \pm 0.03$ | $0.46 \pm 0.03$ | $0.31 \pm 0.02$ |
| | Beta | $0.78 \pm 0.03$ | $0.68 \pm 0.03$ | $0.49 \pm 0.02$ | $0.65 \pm 0.03$ | $0.53 \pm 0.03$ | $0.35 \pm 0.02$ |
| | Banzhaf | $0.48 \pm 0.04$ | $0.41 \pm 0.04$ | $0.29 \pm 0.03$ | $0.33 \pm 0.04$ | $0.27 \pm 0.04$ | $0.18 \pm 0.03$ |

utilities (Table 15) confirm that the geometric insights from $\mathbb{R}^2$ hold in $\mathbb{R}^3$: semivalues with spatial signatures more concentrated along a dominant axis (Figures 12-19 achieve higher robustness, even when the utility varies over the simplex $\Delta^2$ rather than a line segment. These findings align with the geometric interpretation of Proposition 3.1 (and its $K$-dimensional extension) and the ranking-region counts in Corollary B.4: when the spatial signature is *collinear* (has a dominant direction), the number of ranking regions is minimized, leading to the maximum average angular distance to a swap and, consequently, the highest robustness.

# B    ADDITIONAL PROOFS & DERIVATIONS

For the reader's convenience, we first outline the main points covered in this section.

– Appendix B.4: Link between the robustness metric $R_p$ and the Kendall rank correlation.

## B.1    FIRST-ORDER APPROXIMATION OF THE UTILITY IN THE *multiple-valid-utility* SCENARIO FOR BINARY CLASSIFICATION AND EMPIRICAL VALIDATION

This section justifies the approximation used in Section 3, where a linear-fractional utility function $u$ is replaced by its affine surrogate.

Formally, we state in Section 3 that any linear-fractional utility $u$ of the form equation 2 with $d_0 \neq 0$, admits a first-order (Taylor–Young) expansion around $(\lambda, \gamma) = (0, 0)$ of the form

$$u(S) = \frac{c_0}{d_0} + \frac{c_1 d_0 - c_0 d_1}{d_0^2}\lambda(S) + \frac{c_2\,d_0 - c_0\,d_2}{d_0^2}\gamma(S) + o\big(\max\{|\lambda(S)|, |\gamma(S)|\}\big).$$

where $\{c_0, c_1, c_2, d_0, d_1, d_2\}$ are real coefficients which specify the particular utility.

The proof is a direct Taylor expansion of $u$ at $(\lambda, \gamma) = (0, 0)$, followed by an empirical validation of the affine surrogate by inspecting discordance rates.

*Proof.* Define $N(\lambda, \gamma) = c_0 + c_1\,\lambda + c_2\,\gamma$ and $D(\lambda, \gamma) = d_0 + d_1\,\lambda + d_2\,\gamma$ so that $u(S) = f\big(\lambda(S), \gamma(S)\big)$ with

$$f(\lambda, \gamma) = \frac{N(\lambda, \gamma)}{D(\lambda, \gamma)}.$$

Assuming $d_0 \neq 0$ (i.e., the denominator does not vanish at $(0,0)$), the first-order Taylor expansion of $f$ around $(0,0)$ is

$$f(\lambda, \gamma) = f(0,0) + \frac{\partial f}{\partial \lambda}\bigg|_{(0,0)}\lambda + \frac{\partial f}{\partial \gamma}\bigg|_{(0,0)}\gamma + o(\|(\lambda, \gamma)\|).$$

Concretely,

$$f(0,0) = \frac{c_0}{d_0}, \quad \frac{\partial f}{\partial \lambda}\bigg|_{(0,0)} = \frac{c_1 d_0 - c_0 d_1}{d_0^2}, \quad \frac{\partial f}{\partial \gamma}\bigg|_{(0,0)} = \frac{c_2 d_0 - c_0 d_2}{d_0^2}.$$

Moreover, since all norms are equivalent in $\mathbb{R}^2$, the Euclidean norm $\|(\lambda, \gamma)\|$ is equivalent to the infinity norm $\max\{\lambda, \gamma\}$. This concludes the proof. □

To verify that the affine surrogate faithfully preserves the true utility's induced ordering, we compare rankings under $u$ and under its first-order proxy $\hat{u} = \frac{c_0}{d_0} + \frac{c_1 d_0 - c_0 d_1}{d_0^2}\lambda + \frac{c_2 d_0 - c_0 d_2}{d_0^2}\gamma$. For each of the eight public binary-classification datasets introduced in Section A.1, and for each of the three semivalues (Shapley, $(4, 1)$-Beta Shapley, and Banzhaf), we proceed as follows:

1. *Exact ranking*. Compute semivalue scores by using the exact linear-fractional utility $u$, then sort the resulting scores to obtain a reference ranking $r$ of the $n$ data points.

2. *Affine surrogate ranking*. Replace the utility $u$ with its first-order affine approximation around $(\lambda, \gamma) = (0, 0)$ denoted as $\hat{u}$, compute semivalue scores, and sort to obtain an approximate ranking $\hat{r}$.

3. *Discordance measurement*. For each pair of rankings $(r, \hat{r})$, count the number $d$ of discordant pairs (i.e., pairs of points ordered differently between the two rankings), and record the proportion $d/N$, where $N = \binom{n}{2}$.

4. *Repetition and averaging* Repeat steps 1–3 several times, each time using an independent Monte Carlo approximation of the semivalue scores, to capture sampling variability.

Table 21 reports, for each dataset and semivalue, the average proportion of discordant pairs ($\pm$ standard error) between rankings obtained with the exact linear-fractional utility and its first-order affine proxy, for both F1-score and Jaccard coefficient (see Table 22 for their definitions). In all experiments, the sum of the mean discordance rate and its standard error never exceeds 2.3%.

These discordance rates, at most a few percent of all $\binom{n}{2}$ pairs, confirm that, in practice, the omitted higher-order terms of the utility have only a minor effect on the induced semivalue ranking. Consequently, using the affine surrogate instead of the exact linear-fractional form is reasonably justified whenever one's primary interest lies in the *ordering* of data values rather than their precise numerical magnitudes.

Table 21: Proportion of discordant pairs ($\pm$ standard error) between rankings induced by the exact linear-fractional utility $u$ and its first-order affine surrogate $\hat{u}$, for F1-score and Jaccard utilities. Values are computed over $N = \binom{50}{2} = 1225$ pairs and averaged over 5 Monte Carlo trials.

| Dataset | F1-score | | | Jaccard | | |
|---|---|---|---|---|---|---|
| | Shapley | (4,1)-Beta Shapley | Banzhaf | Shapley | (4,1)-Beta Shapley | Banzhaf |
| BREAST | 0.8% (0.1%) | 0.8% (0.2%) | 0.9% (0.1%) | 0.7% (0.1%) | 0.9% (0.1%) | 0.9% (0.2%) |
| TITANIC | 1.3% (0.3%) | 1.3% (0.3%) | 0.8% (0.3%) | 1.6% (0.4%) | 1.3% (0.3%) | 0.7% (0.1%) |
| CREDIT | 1.5% (0.5%) | 1.6% (0.2%) | 1.0% (0.1%) | 1.5% (0.3%) | 1.7%(0.1%) | 0.7% (0.3%) |
| HEART | 1.0% (0.1%) | 0.8% (0.1%) | 0.8% (0.1%) | 1.2% (0.2%) | 1.1% (0.3%) | 0.7% (0.2%) |
| WIND | 1.0% (0.2%) | 0.8% (0.1%) | 0.8% (0.2%) | 0.9%(0.1%) | 1.2% (0.4%) | 1.0% (0.4%) |
| CPU | 1.6% (0.5%) | 1.2% (0.2%) | 0.7% (0.1%) | 1.3% (0.2%) | 1.3% (0.2%) | 0.9% (0.1%) |
| 2DPLANES | 1.7% (0.1%) | 1.9%(0.1%) | 0.8% (0.1%) | 1.3%(0.1%) | 1.6%(0.2%) | 1.1% (0.4%) |
| POL | 1.8% (0.4%) | 2.0% (0.2%) | 1.5% (0.5%) | 2.1% (0.2%) | 2.0% (0.2%) | 1.6% (0.5%) |

### B.2 PROOF OF PROPOSITION 3.1 AND ITS EXTENSION TO $K \geq 2$ BASE UTILITIES

This section provides the formal proof of Proposition B.1, which generalizes Proposition 3.1. It states that the semivalue score of any data point under a utility that is a linear combination of $K$ base utilities can be written as an inner product in $\mathbb{R}^K$. This result forms the backbone of the geometric perspective developed in Section 3.

**Proposition B.1** (Extension of Proposition 3.1 to $K \geq 2$ base utilities). *Let $\mathcal{D}$ be any dataset of size $n$ and let $\omega \in \mathbb{R}^n$ be a semivalue weight vector. Then there exists a map $\psi_{\omega, \mathcal{D}} : \mathcal{D} \longrightarrow \mathbb{R}^K$ such that for every utility $u_\alpha = \sum_{k=1}^K \alpha_k u_k$, $\phi(z; \omega, u_\alpha) = \langle \psi_{\omega, \mathcal{D}}(z), \alpha \rangle$, for any $z \in \mathcal{D}$. We call $\mathcal{S}_{\omega, \mathcal{D}} = \{\psi_{\omega, \mathcal{D}}(z) \mid z \in \mathcal{D}\}$ the spatial signature of $\mathcal{D}$ under semivalue $\omega$.*

The proof is a straightforward application of semivalue linearity. The main contribution is the geometric interpretation of semivalue vectors as projections.

*Proof.* For each data point $z \in \mathcal{D}$, let its semivalue characterized by $\omega$ be denoted by $\varphi(z; \omega, u_\alpha)$ when the utility is $u_\alpha$. Under the standard linearity property of semivalues, the following linear decomposition holds:

$$\phi\Big(z; \omega, u_\alpha\Big) = \phi\Big(z; \omega, \sum_{k=1}^K \alpha_k u_k\Big) = \sum_{k=1}^K \alpha_k \phi(z; \omega, u_k).$$

So if we define for each $z$,

$$\psi_{\omega,\mathcal{D}}(z) = \Big(\phi(z;\omega,u_1),\dots,\phi(z;\omega,u_K)\Big) \in \mathbb{R}^K,$$

then by definition of the scalar (inner) product in $\mathbb{R}^K$,

$$\phi(z;\omega,u_\alpha) = \langle \psi_{\omega,\mathcal{D}}(z),\alpha \rangle.$$

$\square$

In the main text, we focus on the case where $K = 2$, i.e., utilities correspond to directions on the unit circle $\mathcal{S}^1$. Proposition B.1 shows that the same reasoning carries over to any finite family of $K$ base utilities: data points embed as $\psi_{\omega,\mathcal{D}}(z) \in \mathbb{R}^K$, and ranking by a convex combination $u_\alpha = \sum_{k=1}^K \alpha_k u_k$ is equivalent to sorting the inner products $\langle \psi_{\omega,\mathcal{D}}(z),\alpha \rangle$. Since only the direction of $\alpha$ matters, each utility is identified with a point $\bar{\alpha} = \alpha/\|\alpha\|$ on the unit sphere $\mathcal{S}^{K-1}$. Thus, for general $K$, robustness to utility choice reduces to studying how the ordering of these projections varies as $\bar{\alpha}$ moves over $\mathcal{S}^{K-1}$.

### B.3 Ranking regions counts for specific cases of spatial signatures

This section formalizes the notion of *ranking regions*, which play a central role in the robustness analysis developed in Section 3. We begin by considering the hyperplane arrangement induced by all pairwise differences between embedded data points in the spatial signature. This arrangement partitions space into connected components, referred to as *regions* in the theory of hyperplane arrangements (see Definition B.2). In our context, each such region corresponds to a set of utility directions under which the ordering of data points remains constant. We refer to these as *ranking regions*.

**Definition B.2** (Region of a hyperplane arrangement). Let $\mathcal{A} \subset V$ be a finite arrangement of hyperplanes in a real vector space $V$. The *regions* of $\mathcal{A}$ are the connected components of

$$V \setminus \bigcup_{H \in \mathcal{A}} H.$$

Each region is the interior of a (possibly unbounded) polyhedral cone and is homeomorphic to $V$. We denote the number of such regions by $r(\mathcal{A})$.

We now specialize Definition B.2 to our data valuation setting. Let $\mathcal{D} = \{z_1,\dots,z_n\}$ be a dataset and let $\psi_{\omega,\mathcal{D}}(z_i) \in \mathbb{R}^K$ denote the embedding of each point under semivalue weighting $\omega$. For each pair $i < j$, we define

$$H_{ij} = \big\{\alpha \in \mathbb{R}^K : \langle \alpha, \psi_{\omega,\mathcal{D}}(z_i) - \psi_{\omega,\mathcal{D}}(z_j)\rangle = 0\big\}.$$

Each set $H_{ij}$ is defined as the kernel of the linear functional $\alpha \mapsto \langle \alpha, \psi_{\omega,\mathcal{D}}(z_i) - \psi_{\omega,\mathcal{D}}(z_j)\rangle$. Since $\psi_{\omega,\mathcal{D}}(z_i) \neq \psi_{\omega,\mathcal{D}}(z_j)$ for $i \neq j$ (unless the data points are embedded identically), the difference vector $\psi_{\omega,\mathcal{D}}(z_i) - \psi_{\omega,\mathcal{D}}(z_j) \in \mathbb{R}^K$ is nonzero. Therefore, this kernel is a linear subspace of codimension one in $\mathbb{R}^K$, which, by definition, is a hyperplane. Moreover, each $H_{ij}$ contains the origin $\alpha = 0_K$; it is thus a central hyperplane by definition.

Each hyperplane $H_{ij}$ is the set of utility directions that assign equal projection scores to points $z_i$ and $z_j$. The finite arrangement $\mathcal{A}_{\omega,\mathcal{D}} = \{H_{ij} : 1 \leq i < j \leq n\}$ then induces a collection of regions in the sense of Definition B.2, partitioning $\mathbb{R}^K$ into open cones such that, in each region, the relative ordering of projected values $\langle \alpha, \psi_{\omega,\mathcal{D}}(z_i)\rangle$ and $\langle \alpha, \psi_{\omega,\mathcal{D}}(z_j)\rangle$ remains the same for all $i < j$. Therefore, each region determines a unique ordering of the embedded points, corresponding to a distinct way of ranking the data points of $\mathcal{D}$ based on utility direction. To study robustness with respect to directional changes, we project this arrangement onto the unit sphere $\mathcal{S}^{K-1}$. Since all hyperplanes are central, their intersection with the sphere produces great spheres, and the resulting decomposition of $\mathcal{S}^{K-1}$ consists of spherical connected regions over which the ranking of the data points remains invariant. We refer to these regions as *ranking regions*. Formally, a *ranking region* is a connected component of $\mathcal{S}^{K-1} \setminus \bigcup_{i<j} \big(H_{ij} \bigcap \mathcal{S}^{K-1}\big)$.

We now study how the number of such ranking regions depends on the geometry of the spatial signature. In particular, using Proposition B.3, we provide an explicit count of ranking regions in two specific geometric configurations of the embedded points.

**Proposition B.3** (Regions counts). *Let $\mathcal{A} = \{H_1, \ldots, H_m\}$ be an arrangement of $m$ central (i.e., origin-passing) hyperplanes in a real vector space $V$ of dimension $K$.*

1. *If no $K$ hyperplanes in the arrangement intersect in a common subspace of dimension greater than zero (in particular, not in a line), then the number of regions into which $\mathcal{A}$ partitions $V$ is*

$$r(\mathcal{A}) = 2 \sum_{k=0}^{K-1} \binom{m-1}{j}.$$

2. *If all hyperplanes coincide (i.e., $H_1 = \cdots = H_m$), then the number of regions is:*
$$r(\mathcal{A}) = 2.$$

*Proof.* Let $\mathcal{A} = \{H_1, \ldots, H_m\}$ be an arrangement of $m$ hyperplanes in a real vector space $V$ of dimension $K$.

1. Suppose no $K$ hyperplanes in the arrangement intersect in a common subspace of dimension greater than zero (in particular, not in a line).

   Choose any hyperplane $H \in \mathcal{A}$, and define two affine hyperplanes $H^+$ and $H^-$, parallel to $H$ and on opposite sides of the origin, such that the origin lies strictly between them.

   Each of the remaining $m - 1$ hyperplanes of $\mathcal{A}$ intersects $H^+$ in a hyperplane of dimension $K - 2$, and these intersections form an arrangement of $m - 1$ hyperplanes in $H^+$ (which is a space of dimension $K - 1$). By Proposition 2.4 in Stanley (2007) (derived from Zaslavsky's work Zaslavsky (1975)), the number of regions induced by this non-central[4] arrangement in $H^+$ is:

$$\sum_{j=0}^{n-1} \binom{m-1}{j}$$

   These regions correspond exactly to the regions of $V \setminus \bigcup_{H \in \mathcal{A}} H$ that lie entirely on one side of $H$. By symmetry, the same number of regions lies on the opposite side (on $H^-$). Therefore, the total number of regions for the whole arrangement is:

$$r(\mathcal{A}) = 2 \sum_{j=0}^{n-1} \binom{m-1}{j}$$

2. Suppose that all hyperplanes in the arrangement coincide, i.e., $H_1 = \cdots = H_m = H$ for some hyperplane $H \subset V$. Then

$$\bigcup_{H \in \mathcal{A}} H = H,$$

   and the complement $V \setminus \bigcup_{H \in \mathcal{A}} H$ consists of exactly two connected open sets: the two half-spaces determined by $H$. Therefore, the number of regions is
$$r(\mathcal{A}) = 2.$$

$\square$

We now apply Proposition B.3 to the arrangement $\mathcal{A}_{\omega, \mathcal{D}}$ formed by the hyperplanes $H_{ij}$ defined from pairwise differences of embedded points in the spatial signature $S_{\omega, \mathcal{D}}$. Since each $H_{ij}$ is a central hyperplane in $\mathbb{R}^K$, the arrangement $\mathcal{A}_{\omega, \mathcal{D}}$ partitions the space into open polyhedral cones, whose connected components are the regions of the arrangement. Each of these cones intersects the unit sphere $\mathcal{S}^{K-1}$ in a unique open subset, yielding a spherical partition of $\mathcal{S}^{K-1}$. Therefore, the number of ranking regions on $\mathcal{S}^{K-1}$ is equal to the number of regions of the central hyperplane arrangement in $\mathbb{R}^K$, and can be computed directly using Proposition B.3.

Corollary B.4 provides the number of ranking regions for two specific geometric configurations of the spatial signature.

---

[4]Since the $m - 1$ hyperplanes do not all pass through a same point on $H^+$.

**Corollary B.4** (Ranking regions counts). *Let $\mathcal{D} = \{z_1, \ldots, z_n\}$ be a dataset and let $\psi_{\omega, \mathcal{D}}(z_i) \in \mathbb{R}^K$ denote the spatial signature of point $z_i$ under semivalue weighting $\omega$. For each pair $i < j$, define the hyperplane*

$$H_{ij} = \left\{\alpha \in \mathbb{R}^K : \langle \alpha, \psi_{\omega, \mathcal{D}}(z_i) - \psi_{\omega, \mathcal{D}}(z_j) \rangle = 0\right\}.$$

*Since there are $\binom{n}{2}$ pairs $(i, j)$, the arrangement $\mathcal{A}_{\omega, \mathcal{D}} = \{H_{ij} : 1 \le i < j \le n\}$ consists of $N = \binom{n}{2}$ central hyperplanes in $\mathbb{R}^K$. Let $r(\mathcal{A}_{\omega, \mathcal{D}})$ denote the number of connected regions in the complement of this arrangement. Then*

1. *If no $K$ hyperplanes $H_{ij}$ intersect in a common subspace of dimension greater than zero, the number of ranking regions is*

$$r(\mathcal{A}_{\omega, \mathcal{D}}) = 2 \sum_{k=0}^{K-1} \binom{N-1}{k}, \quad \text{where } N = \binom{n}{2}.$$

2. *If all embedded points $\psi_{\omega, \mathcal{D}}(z_i)$ lie on a line in $\mathbb{R}^K$, then all hyperplanes $H_{ij}$ coincide and*

$$r(\mathcal{A}_{\omega, \mathcal{D}}) = 2.$$

Figure 2 illustrates these two specific geometric configurations on the circle $\mathcal{S}^1$ (corresponding to the case $K = 2$). In both cases, the observed number of ranking regions coincides with the counts given by Corollary B.4.

### B.4 LINK BETWEEN THE ROBUSTNESS METRIC $R_p$ AND THE KENDALL RANK CORRELATION

If there are no tied ranks, the Kendall rank correlation between two orderings of $n$ points is defined as $\tau = 1 - \frac{2D}{N}$, where $D$ is the number of discordant pairs and $N = \binom{n}{2}$ is the total number of pairs. Since crossing one ranking region swaps exactly one pair, each such swap increases $D$ by one and thus decreases $\tau$ by $2/N$. Consequently, $p$ swaps lower the correlation from 1 to $1 - \frac{2p}{N}$. Therefore, $R_p$ captures how far in expectation one must move from a utility direction before the Kendall rank correlation degrades by at least $2p/N$.

However, this statement only holds in the setting where no ties occur. In practical scenarios involving ties, the degradation in $\tau$ can be either smaller or larger than what $R_p$ would suggest. The purpose of this subsection is to explain why *worse-than-expected degradation* is possible, which is the main risk when interpreting $R_p$ through the lens of Kendall correlation in practice.

The Kendall rank correlation between rankings $A$ and $B$ is defined as

$$\tau = \frac{c - d}{\sqrt{(N - t_A)(N - t_B)}},$$

where $c$ is the number of concordant pairs, $d$ is the number of discordant pairs ($c$ and $d$ count only untied pairs), $N = \binom{n}{2}$ is the total number of pairs, $t_A$ (resp. $t_B$) is the number of tied pairs in ranking $A$ (resp. $B$).

Performing $p$ pairwise swaps among tied items can amplify the degradation of $\tau$ beyond the idealized $-2p/N$ amount (derived under the no-ties assumption) due to two effects:

– Resolving ties i.e., decreasing $t_A$ or $t_B$, increases the factors $N - t_A$ or $N - t_B$ and thus the denominator. For a fixed numerator $c - d$, this directly reduces the magnitude of $\tau_b$. Critically, even as $c - d$ decreases (due to increased discordance), the growing denominator further exacerbates the decline.

– Swapping two items within a block of $k$ tied points can order up to $\binom{k}{2}$ formerly tied pairs at once. If these newly ordered pairs are discordant, a single swap increases $d$ by up to $\binom{k}{2}$, rather than just 1.

Consequently, when many tied groups exist, one might observe after $p$ swaps,

$$\Delta \tau_b < -\frac{2p}{N},$$

i.e., a larger drop in rank correlation than in the no-ties case.

## B.5 CLOSED-FORM FOR $\mathbb{E}_{\bar{\alpha}}[\rho_p(\bar{\alpha})]$

This section provides the derivation of a closed-form expression for $\mathbb{E}[\rho_p]$, introduced in Section 3, which quantifies how far, on average, one must rotate the utility direction on the sphere before $p$ pairwise ranking swaps occur. In Section 3, we describe how this quantity captures the local stability of the ranking induced by the spatial signature. Here, we formally compute this quantity in the case where $K = 2$ (i.e., in the case where the utilities we consider can be written as a linear combination of two base-utilities). We also show that the closed-form expression derived in the $K = 2$ case can be computed in $\mathcal{O}(n^2 \log n)$ time. Finally, we briefly discuss the higher-dimensional case $K > 2$, for which no closed-form is available, and describe how $\mathbb{E}[\rho_p]$ can be approximated via Monte Carlo sampling.

Recall that $\rho_p(\bar{\alpha})$ measures the minimal geodesic distance one must rotate a utility direction $\bar{\alpha} \in \mathcal{S}^{K-1}$ before the ranking of points in $\mathcal{D} = \{z_i\}_{i \in [n]}$ changes by $p$ pairwise swaps. Each pair of points $(z_i, z_j)$ defines *cuts* on $\mathcal{S}^{K-1}$ which are utility directions along which the scores of $z_i$ and $z_j$ are equal. These cuts partition $\mathcal{S}^{K-1}$ into (ranking) regions where the ranking of points remains fixed. In what follows, we focus on the case $K = 2$, where utility directions lie on the unit circle $\mathcal{S}^1$, and $\rho_p(\bar{\alpha})$ can be treated as a function of the angle associated with $\bar{\alpha} \in \mathcal{S}^1$.

We parametrize the unit circle $\mathcal{S}^1$ by the angle $\varphi \in [0, 2\pi[$, writing $\bar{\alpha}(\varphi) = (\cos\varphi, \sin\varphi) \in \mathcal{S}^1$. Since $\rho_p$ depends only on this angle, we abbreviate $\rho_p(\bar{\alpha}(\varphi))$ by $\rho_p(\varphi)$. Equivalently,

$$\mathbb{E}_{\bar{\alpha} \sim \text{Unif}(\mathcal{S}^1)}[\rho_p(\bar{\alpha})] = \mathbb{E}_{\varphi \sim \text{Unif}[0, 2\pi[}[\rho_p(\varphi)] = \frac{1}{2\pi} \int_0^{2\pi} \rho_p(\varphi) d\varphi.$$

Recall from Section 3 that the $2N$ emphcut angles $\theta_1 \leq \theta_2 \leq \cdots < \theta_{2N} < 2\pi$ partition the interval $[0, 2\pi[$ into arcs of lengths $\lambda_k = \theta_{k+1} - \theta_k$ (with $\theta_{2N+1} = \theta_1 + 2\pi$). Within each arc, the ranking remains fixed, and crossing into the next arc incurs exactly one additional swap.

For $\varphi \in (\theta_k, \theta_{k+1})$, the function $\rho_p(\varphi)$ equals the minimum of the clockwise and counterclockwise distances to the $p$-th next cut:

$$\rho_p(\varphi) = \min\{S_k^+(p) - (\varphi - \theta_k), S_k^-(p) + (\varphi - \theta_k)\},$$

where the quantities $S_k^+(p)$ and $S_k^-(p)$, recalled from Section 3, are defined as

$$S_k^+(p) = \sum_{i=1}^p \lambda_{(k+i) \bmod 2N}, \quad S_k^-(p) = \sum_{i=1}^p \lambda_{(k-i) \bmod 2N}.$$

Hence, the average value of $\rho_p$ can be written as

$$\mathbb{E}_{\bar{\alpha}}[\rho_p(\bar{\alpha})] = \frac{1}{2\pi} \int_0^{2\pi} \rho_p(\varphi) d\varphi = \frac{1}{2\pi} \sum_{k=1}^{2N} \int_{\theta_k}^{\theta_{k+1}} \rho_p(\varphi) d\varphi = \frac{1}{2\pi} \sum_{k=1}^{2N} \int_0^{\lambda_k} \min\{S_k^+(p) - t, S_k^-(p) + t\} dt, \tag{3}$$

where we set $t = \varphi - \theta_k \in [0, \lambda_k]$ as a local coordinate that measures the angular distance from the left endpoint of the $k$-th arc.

The expression inside the integral reflects the shortest of two angular paths along the circle from the start of the $k$-th arc: one going clockwise (of length $S_k^+(p) - t$) and one counterclockwise (of length $S_k^-(p) + t$). These two expressions intersect at

$$t_k^* = \frac{S_k^+(p) - S_k^-(p)}{2}.$$

Intuitively,

- If $t_k^* \leq 0$, even at $t = 0$, the clockwise path is already shorter, so $\rho_p(t) = S_k^+(p) - t$ for all $t \in [0, \lambda_k]$.

- If $t_k^* \geq \lambda_k$, the counterclockwise path is shorter throughout the entire arc, $\rho_p(t) = S_k^-(p) + t$ for all $t \in [0, \lambda_k]$.

- If $0 < t_k^\star < \lambda_k$, then for $t < t_k^\star$ the counterclockwise path is shorter, and for $t > t_k^\star$, the clockwise path is shorter.

We therefore split $\int_0^{\lambda_k} \rho_p(t)dt = \int_0^{\lambda_k} \min\{S_k^+(p) - t, S_k^-(p) + t\}dt$ into the three cases:

1. If $t_k^\star \leq 0$, we have

$$\int_0^{\lambda_k} \min\{S_k^+(p) - t, S_k^-(p) + t\}dt = \int_0^{\lambda_k} (S_k^+(p) - t)dt = S_k^+(p)\lambda_k - \frac{1}{2}\lambda_k^2$$

2. If $t_k^\star \geq \lambda_k$, we have

$$\int_0^{\lambda_k} \min\{S_k^+(p) - t, S_k^-(p) + t\}dt = \int_0^{\lambda_k} (S_k^-(p) + t)dt = S_k^-(p)\lambda_k + \frac{1}{2}\lambda_k^2$$

3. If $0 < t_k^\star < \lambda_k$, we have

$$\int_0^{\lambda_k} \min\{S_k^+(p) - t, S_k^-(p) + t\}dt = \int_0^{t_k^\star} (S_k^-(p) + t)dt + \int_{t_k^\star}^{\lambda_k} (S_k^+(p) - t)dt$$

$$= S_k^-(p)t_k^\star + \frac{1}{2}(t_k^\star)^2 + S_k^+(p)(\lambda_k - t_k^\star)$$

$$- \frac{1}{2}(\lambda_k^2 - (t_k^\star)^2).$$

Putting these three cases together and summing over $k$ yields a piecewise-defined expression for the integral on each arc. Precisely, plugging these into the expression for $\mathbb{E}_{\bar{\alpha}}[\rho_p(\bar{\alpha})]$ in Eq. equation 3, we finally obtain the closed-form

$$\mathbb{E}_{\bar{\alpha}}[\rho_p(\bar{\alpha})] = \frac{1}{2\pi} \sum_{k=1}^{2N} I_k,$$

where $I_k$ denotes the value of the integral over the $k$-th arc, defined as

$$I_k := \int_0^{\lambda_k} \min\{S_k^+(p) - t, S_k^-(p) + t\}dt,$$

and can be computed using the following case distinction

$$I_k = \begin{cases} S_k^+(p)\lambda_k - \frac{1}{2}\lambda_k^2 & \text{if } t_k^* \leq 0, \\ S_k^-(p)\lambda_k + \frac{1}{2}\lambda_k^2 & \text{if } t_k^* \geq \lambda_k, \\ S_k^-(p)t_k^* + \frac{1}{2}(t_k^*)^2 + S_k^+(p)(\lambda_k - t_k^*) - \frac{1}{2}(\lambda_k^2 - (t_k^*)^2) & \text{if } 0 < t_k^* < \lambda_k, \end{cases}$$

with $t_k^* = \frac{S_k^+(p) - S_k^-(p)}{2}$ as previously defined.

**Remark B.5** (Computational cost). The closed-form expression for $\mathbb{E}[\rho_p]$ can be computed in $\mathcal{O}(n^2 \log n)$ time. First, computing the $2N = \mathcal{O}(n^2)$ cut angles defined by all unordered pairs of points $(z_i, z_j)$ requires $\mathcal{O}(n^2)$ time, since each involves a simple trigonometric operation in $\mathbb{R}^2$. Sorting these angles to define the arc intervals costs $\mathcal{O}(n^2 \log n)$. Once sorted, the distances $S_k^+(p)$ and $S_k^-(p)$ to the $p$-th next and previous cuts can be computed efficiently for all $k$ using sliding windows indexing in $\mathcal{O}(n^2)$ time. The final step, i.e., evaluating the integral over each of the $2N$ arcs, also takes $\mathcal{O}(n^2)$ time. Thus, the total computational cost is $\mathcal{O}(n^2 \log n)$, dominated by the sorting step.

**Remark B.6** (Case $K > 2$). The above closed-form derivation relies on the fact that utilities correspond to angles on $\mathcal{S}^1$ (i.e., $K = 2$). When $K > 2$, utilities lie on the higher-dimensional sphere $\mathcal{S}^{K-1}$. In that setting, one can still define $\rho_p(\bar{\alpha})$ as the minimal geodesic distance on $\mathcal{S}^{K-1}$ to incur $p$ swaps, but the integral $\mathbb{E}_{\bar{\alpha} \sim \text{Unif}(\mathcal{S}^{K-1})}[\rho_p(\bar{\alpha})]$ admits no simple closed-form expression. In practice, one must approximate it numerically by Monte Carlo sampling. Specifically, for any unit vector $\bar{\alpha} \in \mathcal{S}^{K-1}$, each pair $(i, j)$ defines a *cut* great–sphere

$$H_{ij} = \{\beta \in \mathcal{S}^{K-1} : \langle \beta, v_{ij} \rangle = 0\} \quad v_{ij} = \psi(z_i) - \psi(z_j).$$

The shortest geodesic distance from $\bar{\alpha}$ to that cut is given in closed form by

$$d_{ij}(\bar{\alpha}) = \arcsin\left|\langle \bar{\alpha}, v_{ij}/\|v_{ij}\|\rangle\right|.$$

Thus, by getting all $N = \binom{n}{2}$ distances $\{d_{ij}\}$, sorting them, and picking the $p$-th smallest, we obtain $\rho_p(\bar{\alpha})$. Repeating for many independent $\bar{\alpha} \sim \text{Unif}(\mathcal{S}^{K-1})$ gives a Monte Carlo estimate of the average.

Let $\hat{\mu}_m := \frac{1}{m}\sum_{\ell=1}^{m} \rho_p(\bar{\alpha}^{(\ell)})$ with i.i.d. draws $\bar{\alpha}^{(\ell)} \sim \text{Unif}(\mathcal{S}^{K-1})$, and let $\mu := \mathbb{E}_{\bar{\alpha}}[\rho_p(\bar{\alpha})]$. Since $0 \leq \rho_p(\bar{\alpha}) \leq \pi/2$, Hoeffding's inequality gives, for any $\delta \in (0,1)$,

$$\mathbb{P}\left(\left|\hat{\mu}_m - \mu\right| \geq \frac{\pi}{2}\sqrt{\frac{\log(2/\delta)}{2m}}\right) \leq \delta.$$

Equivalently, to guarantee $\left|\hat{\mu}_m - \mu\right| \leq \varepsilon$ with probability at least $1 - \delta$, it suffices that

$$m \geq \frac{\pi^2}{8\,\varepsilon^2}\log\frac{2}{\delta}.$$

## B.6 MAXIMUM AVERAGE $p$-SWAPS DISTANCE OCCURS UNDER COLLINEARITY OF THE SPATIAL SIGNATURE

This section provides the theoretical justification for the claim made in Section 3 that the average distance $\mathbb{E}_{\bar{\alpha}}[\rho_p(\bar{\alpha})]$ is maximized when the spatial signature is collinear, and equals $\pi/4$ in this case.

Recall that the spatial signature is the set of embedded vectors

$$\mathcal{S}_{\omega,\mathcal{D}} = \{\psi_{\omega,\mathcal{D}}(z) \in \mathbb{R}^K : z \in \mathcal{D}\},$$

where $\psi_{\omega,\mathcal{D}}(z)$ reflects the contribution of each data point $z \in \mathcal{D}$ to a family of $K$ base utilities, weighted by the semivalue coefficients $\omega$. This embedding allows utility directions $\bar{\alpha} \in \mathcal{S}^{K-1}$ to induce rankings via projection. The quantity $\rho_p(\bar{\alpha})$ measures the minimal geodesic distance on the sphere $\mathcal{S}^{K-1}$ one must rotate $\bar{\alpha}$ before the ranking changes by $p$ pairwise swaps. We focus here on the case $K = 2$, where utility directions lie on the unit circle $\mathcal{S}^1$, and show that the maximum of $\mathbb{E}[\rho_p]$ is achieved when all embedded points lie on a common line through the origin. This derivation provides an upper bound for $\mathbb{E}[\rho_p]$ and motivates the normalization in the robustness metric $R_p$ (see Definition 3.2).

Each pair of points $(z_i, z_j)$ induces a *cut* on the circle $\mathcal{S}^1$, namely the two antipodal points where $\langle \alpha, \psi_{\omega,\mathcal{D}}(z_i) - \psi_{\omega,\mathcal{D}}(z_j)\rangle = 0$. When all embedded points $\psi_{\omega,\mathcal{D}}(z_i)$ lie on a single line through the origin, Corollary B.4 states that there is exactly one cut (of multiplicity $N = \binom{n}{2}$) which splits $\mathcal{S}^1$ into two open arcs, each of length $\pi$. Within either arc, no swaps occur until one crosses that cut, at which point all $N$ pairs flip simultaneously. Concretely, for any direction angle $\theta \in [0, \pi[$, $\rho_p(\theta)$ corresponds to the shortest angular distance to this cut, either clockwise or counterclockwise, and is thus given by

$$\rho_p(\theta) = \min\{\theta, \pi - \theta\},$$

Hence,

$$\mathbb{E}_{\bar{\alpha}}[\rho_p(\bar{\alpha})] = \frac{1}{2\pi}\int_0^{2\pi} \rho_p(\varphi)d\varphi = \frac{1}{2\pi}\cdot 2\int_0^{\pi}\min\{\theta, \pi - \theta\}d\theta = \frac{\pi}{4}.$$

Now, any deviation from perfect collinearity introduces distinct cuts, which can only further subdivide those two $\pi$-length arcs into shorter pieces. Shorter maximal arc-lengths imply a smaller average distance to the nearest swap, so for every spatial signature and every $1 \leq p < N$,

$$\mathbb{E}[\rho_p] \leq \frac{\pi}{4},$$

with equality if and only if the signature is exactly collinear.

**Remark B.7** (Case $K > 2$). For $K > 2$, perfect collinearity of the spatial signature still maximizes the average $p$-swap distance, but the value $\max_{\mathcal{S}_{\omega,\mathcal{D}}} \mathbb{E}_{\bar{\alpha}}[\rho_p(\bar{\alpha})]$ must be evaluated numerically since for $K > 2$ the distribution of angular distances from a uniformly random point on the sphere to a fixed great sub-sphere no longer admits a simple elementary integral like for $K = 2$.

**Remark B.8** (Lower bounds for $\mathbb{E}_{\bar{\alpha}}[\rho_p(\bar{\alpha})]$). Trivially, since $\rho_p(\bar{\alpha}) \geq 0$ for all $\bar{\alpha}$,

$$\mathbb{E}_{\bar{\alpha}}[\rho_p(\bar{\alpha})] \geq 0, \quad \forall p < \binom{n}{2}.$$

If instead we assume the spatial signature to be such that all $N = \binom{n}{2}$ cuts on $\mathcal{S}^1$ are distinct, then Proposition B.3 states that there are exactly $2N$ positive-length arcs of total length $2\pi$. In this case, it is easy to see that considering all ways to choose $\{\lambda_k\}$ summing to $2\pi$, the configuration $\lambda_k = \pi/N$ for all $k$ minimizes the average $\rho_p$. Concretely, for $\lambda_k = \pi/N$ we find

$$\mathbb{E}[\rho_p] = \frac{1}{2\pi} \sum_{k=1}^{2N} \int_0^{\pi/N} \left( p\frac{\pi}{N} - t \right) dt = (p - \tfrac{1}{2})\frac{\pi}{N}.$$

Hence, under the distinct cuts assumption,

$$\mathbb{E}[\rho_p(\varphi)] \geq \left( p - \tfrac{1}{2} \right)\frac{\pi}{N},$$

with equality exactly when the $2N$ cuts are perfectly equally spaced.

In this special setting, where all $\binom{n}{2}$ cuts on $\mathcal{S}^1$ are distinct, one could alternatively define the robustness metric as

$$R_p(S_{\omega,\mathcal{D}}) = \frac{\mathbb{E}_{\bar{\alpha}}[\rho_p(\bar{\alpha})] - (p - \tfrac{1}{2})\frac{\pi}{N}}{\pi/4 - (p - \tfrac{1}{2})\frac{\pi}{N}} \in [0, 1].$$

However, since in practice we cannot detect a priori that this condition on cuts holds, we instead use the general robustness metric $R_p$ as given in Definition 3.2.

## B.7 PROOF OF PROPOSITION 3.3

In this section, we provide the detailed proof of Proposition 3.3. Recall that for any utility $u$,

$$\phi(z_i; \omega, u) = \sum_{j=1}^n \omega_j \, \Delta_j(z_i, u),$$

where $\Delta_j(z_i, u)$ is the marginal contribution of $z_i$ with respect to coalitions of size $j - 1$. By definition of the covariance,

$$\mathrm{Cov}\big(\phi(\lambda), \phi(\gamma)\big) = \frac{1}{n} \sum_{i=1}^n \big(\phi(z_i; \omega, \lambda) - \bar{\phi}(\lambda)\big)\big(\phi(z_i; \omega, \gamma) - \bar{\phi}(\gamma)\big),$$

where $\bar{\phi}(\cdot)$ denotes the mean over $i$. Using billinearity of covariance, we get

$$\mathrm{Cov}\big(\phi(\lambda), \phi(\gamma)\big) = \sum_{j=1}^n \sum_{k=1}^n \omega_j \omega_k \, \mathrm{Cov}\big(\Delta_j(\lambda), \Delta_k(\gamma)\big).$$

where

$$\Delta_j(\lambda) = \Big(\Delta_j(z_1; \omega, \lambda), \ldots, \Delta_j(z_n; \omega, \lambda)\Big) \quad \text{and} \quad \Delta_j(\gamma) = \Big(\Delta_j(z_1; \omega, \gamma), \ldots, \Delta_j(z_n; \omega, \gamma)\Big)$$

Under the assumption $\mathrm{Cov}(\Delta_j(\lambda), \Delta_k(\gamma)) = 0$ for all $j \neq k$, only the $j = k$ terms remain, giving

$$\mathrm{Cov}\big(\phi(\lambda), \phi(\gamma)\big) = \sum_{j=1}^n \omega_j^2 \, \mathrm{Cov}\big(\Delta_j(\lambda), \Delta_j(\gamma)\big).$$

Similarly,

$$\mathrm{Var}\big(\phi(\lambda)\big) = \sum_{j=1}^n \omega_j^2 \, \mathrm{Var}\big(\Delta_j(\lambda)\big), \quad \mathrm{Var}\big(\phi(\gamma)\big) = \sum_{j=1}^n \omega_j^2 \, \mathrm{Var}\big(\Delta_j(\gamma)\big).$$

By the definition of Pearson correlation,

$$
\mathrm{Corr}\big(\phi(\lambda),\phi(\gamma)\big) = \frac{\displaystyle\sum_{j=1}^{n}\omega_j^2\,\mathrm{Cov}(\Delta_j(\lambda),\Delta_j(\gamma))}{\sqrt{\displaystyle\sum_{j=1}^{n}\omega_j^2\,\mathrm{Var}(\Delta_j(\lambda))}\sqrt{\displaystyle\sum_{j=1}^{n}\omega_j^2\,\mathrm{Var}(\Delta_j(\gamma))}}.
$$

with $\mathrm{Cov}(\Delta_j(\lambda),\Delta_j(\gamma)) = \mathrm{Corr}(\Delta_j(\lambda),\Delta_j(\gamma))\sqrt{\mathrm{Var}(\Delta_j(\lambda))\,\mathrm{Var}(\Delta_j(\gamma))}$. Then, the correlation becomes

$$
\mathrm{Corr}\big(\phi(\lambda),\phi(\gamma)\big) = \sum_{j=1}^{n}\omega_j^2\,\frac{\mathrm{Corr}(\Delta_j(\lambda),\Delta_j(\gamma))\sqrt{\mathrm{Var}(\Delta_j(\lambda))\,\mathrm{Var}(\Delta_j(\gamma))}}{\sqrt{\displaystyle\sum_{j=1}^{n}\omega_j^2\,\mathrm{Var}(\Delta_j(\lambda))}\sqrt{\displaystyle\sum_{j=1}^{n}\omega_j^2\,\mathrm{Var}(\Delta_j(\gamma))}}
$$

Each term $r_j := \mathrm{Corr}\big(\Delta_j(\lambda),\Delta_j(\gamma)\big)\sqrt{\mathrm{Var}\big(\Delta_j(\lambda)\big)\,\mathrm{Var}\big(\Delta_j(\gamma)\big)}$ can be understood as the *effective alignment* of marginal contributions at coalition size $j-1$ across the two utilities $\lambda$ and $\gamma$. Specifically, $\mathrm{Corr}(\Delta_j(\lambda),\Delta_j(\gamma))$ measures how similarly data points' marginal contributions at size $j-1$ move under $\lambda$ compared to under $\gamma$ and $\sqrt{\mathrm{Var}(\Delta_j(\lambda))\,\mathrm{Var}(\Delta_j(\gamma))}$ down-weights sizes $j-1$ where marginal contributions are nearly constant (and thus uninformative) for either utility.

### B.8 Link between the robustness metric $R_p$ and top-$k$ stability metrics (Overlap@$k$ and Jaccard@$k$)

In this section, we give an analytical link between $p$ and top-$k$ overlap/Jaccard stability metrics which definitions are provided in Appendix C.6.

Let $\mathcal{D}$ be a training set of size $n$. Let $u$ and $u'$ be two utilities (linear combinations of base utilities) that induce rankings $\pi$ and $\pi'$ on $\mathcal{D}$, and assume $\pi$ and $\pi'$ differ by at most $p$ swaps. Let

$$
S_u^{(k)} := S_{\phi(u,\omega)}^{(k)}, \qquad S_{u'}^{(k)} := S_{\phi(u',\omega)}^{(k)}
$$

denote the top-$k$ sets under $u$ and $u'$ respectively. We then recall from Appendix C.6 the definitions of top-$k$ overlap@$k$ and Jaccard@$k$:

— Top-$k$ overlap@$k$:

$$
\mathrm{Overlap@}k(u,u') := \frac{|S_u^{(k)} \cap S_{u'}^{(k)}|}{k} \in [0,1].
$$

— Jaccard@$k$:

$$
\mathrm{Jaccard@}k(u,u') := \frac{|S_u^{(k)} \cap S_{u'}^{(k)}|}{|S_u^{(k)} \cup S_{u'}^{(k)}|} \in [0,1].
$$

Since both sets have cardinality $k$, this simplifies to

$$
\mathrm{Jaccard@}k(u,u') = \frac{|S_u^{(k)} \cap S_{u'}^{(k)}|}{2k - |S_u^{(k)} \cap S_{u'}^{(k)}|}.
$$

Now let $A := S_u^{(k)}$ and $B := S_{u'}^{(k)}$. Let $L$ be the number of points that leave the top-$k$ set when moving from $A$ to $B$; then $|A \cap B| = k - L$, and the symmetric difference has size $|A \triangle B| = 2L$. Now, a data point can enter or leave the top-$k$ set only if its rank crosses the boundary between positions $k$ and $k+1$. This can happen only when we perform a swap involving the items currently at ranks $k$ and $k+1$. Each such boundary swap can change the membership of at most the two swapped items. Let $b$ be the number of boundary swaps among the $p$ swaps. Then at most $2b$ distinct items can change membership. Since $b \le p$, at most $2p$ distinct items can change membership. Because

$|A \triangle B| = 2L$, we get $2L \leq 2p$, hence $L \leq p$. Therefore, $|A \cap B| = k - L \geq k - p$. This yields the following deterministic bounds (for $p \leq k$; otherwise they become vacuous):

$$\text{Overlap@}k(u, u') = \frac{|A \cap B|}{k} \geq 1 - \frac{p}{k},$$

and since $|A \cup B| = |A| + |B| - |A \cap B| = 2k - |A \cap B|$, we have $|A \cup B| \leq 2k - (k-p) = k + p$, so

$$\text{Jaccard@}k(u, u') = \frac{|A \cap B|}{|A \cup B|} \geq \frac{k - p}{k + p}.$$

Now, by Definition 3.2, for each utility direction $\bar{\alpha} \in \mathbb{S}^{K-1}$, $\rho_p(\bar{\alpha})$ is the minimal geodesic distance such that moving by $\rho_p(\bar{\alpha})$ on the sphere produces exactly $p$ swaps in the ranking induced by $\bar{\alpha}$. So, for a fixed $p$ and $k$, a larger $R_p \propto \mathbb{E}_{\bar{\alpha}}[\rho_p(\bar{\alpha})]$ means that one must move farther in average in the utility space before reaching a regime where top-$k$ overlap/Jaccard can be as low as these bounds.

## C  ADDITIONAL DEFINITIONS

For the reader's convenience, we first outline the main points covered in this section.

- Appendix C.1: Some definitions of linear fractional utilities.
- Appendix C.2: Rank correlation metrics (Kendall & Spearman).
- Appendix C.3: Axioms satisfied by semivalues.
- Appendix C.4: Applications of semivalue-based data valuation methods.
- Appendix C.5: Extension of the *multiple-valide utility* scenario to multiclass classification metrics.
- Appendix C.6: Top-$k$ stability metrics (overlap@$k$ & Jaccard@$k$).

### C.1  SOME DEFINITIONS OF LINEAR FRACTIONAL UTILITIES

Below, we give the concrete coefficients $(c_0, c_1, c_2)$ and $(d_0, d_1, d_2)$ for several commonly used linear-fractional performance metrics. Each of these metrics can be expressed in the form

$$u(S) = \frac{c_0 + c_1 \lambda(S) + c_2 \gamma(S)}{d_0 + d_1 \lambda(S) + d_2 \gamma(S)}.$$

as recalled from equation 2.

Table 22: Some examples of *linear fractional* utilities. For more examples, see Choi et al. (2009). We set $\pi = \frac{1}{m} \sum_{j=1}^{m} \mathbf{1}[y_j = 1]$, the proportion of positive labels in $\mathcal{D}_{\text{test}}$.

| Utility | $(c_0, c_1, c_2)$ | $(d_0, d_1, d_2)$ |
|---|---|---|
| Accuracy | $(1 - \pi, 2, -1)$ | $(1, 0, , 0)$ |
| $F_\beta$-score | $(0, 1 + \beta^2, 0)$ | $(\beta^2 \pi, 0, 1)$ |
| Jaccard | $(0, 1, 0)$ | $(\pi, -1, 1)$ |
| AM-measure | $(\frac{1}{2}, \frac{2}{\pi} + \frac{2}{1-\pi}, -\frac{2}{1\pi})$ | $(1, 0, 0)$ |

### C.2  RANK CORRELATION METRICS (KENDALL & SPEARMAN)

Let $X = (x_1, x_2, \ldots, x_n)$ and $Y = (y_1, \ldots, y_n)$ be two real-valued score vectors on the same $n$ items. And let $\pi_X$ and $\pi_Y$ be their induced rankings. Rank correlations measure monotonic relationships between relative ordering $\pi_X$ and $\pi_Y$.

**Definition C.1** (Kendall rank correlation). Define the set of all pairs of distinct indices $\mathcal{P} = \{(i, j) : 1 \leq i < j \leq n\}$. For each $(i, j) \in \mathcal{P}$, call the pair *concordant* if $(x_i - x_j)(y_i - y_j) > 0$, *discordant* if $(x_i - x_j)(y_i - y_j) < 0$, and a *tie* in $X$ (resp. $Y$) if $x_i = x_j$ (resp. $y_i = y_j$).

Let $c$ the number of concordant pairs, $d$ the number of discordant pairs, and $t_X$ (resp. $t_Y$) the number of ties in $X$ (resp. $Y$). Then, the Kendall rank correlation $\tau$ is

$$\tau = \frac{c - d}{\sqrt{\left[\binom{n}{2} - t_X\right]\left[\binom{n}{2} - t_Y\right]}},$$

which simplify to $\tau = \frac{c-d}{\binom{n}{2}}$ if there are no ties ($t_X = t_Y = 0$).

**Definition C.2** (Spearman rank correlation). Let $\pi_X(i)$ be the rank of $x_i$ in $X$ and likewise $\pi_Y(i)$ for $Y$. Define the rank-differences $d_i = \pi_X(i) - \pi_Y(i)$. The Spearman rank correlation $s$ is the Pearson correlation of the ranked vectors:

$$s = \frac{\sum_{i=1}^{n}(\pi_X(i) - \bar{\pi}_X)(\pi_Y(i) - \bar{\pi}_Y)}{\sqrt{\sum_{i=1}^{n}(\pi_X(i) - \bar{\pi}_X)^2} \sqrt{\sum_{i=1}^{n}(\pi_Y(i) - \bar{\pi}_Y)^2}}$$

where $\bar{\pi}_X = \frac{1}{n}\sum_{i=1}^{n}\pi_X(i)$ and $\bar{\pi}_Y = \frac{1}{n}\sum_{i=1}^{n}\pi_Y(i)$. If there are no ties, it simplifies to

$$s = 1 - \frac{6\sum_{i=1}^{n}d_i^2}{n(n^2 - 1)}.$$

Both metrics lie in $[-1, 1]$, with $+1$ indicating perfect agreement and $-1$ perfect reversal.

### C.3 AXIOMS SATISFIED BY SEMIVALUES

Semivalues as defined in equation 1 satisfy fundamental axioms that ensure desirable properties in data valuation. We formally state these axioms in the following. Let $\phi(.,\omega;.)$ be a semivalue-based data valuation method defined by a weight vector $\omega$ and let $u$ and $v$ be utility functions. Then, $\phi$ satisfies the following axioms:

1. *Dummy*. If $u(S\cup\{z_i\}) = u(S) + c$ for all $S \subseteq \mathcal{D}\backslash\{z_i\}$ and some $c \in \mathbb{R}$, then $\phi(z_i;\omega,u) = c$.
2. *Symmetry*. If $u(S \cup \{z_j\}) = u(S \cup \{z_j\})$ for all $S \subseteq \mathcal{D}\backslash\{z_i, z_j\}$, then $\phi(z_i;\omega,u) = \phi(z_j;\omega,u)$.
3. *Linearity*. For any $\alpha_1, \alpha_2 \in \mathbb{R}$, $\phi(z_i;\omega,\alpha_1 u + \alpha_2 v) = \alpha_1\phi(z_i;\omega,u) + \alpha_2\phi(z_i;\omega,v)$.

While all semivalues satisfy the above axioms, Data Shapley uniquely also guarantees *efficiency*: $\sum_{z\in\mathcal{D}}\phi(z,\omega,u) = u(\mathcal{D})$.

### C.4 APPLICATIONS OF SEMIVALUE-BASED DATA VALUATION METHODS

In practice, semivalue-based methods are mostly applied to perform *data cleaning* or *data subset selection* (Tang et al., 2021; Pandl et al., 2021; Bloch & Friedrich, 2021; Zheng et al., 2024). Both tasks involve ranking data points according to their assigned values.

**Data cleaning.** Data cleaning aims to improve dataset quality by identifying and removing noisy or low-quality data points. Since semivalue-based methods quantify each point's contribution to a downstream task, low-valued points are natural candidates for removal. Specifically, a common approach is to remove points that fall into the set $\mathcal{N}_\tau$, defined as the subset of data points with the lowest values (Ghorbani & Zou, 2019). Formally, $\mathcal{N}_\tau = \{z_i \in \mathcal{D} \mid \phi(z_i;u,\omega) \leq \tau\}$, where $\tau$ is a threshold determined through domain knowledge or empirical evaluation.

**Data subset selection.** Data subset selection involves choosing the optimal training set from available samples to maximize final model performance. Since semivalues measure data quality, prioritizing data points with the highest values is a natural approach. Consequently, a common practice in the literature (Wang & Jia, 2023; Jiang et al., 2023; Wang et al., 2024b) is selecting, given a size budget $k$, the subset $\mathcal{S}_{\phi(u,\omega)}^{(k)}$ of data points with top-$k$ data values, i.e., $\mathcal{S}_{\phi(u,\omega)}^{(k)} = \arg\max_{\mathcal{S}\subseteq\mathcal{D},|\mathcal{S}|=k}\sum_{z_i\in\mathcal{S}}\phi(z_i;u,\omega)$.

### C.5 EXTENSION OF THE *multiple-valid utility* SCENARIO TO MULTICLASS METRICS

Let $\mathcal{Y} = \{1,\ldots,K\}$ be the class set and let $g_S$ be the model trained on $S$. For each class $k$, define the one-vs-rest confusion counts on the test set:

$$\text{TP}_k = \#\{y = k,\ g_S(x) = k\}, \quad \text{FN}_k = \#\{y = k,\ g_S(x) \neq k\}, \quad \text{FP}_k = \#\{y \neq k,\ g_S(x) = k\},$$

and the class supports $n_k = \text{TP}_k + \text{FN}_k$ (true instances of class $k$) and $\hat{n}_k = \text{TP}_k + \text{FP}_k$ (predicted as class $k$).

**Recall.** The per-class recalls form the $K$-vector

$$r(S) := \big(r_1(S),\ldots,r_K(S)\big) \in [0,1]^K, \qquad r_k(S) := \frac{\text{TP}_k}{\text{TP}_k + \text{FN}_k} = \frac{\text{TP}_k}{n_k}.$$

Any average recall can be written as a dot product with a weight vector $w \in \Delta_K := \{w \in \mathbb{R}_{\geq 0}^K : \sum_k w_k = 1\}$:

$$\mathrm{rec}_w(S) := \langle w, r(S) \rangle.$$

Two common choices are immediate:

$$\text{macro-recall: } w^{\mathrm{macro}} = \frac{1}{K}\mathbf{1}, \qquad \text{weighted-recall: } w_k^{\mathrm{wgt}} = \frac{n_k}{\sum_{\ell=1}^{K} n_\ell}.$$

Thus macro- and weighted-recall are the *same linear functional* applied to the per-class recall basis $r(S)$ with different $w$.

**Precision and $F_1$.** Analogously, define the per-class precisions

$$p(S) := \big(p_1(S), \ldots, p_K(S)\big), \qquad p_k(S) := \frac{\mathrm{TP}_k}{\mathrm{TP}_k + \mathrm{FP}_k} = \frac{\mathrm{TP}_k}{\hat{n}_k},$$

and per-class $F_1$'s

$$f_k(S) := \frac{2p_k(S)r_k(S)}{p_k(S) + r_k(S)} \quad (\text{with } f_k = 0 \text{ if } p_k + r_k = 0), \qquad f(S) := (f_1, \ldots, f_K).$$

Macro/weighted versions are again linear averages over the same class-wise basis:

$$\mathrm{prec}_w(S) = \langle w, p(S) \rangle, \qquad \mathrm{F1}_w(S) = \langle w, f(S) \rangle, \quad w \in \Delta_K,$$

with $w^{\mathrm{macro}}$ and $w^{\mathrm{wgt}}$ defined as above.

**Implication for our framework.** Let the *class-wise utilities* be $u_k^{\mathrm{rec}}(S) := r_k(S)$ (or $u_k^{\mathrm{prec}}(S) := p_k(S)$, $u_k^{\mathrm{F1}}(S) := f_k(S)$). Then any macro/weighted multiclass metric is a convex combination

$$u_w(S) = \sum_{k=1}^{K} w_k\, u_k(S), \qquad w \in \Delta_K.$$

By linearity of semivalues,

$$\phi\big(z; \omega, u_w\big) = \sum_{k=1}^{K} w_k\, \phi\big(z; \omega, u_k\big),$$

so the spatial signature lives in $\mathbb{R}^K$ with coordinates given by the class-wise utilities. Robustness to *all* convex mixtures $w \in \Delta_K$ is therefore a $K$-utility instance and $R_p$ is computed via the Monte Carlo procedure on $\mathcal{S}^{K-1}$ described in Appendix B.5.

## C.6 Top-$k$ stability metrics (Overlap@$k$ & Jaccard@$k$)

Given two rankings $\pi$ and $\pi'$ over the same dataset $D = \{z_1, \ldots, z_n\}$, we denote by $\mathrm{Top}_k(\pi)$ the set of the $k$ highest-ranked points under $\pi$. Two standard metrics are then commonly used.

**Top-$k$ overlap@$k$.** The overlap@$k$ between $\pi$ and $\pi'$ is defined as

$$\mathrm{Overlap@}k(\pi, \pi') := \frac{|\mathrm{Top}_k(\pi) \cap \mathrm{Top}_k(\pi')|}{k} \in [0, 1].$$

It measures the fraction of items remaining in the top-$k$ set when switching from one ranking to another.

**Top-$k$ Jaccard@$k$.** The Jaccard@$k$ similarity normalizes the overlap by the size of the union of the two sets:

$$\mathrm{Jaccard@}k(\pi, \pi') := \frac{|\mathrm{Top}_k(\pi) \cap \mathrm{Top}_k(\pi')|}{|\mathrm{Top}_k(\pi) \cup \mathrm{Top}_k(\pi')|} \in [0, 1].$$

It provides a scale-free measure of agreement, where $1$ indicates identical top-$k$ selections.

# D   ADDITIONAL FIGURES

## D.1   ADDITIONAL FIGURES FOR $K = 2$ BASE UTILITIES

In Section 3, we plot the spatial signatures for the WIND dataset (Figure 1) to illustrate the geometric mapping at the heart of our framework. Figures 5, 6, 7, 8, 9, 10 and 11 present the analogous plots for the remaining binary datasets introduced in Table 2.

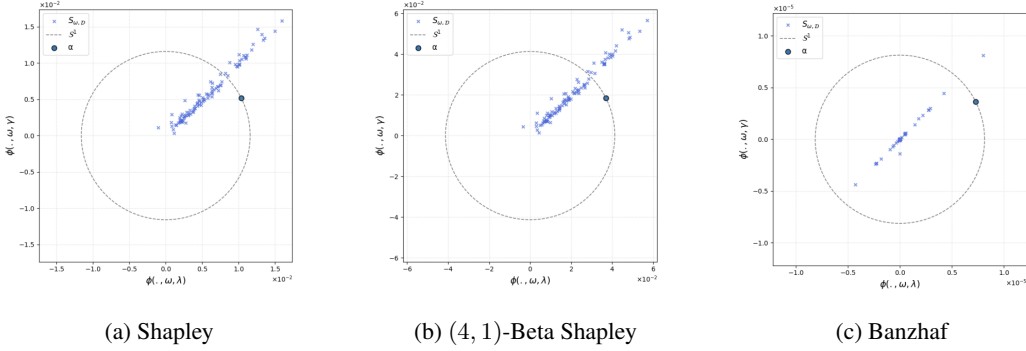

(a) Shapley        (b) $(4, 1)$-Beta Shapley        (c) Banzhaf

Figure 5: Spatial signature of the BREAST dataset for three semivalues (a) Shapley, (b) $(4, 1)$-Beta Shapley, and (c) Banzhaf. Each cross marks the embedding $\psi_{\omega,\mathcal{D}}(z)$ of a data point (with $u_1 = \lambda$, $u_2 = \gamma$), the dashed circle is the unit circle $\mathcal{S}^1$, and the filled dot indicates one utility direction $\bar{\alpha}$.

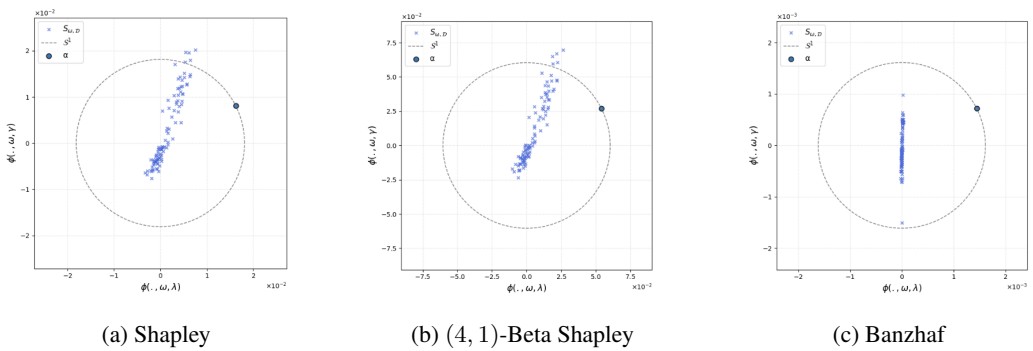

(a) Shapley        (b) $(4, 1)$-Beta Shapley        (c) Banzhaf

Figure 6: Spatial signature of the TITANIC dataset for three semivalues (a) Shapley, (b) $(4, 1)$-Beta Shapley, and (c) Banzhaf. Each cross marks the embedding $\psi_{\omega,\mathcal{D}}(z)$ of a data point (with $u_1 = \lambda$, $u_2 = \gamma$), the dashed circle is the unit circle $\mathcal{S}^1$, and the filled dot indicates one utility direction $\bar{\alpha}$.

## D.2   ADDITIONAL FIGURES FOR $K > 2$ BASE UTILITIES

In this section, we visualize, in three dimensions, the spatial signatures associated with the *utility trade-off experiments* for binary classification with $K = 3$ base utilities (Accuracy, F1, Recall) described in Appendix A.6.2. These plots (corresponding to Figures 12, 13, 14, 15, 16, 17, 18 and 19) provide the geometric counterpart of the robustness scores reported in Table 15.

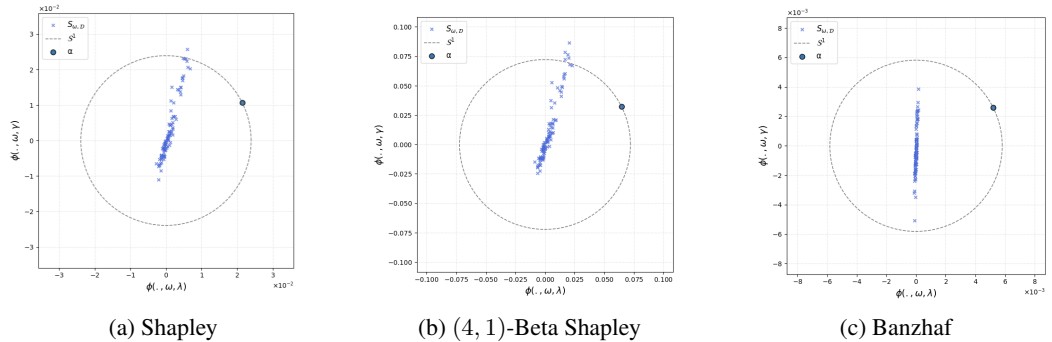

(a) Shapley        (b) $(4, 1)$-Beta Shapley        (c) Banzhaf

Figure 7: Spatial signature of the CREDIT dataset for three semivalues (a) Shapley, (b) $(4, 1)$-Beta Shapley, and (c) Banzhaf. Each cross marks the embedding $\psi_{\omega,\mathcal{D}}(z)$ of a data point (with $u_1 = \lambda$, $u_2 = \gamma$), the dashed circle is the unit circle $\mathcal{S}^1$, and the filled dot indicates one utility direction $\bar{\alpha}$.

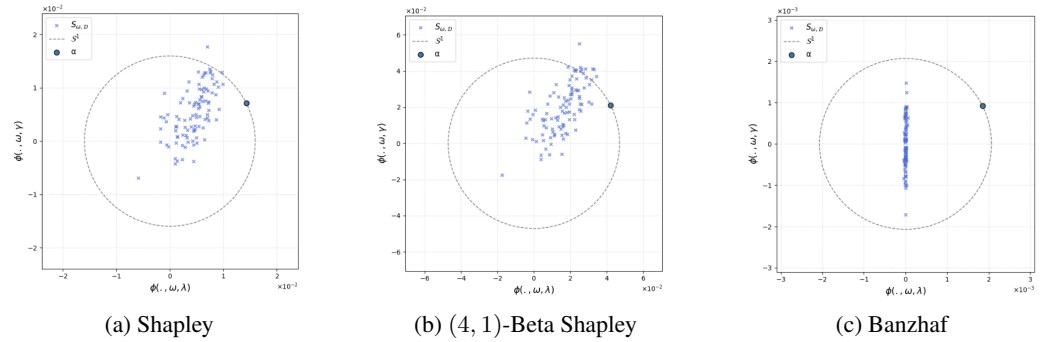

(a) Shapley        (b) $(4, 1)$-Beta Shapley        (c) Banzhaf

Figure 8: Spatial signature of the HEART dataset for three semivalues (a) Shapley, (b) $(4, 1)$-Beta Shapley, and (c) Banzhaf. Each cross marks the embedding $\psi_{\omega,\mathcal{D}}(z)$ of a data point (with $u_1 = \lambda$, $u_2 = \gamma$), the dashed circle is the unit circle $\mathcal{S}^1$, and the filled dot indicates one utility direction $\bar{\alpha}$.

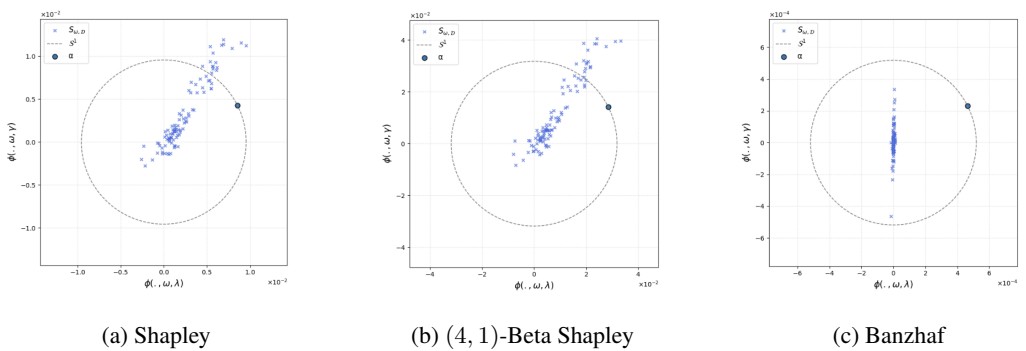

(a) Shapley        (b) $(4, 1)$-Beta Shapley        (c) Banzhaf

Figure 9: Spatial signature of the CPU dataset for three semivalues (a) Shapley, (b) $(4, 1)$-Beta Shapley, and (c) Banzhaf. Each cross marks the embedding $\psi_{\omega,\mathcal{D}}(z)$ of a data point (with $u_1 = \lambda$, $u_2 = \gamma$), the dashed circle is the unit circle $\mathcal{S}^1$, and the filled dot indicates one utility direction $\bar{\alpha}$.

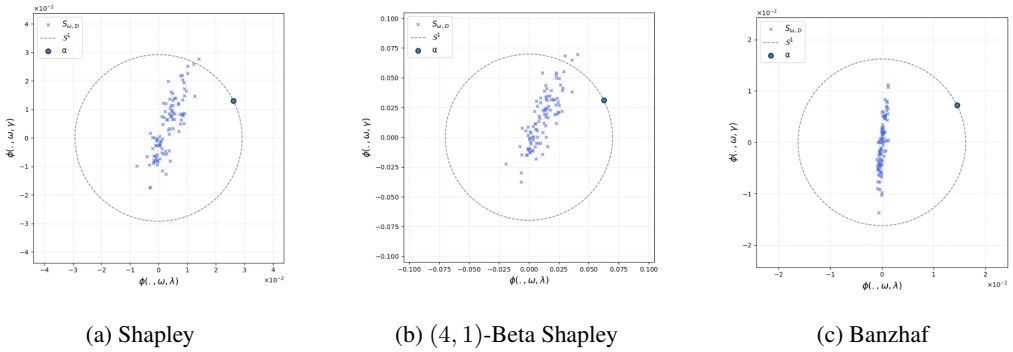

(a) Shapley        (b) $(4, 1)$-Beta Shapley        (c) Banzhaf

Figure 10: Spatial signature of the 2DPLANES dataset for three semivalues (a) Shapley, (b) $(4, 1)$-Beta Shapley, and (c) Banzhaf. Each cross marks the embedding $\psi_{\omega,\mathcal{D}}(z)$ of a data point (with $u_1 = \lambda$, $u_2 = \gamma$), the dashed circle is the unit circle $\mathcal{S}^1$, and the filled dot indicates one utility direction $\bar{\alpha}$.

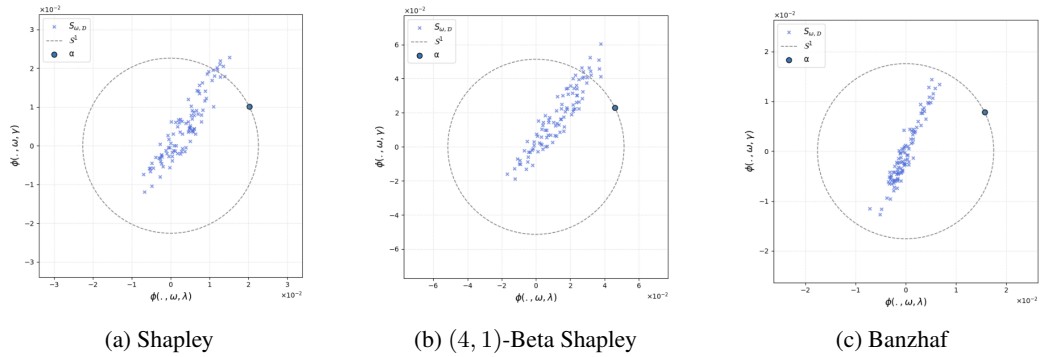

(a) Shapley        (b) $(4, 1)$-Beta Shapley        (c) Banzhaf

Figure 11: Spatial signature of the POL dataset for three semivalues (a) Shapley, (b) $(4, 1)$-Beta Shapley, and (c) Banzhaf. Each cross marks the embedding $\psi_{\omega,\mathcal{D}}(z)$ of a data point (with $u_1 = \lambda$, $u_2 = \gamma$), the dashed circle is the unit circle $\mathcal{S}^1$, and the filled dot indicates one utility direction $\bar{\alpha}$.

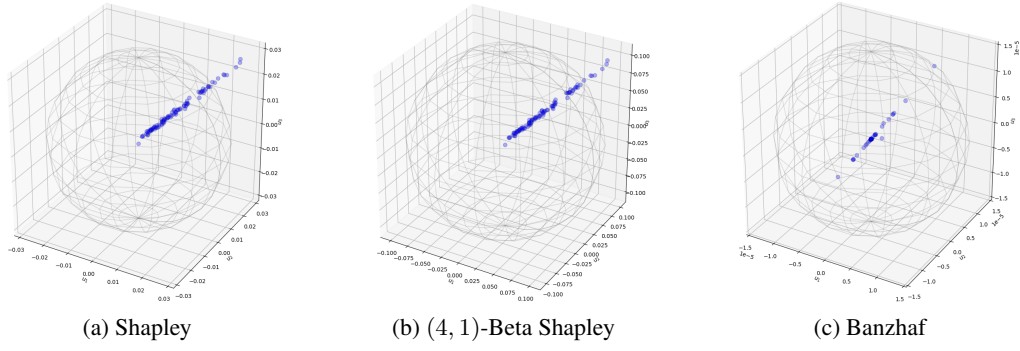

(a) Shapley        (b) $(4, 1)$-Beta Shapley        (c) Banzhaf

Figure 12: Spatial signature of the BREAST dataset for three semivalues (a) Shapley, (b) $(4, 1)$-Beta Shapley, and (c) Banzhaf. Each blue points marks the embedding $\psi_{\omega,\mathcal{D}}(z)$ of a data point $z \in \mathcal{D}$ (with $u_1 = $ accuracy, $u_2 = $ f1, $u_3 = $ recall). The represented sphere is $\mathcal{S}^2$.

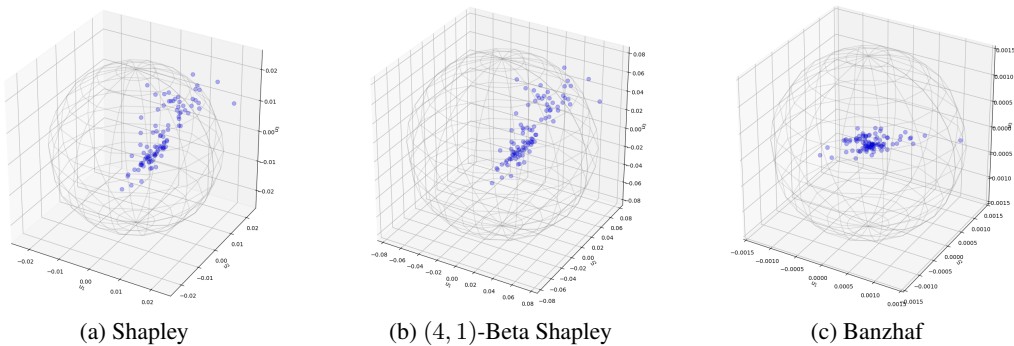

(a) Shapley     (b) $(4, 1)$-Beta Shapley     (c) Banzhaf

Figure 13: Spatial signature of the TITANIC dataset for three semivalues (a) Shapley, (b) $(4, 1)$-Beta Shapley, and (c) Banzhaf. Each blue points marks the embedding $\psi_{\omega,\mathcal{D}}(z)$ of a data point $z \in \mathcal{D}$ (with $u_1 =$ accuracy, $u_2 =$ f1, $u_3 =$ recall). The represented sphere is $\mathcal{S}^2$.

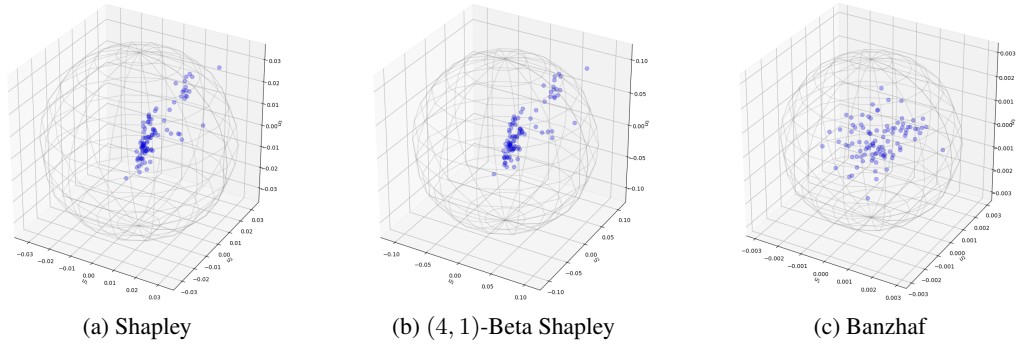

(a) Shapley     (b) $(4, 1)$-Beta Shapley     (c) Banzhaf

Figure 14: Spatial signature of the CREDIT dataset for three semivalues (a) Shapley, (b) $(4, 1)$-Beta Shapley, and (c) Banzhaf. Each blue points marks the embedding $\psi_{\omega,\mathcal{D}}(z)$ of a data point $z \in \mathcal{D}$ (with $u_1 =$ accuracy, $u_2 =$ f1, $u_3 =$ recall). The represented sphere is $\mathcal{S}^2$.

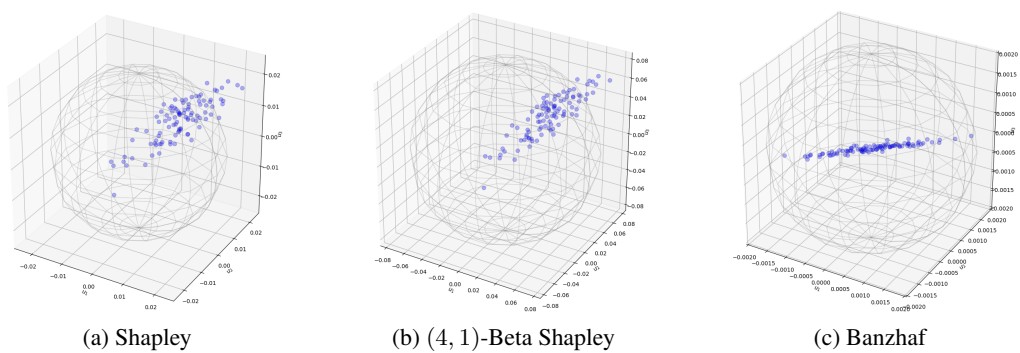

(a) Shapley     (b) $(4, 1)$-Beta Shapley     (c) Banzhaf

Figure 15: Spatial signature of the HEART dataset for three semivalues (a) Shapley, (b) $(4, 1)$-Beta Shapley, and (c) Banzhaf. Each blue points marks the embedding $\psi_{\omega,\mathcal{D}}(z)$ of a data point $z \in \mathcal{D}$ (with $u_1 =$ accuracy, $u_2 =$ f1, $u_3 =$ recall). The represented sphere is $\mathcal{S}^2$.

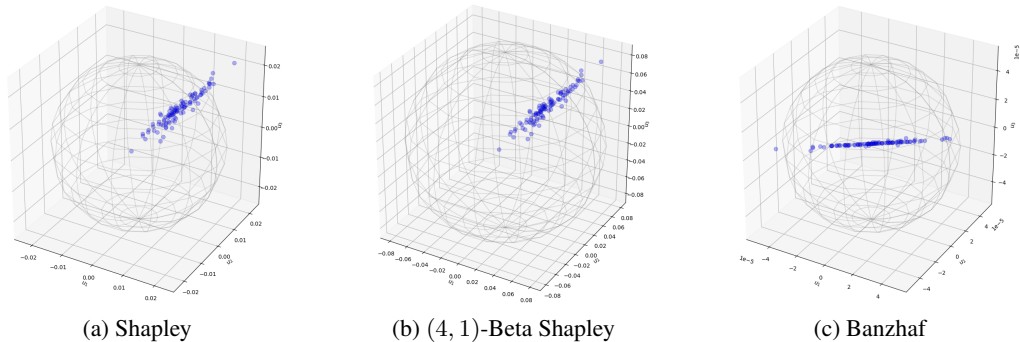

(a) Shapley      (b) $(4, 1)$-Beta Shapley      (c) Banzhaf

Figure 16: Spatial signature of the WIND dataset for three semivalues (a) Shapley, (b) $(4, 1)$-Beta Shapley, and (c) Banzhaf. Each blue points marks the embedding $\psi_{\omega,\mathcal{D}}(z)$ of a data point $z \in \mathcal{D}$ (with $u_1 =$ accuracy, $u_2 =$ f1, $u_3 =$ recall). The represented sphere is $\mathcal{S}^2$.

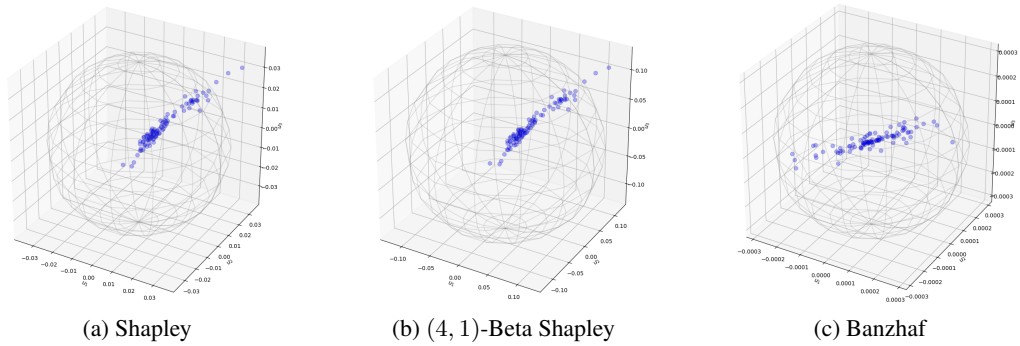

(a) Shapley      (b) $(4, 1)$-Beta Shapley      (c) Banzhaf

Figure 17: Spatial signature of the CPU dataset for three semivalues (a) Shapley, (b) $(4, 1)$-Beta Shapley, and (c) Banzhaf. Each blue points marks the embedding $\psi_{\omega,\mathcal{D}}(z)$ of a data point $z \in \mathcal{D}$ (with $u_1 =$ accuracy, $u_2 =$ f1, $u_3 =$ recall). The represented sphere is $\mathcal{S}^2$.

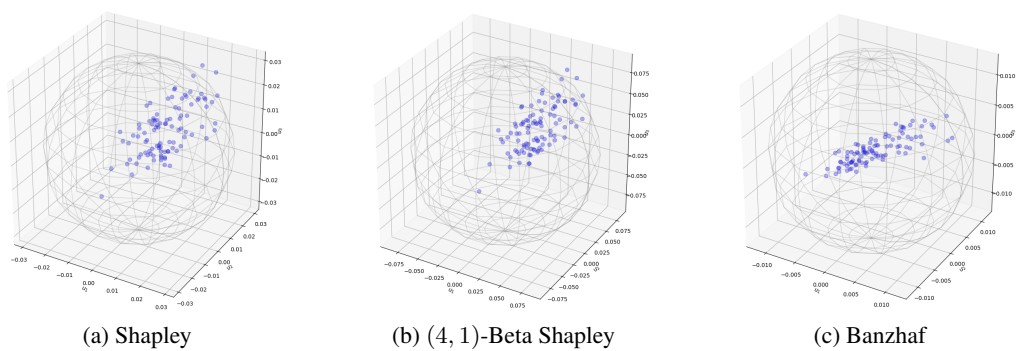

(a) Shapley      (b) $(4, 1)$-Beta Shapley      (c) Banzhaf

Figure 18: Spatial signature of the 2DPLANES dataset for three semivalues (a) Shapley, (b) $(4, 1)$-Beta Shapley, and (c) Banzhaf. Each blue points marks the embedding $\psi_{\omega,\mathcal{D}}(z)$ of a data point $z \in \mathcal{D}$ (with $u_1 =$ accuracy, $u_2 =$ f1, $u_3 =$ recall). The represented sphere is $\mathcal{S}^2$.

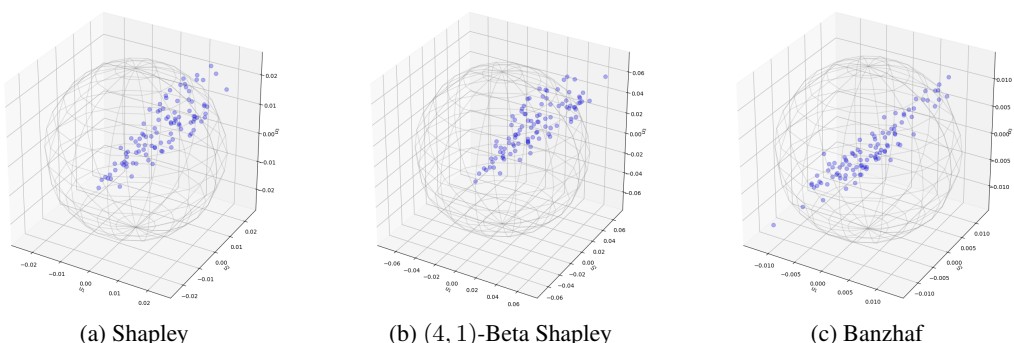

(a) Shapley         (b) $(4, 1)$-Beta Shapley         (c) Banzhaf

Figure 19: Spatial signature of the POL dataset for three semivalues (a) Shapley, (b) $(4, 1)$-Beta Shapley, and (c) Banzhaf. Each blue points marks the embedding $\psi_{\omega, \mathcal{D}}(z)$ of a data point $z \in \mathcal{D}$ (with $u_1 = $ accuracy, $u_2 = $ f1, $u_3 = $ recall). The represented sphere is $\mathcal{S}^2$.

