# OpenReview forum: "On the Impact of the Utility in Semivalue-based Data Valuation"
_ICLR.cc/2026/Conference — ICLR 2026 Poster_

### Official Review · Reviewer_t7pG · 2025-10-30

**Soundness:** 3
**Presentation:** 3
**Contribution:** 2
**Rating:** 4
**Confidence:** 3

**Summary:**

The authors propose a unified geometric framework based on the spatial signature and also the robustness metric, designed to help data valuation practitioners assess the sensitivity of semivalue-based data valuation results to the choice of utility in both the multiple-valid utility and utility trade-off settings. They evaluate the effectiveness of the proposed metric through experiments on multiple real-world datasets, considering both the multiple-valid-utility and utility trade-off scenarios.

**Strengths:**

Authors present another perspective on how the choice of the utility affects the resulting data values and rankings during data valuation.

To mitigate the noise and randomness introduced by Monte Carlo sampling, they propose the use of aligned sampling.

The theory and empirical setups are well-written and easy to follow.

Authors conduct multiple experiments to assess the efficacy of their metric, and also make their code available.

**Weaknesses:**

Restricting the utility to a 2D space spanned by two fixed base utilities u1 and u2 (and extension to class-wise utilities ) makes the robustness metric hard to scale.

It's possible that some information is lost when embedding data to a low dimensional space. It's unclear to me how this affects the results.

The data values from Monte Carlo approximations in regression problems tend to be unreliable due to greater variability in utility estimates when sample sizes are small. So, is there a specific reason why authors restricted the use of regression problems to utility trade-off scenarios, rather than including them in multiple-utility analyses?

There appears to be a misalignment between the claims made in the introduction and the empirical setup. In particular, the treatment of utility trade-offs in the experiments does not accurately reflect the notion of trade-off presented in the introduction.

There are very few results presented for the utility trade-off scenario, with none in the main paper and few in the appendix. Also, the results relegated to the appendix are neither discussed nor explained. For instance, why do Tables 9–14 only consider p=500? And why are only regression tasks used?

While the robustness metric could, in principle, address the question “How robust are my data valuation results to the choice of utility?”, it is unclear how distinct the observed patterns are from those obtained using Kendall or Spearman correlations. For instance, based on the results for both Kendall and the robustness metric, the cases where Banzhaf outperforms Shapley appear identical. The authors note that, in the absence of ties, the robustness metric “captures how far, in expectation, one must move from a utility direction before the Kendall rank correlation degrades by at least 2p/N.” While this is insightful, the results presented in the tables and figures reveal only broad trends, which seem largely similar across both metrics.



The authors briefly mention in the appendix that “one can alter the utility either by changing the algorithm A or by changing the performance metric,” an important consideration also discussed in several of the cited works. In my opinion, this is a strong consideration, and seeing how the robustness metric performs across datasets would have been helpful. However, the results presented in Table 15 are reported using Spearman correlation rather than the proposed robustness metric. Also, these results are not discussed, and key details of the experimental setup are not given, such as the number of runs.



The authors should consider adding a discussion section. Overall, the experimental results are insufficiently discussed, making it difficult to highlight the main contributions of the proposed metric, both theoretical and empirical.



Minor.
The references section needs a second look. Cite the published versions of the papers where they exist and not their arXivs. E.g., Jiachen T. Wang, Tianji Yang, James Zou, Yongchan Kwon, and Ruoxi Jia. Rethinking data shapley
for data selection tasks: Misleads and merits, ICML’24

**Questions:**

How does embedding the data point to a low dimensional space affects its value and ranking?

Is there a specific reason why authors typically restrict the use of regression problems to utility trade-off scenarios, rather than including them in multiple-utility analyses?

Why do Tables 9–14 only consider p=500?

How distinct are the observed patterns with the robustness metric from those obtained using Kendall or Spearman correlations? In what ways is the robustness metric a great addition to the data valuation practitioner’s toolbox?

In what ways do dataset characteristics, data preprocessing choices, and algorithmic choices, e.g., chosen classification/regression algorithms, affect the results?

Why restrict the multiple-valid-utility scenario to binary classification problems?

---

> ### Comment · Reviewer_t7pG · 2025-11-13
>
> Although the authors haven't made a rebuttal, I will raise my score after reading other reviewers' comments and in the hope that the authors address the concerns raised by all reviewers, both here and in the updated paper.

---

> ### Author Response · Authors · 2025-11-20
>
> We thank the reviewer for their time and feedback. We address their comments in the order raised: first, the identified weaknesses, then their questions. Throughout our answers, all references to sections, tables, or appendices refer to the paper revision submitted with this rebuttal, which incorporates all reviewer-suggested updates. A summary of these changes is provided in the general comment titled **Paper revision**.
>
> ### Weaknesses [1/2]
>
> **w.1.** The robustness metric $R_p$ is defined for any finite family of base utilities {$u_1,\dots,u_K$}, $K \ge 2$. By linearity, there exists $\psi_{\omega,D}(z)\in\mathbb{R}^K$ such that for every linear combination $u_\alpha = \sum_{k=1}^K \alpha_k u_k$ we have $\phi(z;\omega,u_\alpha) = \langle \psi_{\omega,D}(z), \alpha\rangle$. Thus $R_p$ is naturally defined over all directions $\bar\alpha \in \mathbb{S}^{K-1}$, not just in 2D. The 2D case (span of $(u_1,u_2)$) is used in the main text because it allows clear geometry and figures. The underlying construction and definition of $R_p$ do not change when $K>2$; only the ambient dimension of $\psi_{\omega,D}(z)$ and of the sphere of directions increases.
>
> However, we do agree that our experiments, in both the *multiple-valid utility* and *utility trade-off* scenarios focus on 2D. That is why, for completeness, we conduct the following experiments:
> - *(Multiple-valid utility scenario with $K > 2)$.* Using the analytical decomposition derived in Appendix C.5, each multiclass utility can be written as $u_{\alpha} = \sum_{c=1}^{C} a_c u_c$, where $u_c$ is a class-wise utility for class $c$. Particularly, in our experiment, we consider the per-class precision for $u_c$ that is defined in Appendix C.5. Hence $K = C$, the number of classes. We evaluate this setting on the three multiclass datasets used for the \emph{utility trade-off scenario} experiments (namely Digits, Wine, and Iris) and approximate $R_p$ by sampling directions $\alpha \in \mathcal{S}^{C-1}$ and applying the approximation scheme of Remark B.6.
> - *(Utility trade-off scenario with $K > 2$).* To study trade-offs beyond pairs of utilities, we consider three base utilities simultaneously, which allows us to visualize spatial signatures in 3D. Specifically, we consider Accuracy, F1, and Recall for binary classification; macro-F1, macro-Recall, and Accuracy for multiclass classification; and MSE, MAE, and $R^2$ for regression. We reuse the same datasets as in the $K=2$ experiments. For each task, we compute the 3D spatial signatures $S_{\omega, \mathcal{D}} \in \mathbb{R}^3$ and then approximate $R_p$ using the sampling scheme of Remark B.6.
>
> The corresponding empirical results are reported in Appendix A.7 for the *multiple-valid-utility* scenario with $K > 2$ (Table 18), and in Appendix A.6.2 for the *utility trade-off* scenario (Tables 15 and 17 for classification and Table 16 for regression, together with examples of 3D spatial signatures in Appendix D.2). Across all datasets and tasks, the same qualitative pattern observed for $K=2$ persists: Banzhaf value consistently achieves the highest robustness scores $R_p$. A more detailed discussion of these results is provided in Appendix A.10.
>
> **w.2.** See answer to Question 1 (**q.1**).
>
> **w.3.** See answer to Question 2. (**q.2**).
>
> **w.4.** In the paper, we define a *utility trade-off* as follows: in the 2D setting, given two utilities $u_A$ and $u_B$, a trade-off corresponds to considering the family {$u_\lambda = \lambda u_A + (1-\lambda)u_B : \lambda \in [0,1]$}, where different values of $\lambda$ encode different priorities between the two objectives. The helpfulness vs. harmlessness example in Section 1 is meant as a conceptual illustration of this general notion of balancing conflicting desiderata, not as the specific pair of utilities used in our experiments. In the empirical section (Section 4), we instantiate the same trade-off structure using standard classification and regression metrics. These pairs fit into the formal family {$u_\lambda$} studied in the theory. The reason we do not run the experiments on a helpfulness–harmlessness LLM setting is practical and reproducibility-related, not conceptual. In fact, evaluating $R_p$ requires semivalue computations, which involve repeated model retraining. In our current setup, even for small models and datasets, a semivalue estimation already takes non-trivial compute. Moreover, combining this with RLHF-style fine-tuning of large language models under a composite helpfulness/harmlessness objective would make the experiments even more computationally expensive.

---

> ### Author Response · Authors · 2025-11-20
>
> ### Weaknesses [2/2]
>
> **w.5.** We acknowledge that the presentation of the utility trade-off results may have been too condensed in the initial version. Section 4.2 already reports *utility trade-off* experiments for both regression and multiclass classification, summarizing the behavior of $R_p$ along convex combinations $u_\nu = \nu u_1 + (1-\nu)u_2$ for three regression pairs (MSE/MAE, MSE/$R^2$, MAE/$R^2$) on Diabetes, California housing, and Ames, and three multiclass pairs Accuracy/macro-F1, Accuracy/macro-Recall, macro-F1/macro-Recall) on Digits, Wine, and Iris. Due to space constraints, the numerical values are in Tables 9–14 of Appendix A.6, but the main qualitative conclusion (that Banzhaf achieves the highest $R_p$ across all settings) is stated in Section 4.2.
>
> Regarding the choice of $p = 500$, $p$ controls how much the ranking is allowed to change: $R_p$ measures the average distance in utility-direction space until $p$ pairwise swaps occur. Appendix B.4 shows that, in the no-ties case, $p$ swaps correspond to a drop of $2p/N$ in Kendall correlation, where $N = \binom{n}{2}$ and $n$ is the training set's size. For the training set sizes in our experiments ($n=100$ for classification and $n=300$ for regression), choosing $p=500$ places us in the regime of local but non-trivial perturbations of the ranking (a drop of about $0.2$ in Kendall rank correlation for $n=100$). In the *multiple-valid-utility* scenario, we already report $R_p$ for $p\in$ {$500$,$1000$,$1500$} (Figure 3 and Table 7), and observe that the relative ordering of semivalues is stable across $p$. Having established this, we fix $p=500$ in the trade-off tables to avoid a 3 times increase in table size while still reporting a representative, interpretable level of robustness.
>
> We agree that, under the 9-page limit, our discussion of the trade-off tables is brief. In the revised 9-page version, we have added an overall discussion section in Appendix A.10 that interprets the trade-off results (regression and classification) and connects them to our main claims (e.g., the consistently higher $R_p$ for Banzhaf). If the paper is accepted, the camera-ready version will have more space, and we will move a compact subset of these utility trade-off results and their discussion into the main paper.
>
> **w.6.** See answer to Question 4 (**q.4**).
>
> **w.7.** The experiment behind Table 15 (which corresponds to Table 19 in the paper revision) was intended as a complementary study to our work for robustness to algorithmic choices (e.g., logistic regression vs. MLP) under a fixed performance metric (which still corresponds to changing the utility). In that setting, there is no family of base metrics to average over; we are simply comparing the rankings induced by two discrete pipelines. For this reason, we reported Spearman correlations there, which are a standard tool for pairwise comparison of rankings. By contrast, $R_p$ is designed to summarize robustness over a whole family of utilities (all directions in the span of {$u_k$} rather than for a single pair.
>
> Moreover, Table 19 is discussed immediately after its presentation in Appendix A.8, where we highlight that Banzhaf exhibits the smallest variance across learning algorithms, a finding consistent with prior literature reporting that Banzhaf values fluctuate less under changes in model training conditions [1]. The table caption also specifies that all statistics are averaged over five runs.
>
> **w.8.** In the revised version of the 9-page paper, we have added an overall discussion section in Appendix A.10, which summarizes the main empirical findings across all settings (multiple-utility vs. trade-off, classification vs. regression, binary vs. multiclass), connects these findings to the theoretical properties of $R_p$, and highlights the practical implications of the results. If the paper is accepted, the camera-ready version will not be bound by the 9-page limit, and we will move this discussion section into the main paper.
>
> **w.9.** In our paper revision, we have modified the references section to cite published versions whenever available instead of arXiv preprints.

---

> > ### Author Response · Authors · 2025-11-20
> >
> > ### Questions [1/2]
> >
> > **q.1.** We clarify that what we embed is the vector of semivalue scores across base utilities. Formally, for base utilities {$u_1,\dots,u_K$}, and point $z \in \mathcal{D}$, we define
> > \begin{align*}
> >     \psi_{\omega,D}(z) = \big(\phi(z;\omega,u_1),\dots,\phi(z;\omega,u_K)\big) \in \mathbb{R}^K.
> > \end{align*}
> > This is just a reparameterization of the semivalue outputs. No information about ${\phi(z;\omega,u_k)}\_{k=1}^K$ is lost. For any linear combination $u\_\alpha = \sum_{k=1}^K \alpha\_k u\_k$, we have
> > \begin{align*}
> >     \phi(z;\omega,u\_\alpha) = \langle \psi\_{\omega,D}(z),\alpha\rangle.
> > \end{align*}
> > Hence, all rankings induced by all utilities in the span of ${u\_k}$ are exactly preserved by the embedding: ordering the points by $\phi(\cdot;\omega,u_\alpha)$ is identical to ordering them by the projection $\langle \psi_{\omega,D}(\cdot),\alpha\rangle$. The robustness metric $R_p$ only depends on these rankings, so its value is unaffected by the embedding. So, to directly answer the reviewer’s question: embedding into our utility space $\psi_{\omega,D}$ does not change the semivalue of any point under any utility in the span of ${u_k}$, and therefore does not change its ranking. The embedding is an exact representation of these values, not an approximation that discards information.
> >
> > **q.2.** Our framework is agnostic to the prediction task (classification or regression): all results are stated for arbitrary base utilities {$u_1, \cdots, u_K$}. The reason we restrict *multiple equally valid* utilities to classification in our experiments is that this scenario requires a common decomposition: we need a family of metrics that can all be written as linear combinations of a shared set of base quantities. For binary and multiclass classification, we explicitly exhibit such decompositions in Section 3 and Appendix C.5, which makes it natural to study robustness to shifts within a family of equally valid metrics in that setting. For regression, while the framework itself would apply as soon as one identifies a similar common base (e.g., a small set of shared sufficient statistics from which several equally valid regression metrics can be derived), we are not aware of a similarly natural, widely accepted decomposition covering a broad family of equally valid regression utilities. Because of this lack of a canonical common base, we chose to evaluate regression tasks only in the *utility trade-off* scenario, where the practitioner explicitly specifies a small set of base regression utilities to interpolate between.
> >
> > **q.3.** See answer to Weakness 5 (**w.5**)
> >
> > **q.4.** Rank correlation metrics (Kendall, Spearman) take as input two rankings (e.g., induced by $u$ and $u^{\prime}$) and output a similarity score. In the paper context, they answer the question: “How similar are the rankings under these two particular utilities?” In contrast, $R_p$ takes as input a family of utilities {$u_{\alpha}$} (all directions in the span of {$u_k$}) and returns a scalar that summarizes, in expectation over $\alpha$, how far one needs to move in utility space before the ranking changes by at least $2p/\binom{n}{2}$ in Kendall correlation for a dataset of size $n$. Thus, $R_p$ does not compare two chosen utilities; it measures global robustness to all shifts inside the utility family.
> >
> > In our experiments, when Banzhaf is more robust than Shapley, it is likely that this shows up both in (i) Kendall/Spearman between a few selected utilities and (ii) $R_p$, which integrates over the entire continuum of utilities in the span. This is exactly what the reviewer observes: the ordering (“Banzhaf $>$ Shapley” in robustness) is the same.
> >
> > For a practitioner, Kendall/Spearman are useful when there is a small, fixed set of candidate utilities (say, “Accuracy vs. F1”) and one wants to check stability between those specific choices. $R_p$ targets different and common settings: (i) when the practitioner knows that the true utility belongs to a continuous family (e.g., convex combinations of two metrics) or (ii) must pick arbitrarily among several equally reasonable utilities. In both regimes, it is infeasible to check all possible utilities pairwise. $R_p$ then provides: a single scalar summary of robustness to all such shifts and a scale parameter $p$ that gives the user a direct geometric interpretation (precisely the expected distance in utility space until a given drop in Kendall correlation), which makes it easy to interpret what more robust means.

---

> ### Author Response · Authors · 2025-11-20
>
> ### Questions [2/2]
>
> **q.5.** The computation of $R_p$ is always defined conditional on a learning context. For any fixed training dataset $\mathcal{D}$, preprocessing pipeline, and learning algorithm $\mathcal{A}$, the utilities have the form
> \begin{align*}
> u(S) = \mathrm{perf}(\mathcal{A}(S), D_{\text{test}}),
> \end{align*}
> and $R_p$ measures the robustness of semivalue-induced rankings within that fixed context. In our experiments, we deliberately fix these ingredients so as to isolate the effect of changing the utility.
>
> **Dataset characteristics.**
> Properties such as class imbalance, noise level, task difficulty, and sample size all influence the spatial signature $S_{\omega,\mathcal{D}} = {\psi_{\omega,\mathcal{D}}(z)}$ and, in turn, the resulting rank correlations and $R_p$. Empirically, this is precisely what we observe when comparing, for instance, datasets in Figures 1 and 3: some datasets yield highly collinear signatures and large $R_p$, while others produce more dispersed signatures and lower $R_p$. This dependence on dataset characteristics is expected and, in our view, desirable: $R_p$ is meant to inform the practitioner, for their specific dataset, how sensitive semivalue-based data valuation is to the utility choice.
>
> **Preprocessing choices.**
> All preprocessing steps (subsampling rules, train/test splits, feature preparation, and decision-threshold calibration for binary classification) are held fixed across utilities and semivalues, and fully documented in Appendix A.1. In particular, we calibrate the classification threshold to match the empirical class prevalence so that differences in rankings arise from the utility definition rather than arbitrary cutoffs. Different preprocessing pipelines (e.g., alternative feature scaling, resampling, or thresholding schemes) would change the learned model and thus the semivalue scores, so $R_p$ naturally inherits this dependence. Our design choice in this paper is to fix the preprocessing and learning pipeline so that the only moving part is the utility; in practice, practitioners can plug in their own preprocessing pipeline and recompute $R_p$ accordingly.
>
> **Algorithmic choices.**
> Similarly, in the main experiments we fix the learning algorithm $\mathcal{A}$, so that changing the utility amounts purely to changing $\mathrm{perf}$. Different algorithms (or hyperparameters) would induce different marginal contributions and therefore different spatial signatures and $R_p$ values. This dependence is illustrated, in Table 19, where switching between learning algorithms leads to changes in robustness.
>
> In summary, dataset characteristics, preprocessing, and algorithmic choices all affect the spatial signature and thus the numerical value of $R_p$. However, in our experiments, we fix these components on purpose to cleanly isolate and study the effect of the utility choice on robustness; but the framework itself is designed to reflect whatever learning pipeline the practitioner considers.
>
> **q.6.** The *multiple-valid-utility* scenario is not theoretically restricted to binary classification in our framework. Formally, it applies whenever a family of equally valid utilities can be expressed as linear combinations of a shared base of metrics. In the paper, we construct such bases for binary classification in Section 3 and for class-wise multiclass metrics in Appendix C.5. We agree that, in the initial empirical setup, we instantiated the *multiple-valid-utility* scenario only on binary classification datasets. This was a practical choice, not a theoretical limitation. In the revised version, we address this concern by adding *multiple-valid-utility* experiments for multiclass classification, using class-wise decompositions derived in Appendix C.5 to generate families of equally valid multiclass utilities (macro/weighted variants built from per-class metrics). The design of these new experiments is described in detail in our answer to Weakness 1 (**w.1**) above, and the corresponding empirical results are reported in Appendix A.7 (Table 18).
>
> ### References
>
> [1] Wang, J.T. &amp; Jia, R.. (2023). Data Banzhaf: A Robust Data Valuation Framework for Machine Learning. Proceedings of The 26th International Conference on Artificial Intelligence and Statistics, in Proceedings of Machine Learning Research 206:6388-6421 Available from https://proceedings.mlr.press/v206/wang23e.html.

---

> > ### Comment · Reviewer_t7pG · 2025-11-25
> >
> > Thank you for addressing my concerns both here and in the revised version. I raised my score

---

### Official Review · Reviewer_X13z · 2025-11-01

**Soundness:** 3
**Presentation:** 3
**Contribution:** 3
**Rating:** 6
**Confidence:** 5

**Summary:**

This paper investigates how sensitive semivalue-based data valuation scores are to the practitioner’s choice of utility function. It introduces the dataset spatial signature, an embedding in ℝ² where any candidate utility function corresponds to a linear functional. In this space, ranking data points with a semivalue reduces to projecting the dataset signature onto a direction that represents the chosen utility.

**Strengths:**

The paper is clearly written and easy to follow.

It presents an elegant and intuitive geometric representation that translates variations in utility functions into simple linear projections.

The proposed method is validated across a wide range of datasets and semivalues (Shapley, Beta Shapley, and Banzhaf).

The authors have thoroughly addressed the major limitations of the previous version by adding regression and multiclass tasks, introducing a formal robustness metric, and conducting experiments to assess algorithmic modifications.

**Weaknesses:**

I have reviewed this paper before, and the authors have resolved most of my confusion, but I still have a remaining question.

The experiments on multiple-valid-utility scenarios remain somewhat limited and would benefit from further expansion.

**Questions:**

1. Currently, empirical studies of multiple-valid-utility systems are mainly conducted in binary classification. It is also recommended to conduct multiple-valid comparisons of "multiple equally-valid indicators" in multi-class classification, rather than just trade-offs (e.g., macro-F1 vs macro-Recall vs Accuracy). 2. While $R_p$ provides a theoretically elegant proxy for stability under utility shifts, the paper would be significantly strengthened by connecting $R_p$ to downstream selection stability. It is recommended to report the relationship between top-k overlap@k/Jaccard@k and $R_p$.

---

> ### Author Response · Authors · 2025-11-20
>
> We thank the reviewer for their time and feedback. We address their comments in the order raised: first, the identified weaknesses, then their questions. Throughout our answers, all references to sections, tables, or appendices refer to the paper revision submitted with this rebuttal, which incorporates all reviewer-suggested updates. A summary of these changes is provided in the general comment titled **Paper revision**.
>
> ### Weaknesses
>
> **w.1.** We thank the reviewer for acknowledging previous revisions. We address their remaining questions below.
>
> **w.2.** See answer to Question 1 (**q.1**).
>
> ### Questions [1/2]
>
> **q.1.** We agree with the reviewer that the *multiple-valid utility* scenario is evaluated only for binary classification metrics. At the same time, it can be extended to multiclass classification metrics as explained in Appendix C.5. To address the reviewers' valid point directly, we use the analytical derivation in Appendix C.5 and conduct an additional experiment. Using the analytical decomposition derived in Appendix C.5, each multiclass utility can be written as $u_{\alpha} = \sum_{c=1}^{C} a_c u_c$, where $u_c$ is a class-wise utility for class $c$. Hence $K = C$, the number of classes. We evaluate this setting on three multiclass datasets and approximate $R_p$ by sampling directions $\alpha \in \mathcal{S}^{C-1}$ and applying the approximation scheme of Remark B.6. The corresponding empirical results are reported in Appendix A.7 (Table 18).
>
> Moreover, the reviewer rightfully suggests that our experiments, in both the *multiple-valid utility* and *utility trade-off* scenarios, focus on $K = 2$ base utilities. The new multiclass experiment remedies the previous restriction to $K=2$ in the *multiple-valid-utility* scenario by considering $K > 2$, where $K$ is equal to the number of classes in the datasets. To also address this valid point in the *utility-trade-off* scenario, we conduct additional experiments for both classification and regression. We consider three base utilities simultaneously, which allows us to visualize spatial signatures in 3D. Specifically, we consider Accuracy, F1, and Recall for binary classification; macro-F1, macro-Recall, and Accuracy for multiclass classification; and MSE, MAE, and $R^2$ for regression. We reuse the same datasets as in the $K=2$ experiments. For each task, we compute the 3D spatial signatures $S_{\omega, \mathcal{D}} \in \mathbb{R}^3$ and then approximate $R_p$ using the sampling scheme of Remark B.6. The corresponding empirical results are reported in Appendix A.6.2 (Tables 15 and 17 for classification and Table 16 for regression, together with examples of 3D spatial signatures in Appendix D.2). Across all datasets and tasks, the same qualitative pattern observed for $K=2$ persists: Banzhaf value consistently achieves the highest robustness scores $R_p$. A more detailed discussion of these results is provided in Appendix A.10.

---

> ### Author Response · Authors · 2025-11-20
>
> ### Questions [2/2]
>
> **q.2.** To address the reviewer’s request for a connection between $R_p$ and selection stability metrics, we provide both (1) an analytical link and (2) a dedicated empirical study.
>
> **(1) Analytical link.** Let $\mathcal{D}$ be a training set of size $n$. Let $u$ and $u^{\prime}$ be two utilities that induce rankings $\pi$ and $\pi^{\prime}$ on $\mathcal{D}$, and assume $\pi$ and $\pi^{\prime}$ differ by at most $p$ swaps (this is exactly the situation at distance $\rho_p(\bar\alpha)$ in Definition 3.2). Let
> \begin{align*}
>     S^{(k)}\_u := S^{(k)}\_{\phi(u,\omega)},\qquad S^{(k)}\_{u^{\prime}} := S^{(k)}\_{\phi(u^{\prime},\omega)}
> \end{align*}
> denote the top-$k$ sets under $u$ and $u^{\prime}$ respectively. In Appendix B.8, we derive the following deterministic bounds (for $p \le k$, otherwise they become vacuous) for both top-$k$ overlap@$k$ and Jaccard@$k$:
> \begin{align*}
>     \mathrm{Overlap}@k(u,u^{\prime}) = \frac{\lvert S^{(k)}\_u \cap S^{(k)}\_{u^{\prime}}\rvert}{k} \ge 1 - \frac{p}{k}.
> \end{align*}
> \begin{align*}
>     \mathrm{Jaccard}@k(u,u') = \frac{\lvert S^{(k)}\_u \cap S^{(k)}\_{u^{\prime}}\rvert}{\lvert S^{(k)}\_u \cup S^{(k)}\_{u^{\prime}}\rvert} \ge \frac{k-p}{k+p}.
> \end{align*}
> Now, by Definition 3.2, for each utility direction $\bar\alpha\in\mathbb{S}^{K-1}$, $\rho_p(\bar\alpha)$ is the minimal geodesic distance such that moving by $\rho_p(\bar\alpha)$ on the sphere produces exactly $p$ swaps in the ranking induced by $\bar\alpha$. So, for a fixed $p$ and $k$, a larger $R_p \propto \mathbb{E}_{\bar{\alpha}}[\rho_p(\bar{\alpha})]$ means that one must move farther in utility space before reaching a regime where top-$k$ overlap/Jaccard can be as low as these bounds.
>
> **(2) Empirical study.** We complement this analysis with an empirical study that directly relates $R_p$ to these metrics in the same spirit as our comparison between $R_p$ and rank correlation measures (Kendall and Spearman) in Section 4.1. We consider the family of binary classification utilities spanned by base utilities $(\lambda, \gamma)$ used to compute $R_p$ in Section 4.1. For one pair of distinct metrics in this family, precisely Accuracy vs. F1 score, we compute the rankings induced by the corresponding semivalue and, for different values of $k$, we evaluate the associated top-$k$ overlap@$k$ and Jaccard@$k$ between the two rankings. This yields, for each dataset and semivalue, a collection of top-$k$ stability scores that summarize how sensitive top-$k$ selections are to switching between these utilities. We then compare the top-$k$ stability scores in Table 20 with the robustness scores $R_p$ in Figure 3 and Table 7.
>
> Across most settings, we observe the same qualitative pattern as with Kendall and Spearman correlation metrics: semivalues that achieve larger $R_p$ for a given utility family (typically Banzhaf) also exhibit higher overlap@$k$ and Jaccard@$k$ when comparing the induced rankings under different utilities in that family. In other words, methods that are more robust according to $R_p$ are also those whose top-$k$ selections change the least when moving between these equally valid utility choices.

---

### Official Review · Reviewer_MwCd · 2025-11-03

**Soundness:** 3
**Presentation:** 2
**Contribution:** 3
**Rating:** 8
**Confidence:** 2

**Summary:**

This paper provides an in-depth study of stability of semi-value based data valuation methods when there are shifts in utility. The authors  introduce the concept of spatial signature, where a training data point is projected on to a low-dimensional space. The authors assess the robustness of semi-value using a geometric representation of the utility space, specifically a unit circle, such that the metric quantifies how much utility direction must be rotated before the ranking undergoes a significant change. The authors use a geodesic angular distance over the unit circle, and the robustness metric Rp is measured as normalized minimal geodesic distance of rotation needed to achieve p swaps in ranking. The paper offers an evaluation of their metric against data valuation methods Data Shapley, Data Banzhaf and Beta Shapley.

**Strengths:**

1. The introduction of a novel geometric perspective on and a formal robustness metric for semi-value based data valuation offers a way to benchmark data valuation frameworks and test their stability. In real-world datasets, the utility function can change anytime, reinforcing the importance of this metric.
2.  The authors provide a rigorous evaluation of their metric on existing data valuation methods and utility functions.
3. The paper also offers insights into why some valuation methods tend to collinearize the spatial signature and in doing so achieve higher robustness to utility shifts.
4. The robustness metric proposed is shown to be computationally efficient via the closed-form expression for the expected minimal geodesic distance needed to induce p pairwise swaps in ranking.

**Weaknesses:**

1. The paper focuses on utility functions specific to binary classification and raises the question of whether the findings will translate to other utility functions / learning tasks. Similarly the evaluations focus on two utility systems, and multi-utility systems are yet to be explored.
2. Benchmarking the stability of data valuation methods is not new. The key contribution of the paper is the geometric perspective - and this paper would be strengthened by additional experiments that show the true value of this geometric nature of the stability problem. While this paper discusses collinearity of spatial signatures leading to better robustness, the experimental validations focus on the robustness metric and not the geometric properties (eg. how the spatial signatures are distributed for a given valuation framework / dataset or the practical impact of this collinearity in geometric structure).

**Questions:**

1. I’d be interested in the authors' perspective on whether the robustness metric reflects an inherent property of the data valuation framework, or if it is more influenced by the dataset and the choice of utility function. For example, in imbalanced binary classification tasks, the rankings generated by different utility functions (e.g., F1 score vs. accuracy) can vary substantially. How does this variation impact the stability of data rankings and the robustness metric? And if it is affected by dataset and utility functions, then in what setting do the authors advice using this robustness metric in practice ?

2. How do the authors propose choosing p (especially as a factor of the size of the training dataset) for a given dataset?

---

> ### Author Response · Authors · 2025-11-20
>
> We thank the reviewer for their time and feedback. We address their comments in the order raised: first, the identified weaknesses, then their questions. Throughout our answers, all references to sections, tables, or appendices refer to the paper revision submitted with this rebuttal, which incorporates all reviewer-suggested updates. A summary of these changes is provided in the general comment titled **Paper revision**.
>
> ### Weaknesses [1/2]
> **w.1.** The paper does not actually focus on utility functions tailored for binary classification. Our methodological contributions (the spatial signature and the robustness metric $R_p$) are defined for arbitrary collections of base utilities {$u_1, \cdots, u_K$}. They are instantiated in both regression (in the utility trade-off experiments of Section 4.2) and multiclass settings (in both Appendix C.5, where macro/weighted metrics are written in terms of per-class utilities, and in the utility trade-off experiments of Section 4.2). However, we agree that we empirically illustrate the *multiple-valid-utility* scenario only in the binary classification setting (Section 4.1), where we consider equally valid linear-fractional utilities. But in theory, it is not limited to it. It applies to any setting where the set of candidate valid utilities can be written as linear combinations of base utilities $u_{\alpha} = \sum_{k=1}^{K} \alpha_k u_k$. If such a decomposition exists, our analysis, which revolves around the spatial signature and the robustness metric, is valid for the considered setting.
>
> Also, the reviewer is correct that our empirical results focus on $K=2$ base utilities in both the *multiple-valid-utility* scenario and *utility trade-off* scenario. However, it is essential to recall that our theoretical framework is fully general to $K \ge 2$ base utilities. Simply, in the case where we have $K > 2$ base utilities, we are only able to approximate the robustness metric $R_p$ (there is no easy to compute closed-form comparing to $K = 2$). The approximation procedure to compute $R_p$ on the the hypersphere $S^{K-1}$ for $K>2$ is given in Remark B.6 with convergence guarantees.
>
> To directly address the reviewer’s valid points and empirically evaluate the *multiple-valid-utility* scenario in settings beyond binary classification, as well as to explore the case $K>2$ for both scenarios, we conduct the following experiments:
>  - *(Multiple-valid utility scenario with $K > 2$)*. Using the analytical decomposition derived in Appendix C.5, each multiclass utility can be written as $u_{\alpha} = \sum_{c=1}^{C} a_c u_c$, where $u_c$ is a class-wise utility for class $c$. Hence $K = C$, the number of classes. We evaluate this setting on three multiclass datasets and approximate $R_p$ by sampling directions $\alpha \in \mathcal{S}^{C-1}$ and applying the approximation scheme of Remark B.6.
> - *(Utility trade-off scenario with $K > 2$)*. To study trade-offs beyond pairs of utilities, we consider three base utilities simultaneously, which allows us to visualize spatial signatures in 3D. Specifically, we consider Accuracy, F1, and Recall for binary classification; macro-F1, macro-Recall, and Accuracy for multiclass classification; and MSE, MAE, and $R^2$ for regression. We reuse the same datasets as in the $K=2$ experiments. For each task, we compute the 3D spatial signatures $S_{\omega, \mathcal{D}} \in \mathbb{R}^3$ and then approximate $R_p$ using the sampling scheme of Remark B.6.
>
> The corresponding empirical results are reported in Appendix A.7 for the *multiple-valid-utility* scenario with $K > 2$ (Table 18), and in Appendix A.6.2 for the *utility trade-off* scenario (Tables 15 and 17 for classification and Table 16 for regression, together with examples of 3D spatial signatures in Appendix D.2). Across all datasets and tasks, the same qualitative pattern observed for $K=2$ persists: Banzhaf value consistently achieves the highest robustness scores $R_p$. A more detailed discussion of these results is provided in Appendix A.10.

---

> > ### Author Response · Authors · 2025-11-20
> >
> > ### Weaknesses [2/2]
> >
> > **w.2.** To the best of our knowledge, benchmarking the stability of data valuation methods is recent and has not been carried out in the way we propose here. Most prior work has focused on one specific source of instability rather than providing a general robustness measure. For example, [1] studies robustness to inherent randomness in the learning algorithm. Still, it does not address robustness to changes in the utility function itself or to whole families of utilities. More recently, [2] shows that changing the utility can make Data Shapley unreliable for data selection, but this is again not analyzed using a general, task-agnostic robustness metric.
> >
> > We also wish to clarify that the robustness metric $R_p$ is, by its construction, a direct synthesis of the geometric perspective. It is computed from the spatial signature's arrangement on the hypersphere $S^{K-1}$, where the cut hyperplanes partition the sphere into ranking regions. Therefore, analyzing robustness via $R_p$ is inherently a geometric analysis. To further satisfy the reviewer's request to provide additional analysis that more directly connects the geometric properties of spatial signatures to the ranking stability against utility change, we propose to use the already in-hand spatial signatures we present in Figures 1 and 5-11 and study two additional geometric descriptors:
> > 1. $\lambda_{\max}:= \max_{k} \lambda_k \in ]0, \pi]$ which is the largest angular distance one can rotate the utility direction without inducing a single pairwise swap. This measures the best-case local stability on $\mathcal{S}^1$.
> > 2. $\eta:= \frac{\lambda_{\max}}{2\pi} \in ]0,1/2]$ which is the maximum fraction of all possible utility directions for which the ranking remains completely unchanged.
> >
> > Results are reported in Table A below.
> >
> > **Table A. Geometric descriptors $(\lambda_{\max}, \eta)$ ($\pm$ standard errors over $5$ Monte Carlo approximations) computed for each spatial signature (i.e., each dataset and each semivalue) mentioned in Section 4.1. Boldface entries indicate, for each dataset, the largest value of each descriptor across semivalues.**
> >
> > | Dataset   | Semivalue   | $\lambda_{\max}$ (rad)       | $\eta$            |
> > |-----------|-------------|-------------:|-------------:|
> > | breast    | Shapley     | 0.036 $\pm$ 0.003 | 0.006 $\pm$ 0.001 |
> > |           | Banzhaf     | **0.182 $\pm$ 0.027** | **0.029 $\pm$ 0.004** |
> > |           | (4,1)-Beta Shapley  | 0.040 $\pm$ 0.006 | 0.006 $\pm$ 0.001 |
> > | titanic   | Shapley     | 0.030 $\pm$ 0.004 | 0.005 $\pm$ 0.001 |
> > |           | Banzhaf     | **0.122 $\pm$ 0.021** | **0.019 $\pm$ 0.003** |
> > |           | (4,1)-Beta Shapley  | 0.032 $\pm$ 0.003 | 0.005 $\pm$ 0.001 |
> > | credit    | Shapley     | 0.042 $\pm$ 0.006 | 0.007 $\pm$ 0.001 |
> > |           | Banzhaf     | **0.168 $\pm$ 0.009** | **0.027 $\pm$ 0.001** |
> > |           | (4,1)-Beta  | 0.050 $\pm$ 0.010 | 0.008 $\pm$ 0.002  |
> > | heart     | Shapley     | 0.012 $\pm$ 0.001 | 0.002 $\pm$ 0.001 |
> > |           | Banzhaf     | **0.073 $\pm$ 0.006** | **0.012 $\pm$ 0.001** |
> > |           | (4,1)-Beta Shapley  | 0.012 $\pm$ 0.001 | 0.002 $\pm$ 0.001 |
> > | wind      | Shapley     | 0.016 $\pm$ 0.001 | 0.003 $\pm$ 0.001 |
> > |           | Banzhaf     | **0.329 $\pm$ 0.092** | **0.052 $\pm$ 0.015** |
> > |           | (4,1)-Beta Shapley  | 0.019 $\pm$ 0.002 | 0.003 $\pm$ 0.001 |
> > | cpu       | Shapley     | 0.021 $\pm$ 0.001 | 0.003 $\pm$ 0.001 |
> > |           | Banzhaf     | **0.069 $\pm$ 0.008** | **0.011 $\pm$ 0.001** |
> > |           | (4,1)-Beta Shapley  | 0.020 $\pm$ 0.002 | 0.003 $\pm$ 0.001 |
> > | 2dplanes  | Shapley     | 0.019 $\pm$ 0.001 | 0.003 $\pm$ 0.001 |
> > |           | Banzhaf     | **0.043 $\pm$ 0.005** | **0.007 $\pm$ 0.001** |
> > |           | (4,1)-Beta Shapley  | 0.019 $\pm$ 0.003 | 0.003 $\pm$ 0.001 |
> > | pol       | Shapley     | 0.022 $\pm$ 0.002 | 0.003 $\pm$ 0.001 |
> > |           | Banzhaf     | **0.026 $\pm$ 0.003** | **0.004 $\pm$ 0.001** |
> > |           | (4,1)-Beta Shapley  | 0.023 $\pm$ 0.002 | 0.004 $\pm$ 0.001 |
> >
> >
> > Across all datasets and semivalues, the geometric descriptors $\lambda_{\max}$ and $\eta$ exhibit the same qualitative behavior as the robustness scores $R_p$ reported in Figure 1 and Table 7. In particular, Banzhaf often achieves values of $\lambda_{\max}$ and $\eta$ that are one order of magnitude larger than those of Shapley and $(4,1)$-Beta Shapley. This indicates that, for Banzhaf, the geometry of spatial signatures likely implies wider angular regions in which no pairwise swap occurs, meaning that the ranking is preserved under a much larger set of utility directions. These geometric patterns directly mirror the robustness trends captured by $R_p$.

---

> ### Author Response · Authors · 2025-11-20
>
> ### Questions
>
> **q.1.** The robustness metric $R_p$ does not depend on any single utility choice. The base utilities $(u_1, \cdots, u_K)$ define the coordinate system of the embedding space $\mathbb{R}^K$. These are chosen based on the scenario: in the *utility trade-off* scenario, they are the fixed criteria to be balanced; in the *multiple-valid-utility* scenario, they form a basis that spans the space of considered metrics (e.g., $u_1 = \lambda, u_2 = \gamma$ for linear fractional performance metrics). The spatial signature $\mathcal{S}\_{\omega,\mathcal{D}}$ is computed once from the semivalue $\omega$ and dataset $\mathcal{D}$. Each utility that can be written as a linear combination of those base utilities $(u_{\alpha} = \sum_{k=1}^{K} \alpha_k u_k$) corresponds to a direction $\bar{\alpha} \in \mathcal{S}^{K-1}$. The robustness metric $R_p$ is then computed by analyzing how the data values' ranking changes as we vary $\bar{\alpha}$ over all possible directions on $\mathcal{S}^{K-1}$. Therefore, $R_p$ is a property of the triplet (dataset, semivalue, base utility space), not of any particular utility choice. It measures the inherent stability of the data valuation framework across the entire family of utilities defined by the base utility space.
>
> Regarding imbalanced classification, the variation between F1-score and accuracy rankings is exactly the motivating example of the *multiple-valid utility* scenario presented in the introduction (Section 1), and it is what our framework can analyze geometrically. When the family of utilities to which F1 and accuracy belong (corresponding to base utilities $(\lambda, \gamma)$) spans directions that induce diverse rankings, the spatial signature becomes scattered, leading to low $R_p$. When it produces similar rankings, the signature is nearly collinear, leading to a high $R_p$. Therefore, F1 and accuracy are likely to produce similar rankings when $R_p(S_{\omega, \mathcal{D}})$ is high and inversely. We recall that the term $S_{\omega, \mathcal{D}}$ denotes here the spatial signature for base utilities $(\lambda, \gamma)$, a semivalue $\omega$, and a dataset $\mathcal{D}$.
>
> We advise practitioners to use $R_p$ precisely when they face ambiguity in utility specification, whether their utility choice is a trade-off between criteria with a weighting that is not absolute in time (*utility-trade-off* scenario) or whether they chose their utility among multiple legitimate metrics (*multiple-valid-utility* scenario). In those cases, $R_p$ reveals how much the valuation conclusions depend on the arbitrariness of their choice in terms of weighting or the metric itself.
>
> **q.2.** This question relates directly to Appendix B.4, where we make the connection between the robustness metric $R_p$, the number of region crossings $p$, and the Kendall rank correlation between the resulting rankings. As the reviewer rightly suggests, $p$ should not be chosen in absolute terms but rather relative to the size of the training dataset. As shown in Appendix B.4, when there are no ties, the Kendall rank correlation between two rankings can be written as
> \begin{align*}
>         \tau = 1 - \frac{2D}{N},
> \end{align*}
> where $D$ is the number of discordant pairs and $N = \binom{n}{2}$. Crossing one ranking region swaps one pair, so $D$ increases by one and $\tau$ decreases by $2/N$. After $p$ swaps, the correlation has dropped from $1$ to
> \begin{align*}
>         \tau = 1 - \frac{2p}{N}.
> \end{align*}
> Thus, choosing $p$ is equivalent to choosing a tolerated degradation in Kendall correlation $\tau$. For a dataset of size $n$, we propose to:
> 1. Fix a minimum acceptable correlation $\tau_{\min}$ that defines when two rankings are similar enough for the application (e.g., $\tau_{\min} = 0.8$).
> 2. Set $p = \frac{1 - \tau_{\min}}{2} N = \frac{1 - \tau_{\min}}{2} \binom{n}{2}$.
>
> This automatically scales $p$ with $n$. Concretely, for $n=100$, our experimental choices $p \in$ {$500$, $1000$, $1500$} correspond roughly to allowing a Kendall drop of about $0.2$, $0.4$, and $0.6$, respectively. This is why we interpret $R_{500}$, $R_{1000}$, and $R_{1500}$ as measuring robustness at increasingly larger allowed changes in the ranking.
>
> ### References
>
> [1] Wang, J.T. &amp; Jia, R.. (2023). Data Banzhaf: A Robust Data Valuation Framework for Machine Learning. Proceedings of The 26th International Conference on Artificial Intelligence and Statistics, in Proceedings of Machine Learning Research 206:6388-6421 Available from https://proceedings.mlr.press/v206/wang23e.html.
>
> [2] Wang, J.T., Yang, T., Zou, J., Kwon, Y. &amp; Jia, R.. (2024). Rethinking Data Shapley for Data Selection Tasks: Misleads and Merits. Proceedings of the 41st International Conference on Machine Learning, in Proceedings of Machine Learning Research 235:52033-52063 Available from https://proceedings.mlr.press/v235/wang24cg.html.

---

### Author Response · Authors · 2025-11-20
**Paper revision**

We thank all reviewers for their valuable feedback. In addition to the individual rebuttals, we have submitted a revised version of the paper, with **all changes highlighted in blue**. Below, we summarize the main modifications, with the specific weaknesses and questions they address.
1. *(Reviewer MwCd [w.1], X13z [w.2, q.1] and t7pG [w.1, q.6]).*
We have extended the *multiple-valid-utility* scenario experiments to multiclass classification (and thus to $K > 2$ base utilities). Experiment details are provided in Appendix A.7, and the corresponding robustness scores are reported in Table 18. These new experiments are mentioned in the main paper (Section 4.1).
2. *(Reviewer MwCd [w.1], X13z [w.2, q.1] and t7pG [w.1]).* We have extended the *utility trade-off* scenario experiments to $K > 2$ base utilities in binary classification, multiclass classification, and regression settings. Experiment details are provided in Appendix A.6.2, and the corresponding robustness scores are reported in Tables 15-17, with 3D spatial signatures illustrated in Appendix D.2 (Figures 12-19). These new experiments are mentioned in the main paper (Section 4.2).
3. *(Reviewer X13z [q.2]).* We have added formal definitions of two top-$k$ stability metrics (overlap@$k$ and Jaccard@$k$) in Appendix C.6. We also have derived an analytical connection between these metrics and the robustness metric $R_p$ in Appendix B.8. We have complemented this with an empirical study detailed in Appendix A.9. The corresponding results are reported in Table 20.
4. *(Reviewer t7pG [w.5, w.8]).* We have added a discussion section in Appendix A.10 that provides an overall synthesis of the empirical robustness results, clarifies how $R_p$ aligns with rank-based and top-$k$ stability metrics, and compares the behavior of different semivalues across all experimental settings.
5. *(Reviewer t7pG [w.9]).* We have updated the references section to cite published versions whenever available instead of arXiv preprints.

---

### Author Response · Authors · 2025-12-01
**Contextual note for the Area Chair**

This note is shared respectfully as contextual information that may assist the Area Chair in their evaluation.

Reviewer t7pG increased their score from 4 to 6 during the discussion period, as indicated in their public comments. Since the system has reverted all scores to their pre-discussion state, this change is no longer reflected in the current ratings.

This information is offered only as context, with full understanding that the Area Chair will use their own judgment and may disregard it if it is not relevant. We thank the Area Chair for their time and engagement.

---

### Meta-Review · Area_Chair_6m71 · 2025-12-30

**Summary:**

The authors propose a geometric framework to analyze the robustness of semivalue-based data valuation to the choice of utility function. In particular, the authors propose a geometric formulation based on spatial signatures and a robustness metric.

The Reviewers agree that the proposed approach is technically sound. The geometric interpretation provides a mechanism to analyze utility-induced ranking instability. Its main strengths lie in the geometric perspective and broad empirical evaluation across datasets and semivalues. The empirical finding results on the higher robustness of Banzhaf over Shapley methods are interesting for the community. In the rebuttal, the authors also provide further empirical experiments for multiclass and higher-dimensional utility spaces.

One of critical concerns from the Reviewers are the weak empirical supports for multiple-valid-utility, and beyond binary classification settings. We strongly urge the authors to describe the related theoretical finding results and supporting empirical results into the main manuscript rigorously.

Overall, the submission mainly offers a principled analysis in robustness aspects of data valuation. We think the submission is on the borderline, which may be accepted if the slots are still available.

**Reviewer Concerns:**

The Reviewers have some following concerns:

+ Reviewer MwCd: limitation on utility functions specific to binary classification; mainly for two utility systems settings, limitation for multiple utility beyond two systems settings; weak experimental settings; effects on imbalanced binary classification tasks; hyperparameters;

+ Reviewer X13z: empirical supporting results on multiple-valid-utility scenarios are limited; relation between the proposed robustness metric and evaluated metric;

+ Reviewer t7pG: Restricting the utility to a 2D space; misalignment between the claims made in the introduction and the empirical setup; beyond binary classification problems?

**Reviewer Scores:**

In the rebuttal, the authors provide additional empirical results on utility trade-off scenario with $K=3$, geometric descriptors. Notice that one of critical concerns of the Reviewers is the multiple-valid-utility setup (beyond $K > 2$) and supporting empirical results beyond binary classification. We think the authors may address such concern at a certain level. It is better if the authors provide theoretical finding results corresponding to $K>2$ rigorously; and supporting empirical results for such settings rigorously.

---

### Decision · Program_Chairs · 2026-01-26

Accept (Poster)